# Minimax learning rates for estimating binary classifiers under margin conditions

## Abstract

We study classification problems using binary estimators where the decision boundary is described by horizon functions and where the data distribution satisfies a geometric margin condition. A key novelty of our work is the derivation of lower bounds for the worst-case learning rates over broad classes of functions, under a geometric margin condition—a setting that remains theoretically challenging. Moreover, we work in the noiseless setting, where lower bounds are particularly hard to establish. Our general results cover, in particular, classification problems with decision boundaries belonging to several classes of functions: for Barron-regular functions, Hölder-continuous functions, and convex-Lipschitz functions with strong margins, we identify optimal rates close to the fast learning rates of $\mathcal{O}(n^{-1})$ for $n \in \mathbb{N}$ samples.

**Keywords:** Neural networks, binary classification, learning bounds, entropy, margin.

**Mathematics Subject Classification:** 68T05, 62C20, 41A25, 41A46.

## 1 Introduction

How well can we solve classification problems with complex decision boundaries in deep learning? A lot of emphasis has been put on the noise and the decision boundary in the problem. However, in practice, data sets may have very strong margins (formally defined in C3) below) between the classes, which makes learning much simpler (see e.g. Figure 1). The presence of a margin seriously complicates the identification of lower bounds on learning success. This is intuitively clear, since in the extreme case, where certain regions between the classes almost surely do not contain any data points, many decision boundaries are valid. In this work, we overcome these issues and present lower bounds on learning under margin conditions.

### 1.1 Statistical framework for binary classification

We consider $n \in \mathbb{N}$ samples $\boldsymbol{S}_n := ((\boldsymbol{x}_i, y_i))_{i=1}^n$, where $\boldsymbol{x}_i \in \mathcal{X} := [0,1]^d$ with $d \in \mathbb{N}_{\geq 2}$, are input vectors; and $y_i \in \{0,1\}$ are class labels. We assume the random variables $(\boldsymbol{x}_i, y_i)$ to be independent and identically distributed (iid) according to an unknown joint probability measure $\boldsymbol{\mu}$ on $\Lambda := \mathcal{X} \times \{0,1\}$, and we denote this by $(\boldsymbol{x}_i, y_i) \overset{\text{iid}}{\sim} \boldsymbol{\mu}$. We call $\mu$ the marginal probability measure on $\mathcal{X}$ induced by $\boldsymbol{\mu}$, and assume that $\mu$ admits a density $f : \mathcal{X} \to [0, \infty)$ with respect to the Lebesgue measure $\lambda$, such that $f \in L^\infty(\lambda)$. Moreover, we fix a reference measure on $\Lambda$ of the form

$$\boldsymbol{\lambda} := \lambda \times \eta, \quad \text{where} \quad \eta \text{ is a probability measure on } \{0,1\} \text{ with } \eta(\{0\}), \eta(\{1\}) > 0. \tag{1}$$

With this convention, $\boldsymbol{\mu}$ admits a density with respect to $\boldsymbol{\lambda}$. Consequently, the distribution of the iid sample $\boldsymbol{S}_n$ on $\Lambda^n$ is $\boldsymbol{\mu}^{\otimes n}$ and admits a density with respect to the product reference measure $\boldsymbol{\lambda}^{\otimes n}$.

A binary classifier can be defined as an indicator function $h := \mathbb{1}_\Omega$, where $\Omega \subset \mathcal{X}$ is a decision region, such that in a noiseless setting—assumed throughout this paper—$h(\boldsymbol{x}_i) = y_i$, for all $i = 1, \ldots, n$.

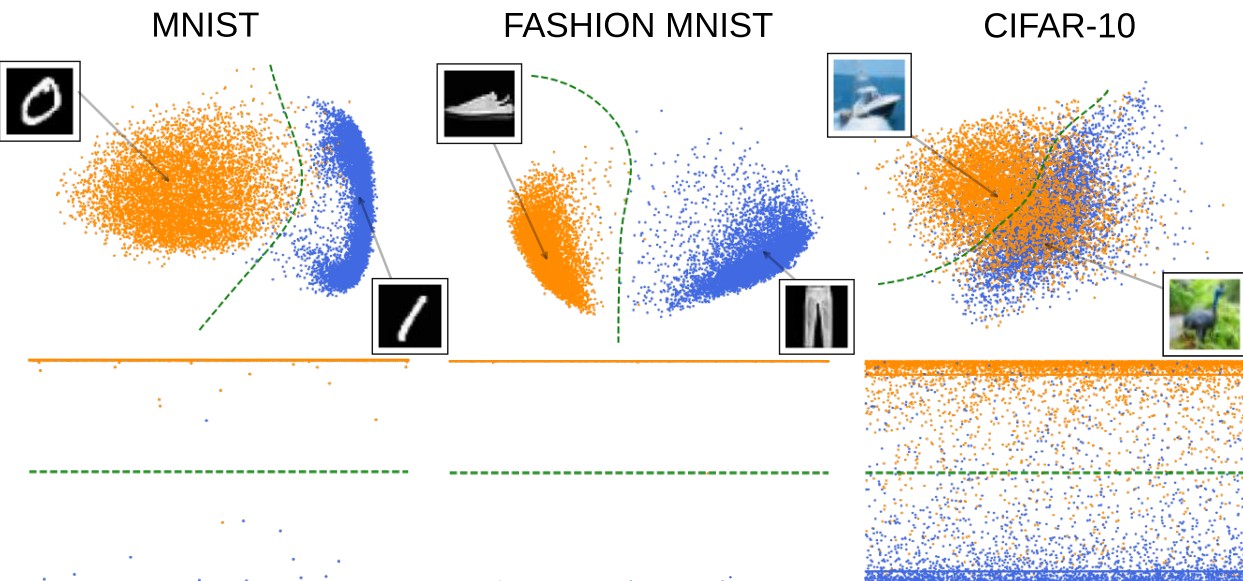

Figure 1: Geometric margin in common classification problems. The top row shows a two-dimensional embedding on the first two principal components and a decision boundary identified by a support vector machine: Clearly MNIST Lecun et al. (1998) and Fashion MNIST Xiao et al. (2017) exhibit a strong margin between some classes, and for the CIFAR-10 data Krizhevsky (2009) the margin is not immediately visible since the projection onto the first two principal components can hide structure in the remaining directions. However, in the second row we show the class probabilities predicted by a support vector classifier, which again shows an extremely strong margin for MNIST and Fashion MNIST, but also reveals that the CIFAR-10 data exhibits a form of margin, albeit a weaker one: Here, the density of data points reduces substantially closer to the decision boundary, even though the classifier does not produce a perfect separation. Which lower bounds on learning can be found in the presence of such various types of margins will be demonstrated in our main results (Section 2).

**Remark 1.**

- *Although many real-world datasets contain some level of label noise, there are also important situations in which the labels are generated by a deterministic rule or by highly reliable annotations. Examples include classification tasks arising from physical simulations, quality-control problems, and certain hand-labeled datasets with clear class definitions. Moreover, the noiseless setting serves as a natural benchmark for understanding the intrinsic statistical complexity of classification problems.*

- *The noiseless setting corresponds to the special case of zero label noise. Hence, any distribution class allowing label noise contains the noiseless class as a subclass, and any minimax lower bound proved here automatically remains valid for such larger classes.*

- *The noiseless assumption is important if we want to resolve the precise role of the regularity of the decision boundary and the margin conditions. Indeed, even the presence of low noise could yield vastly different lower bounds, because the learning problem then requires resolving the noise and this complication can mask the role of the decision boundary and the margin. An extended discussion of this is given in (Petersen & Voigtlaender, 2021, Section 1.1, Point 1). For a quick argument, we highlight, e.g., Stone (1982), where it was obtained that the optimal learning rate, to learn a function $f \in C^k([0,1]^d)$ with $\|f\|_{C^k} \le 1$ and noise defined as a parameter $\varepsilon \overset{iid}{\sim} N(0, \sigma^2)$ for $\sigma > 0$, is of the order of $O(n^{-k/(2k+d)})$, and decays slower than $n^{-1/2}$. On the other hand, in Krieg & Sonnleitner (2023), where the same problem is considered without noise, learning rates of the order of $\mathcal{O}(n^{-k/d})$ were obtained, in some cases faster than $n^{-1}$.*

An estimator $\hat{h}_n$ is a classifier constructed from the observed sample $\boldsymbol{S}_n$, intended to approximate the true classifier $h$ over $\mathcal{X}$. Such estimators are produced by learning algorithms, which are measurable mappings

$$A \in \mathcal{A}_n(\mathcal{G}) := \{A : \Lambda^n \to \mathcal{G} \mid A \text{ is measurable}\}, \quad \text{where} \quad \mathcal{G} \subseteq L^2(\lambda). \tag{2}$$

Here, the algorithm $A$ specifies the procedure that maps the observed sample to the estimator.

## 1.2 Conditions

We now specify the conditions on the classifiers and distributions that allow us to formulate our minimax problem.

C1) The decision regions can be described by horizon functions (see Petersen & Voigtlaender (2021)):

We consider a class of continuous functions $\mathscr{C} \subset C([0,1]^{d-1}; [0,1])$ and define, for each $b \in \mathscr{C}$, the associated horizon function as

$$h_b : \mathcal{X} \to \{0,1\},$$
$$\boldsymbol{x} = (x_1, \ldots, x_d) \mapsto \mathbb{1}_{b(\boldsymbol{x}^{(d)}) \leq x_d}, \tag{3}$$

where $\boldsymbol{x}^{(d)} := (x_1, \ldots, x_{d-1})$. We denote by $H_{\mathscr{C}} := \{h_b \mid b \in \mathscr{C}\}$ the set of horizon functions associated to $\mathscr{C}$. We assume that

$$\Omega := \Omega_h = \{\boldsymbol{x} \in \mathcal{X} \mid h(\boldsymbol{x}) = 1\} \quad \text{where} \quad h \in H_{\mathscr{C}}. \tag{4}$$

**Remark 2.**

*(a) For each $h \in H_{\mathscr{C}}$, there exists a unique $b \in \mathscr{C}$ such that $h = h_b$, and conversely, each $b \in \mathscr{C}$ is uniquely determined by its associated $h_b$. This follows from the bijectivity of the map $b \mapsto h_b$.*

*(b) Each horizon function $h \in H_{\mathscr{C}}$, associated with a function $b \in \mathscr{C}$, defines a decision region $\Omega_h$ with its corresponding boundary*

$$\partial\Omega_h = \left\{\boldsymbol{x} \in \mathcal{X} \mid b(\boldsymbol{x}^{(d)}) = x_d\right\}.$$

C2) Regular boundary (Hölder continuity): Let $\alpha \in (0,1]$ be fixed. For all $b \in \mathscr{C}$, there exists a constant $K_b > 0$, such that

$$|b(\boldsymbol{z}) - b(\boldsymbol{w})| \leq K_b \|\boldsymbol{z} - \boldsymbol{w}\|_2^{\alpha} \quad \text{for all} \quad \boldsymbol{z}, \boldsymbol{w} \in [0,1]^{d-1}. \tag{5}$$

C3) Geometric margin near the decision boundary: Let $(h, \mu)$ be a pair consisting of a classifier $h \in H_{\mathscr{C}}$ and the marginal distribution $\mu$ of the input vectors $\{\boldsymbol{x}_i\}_{i=1}^n$ in the sample $\boldsymbol{S}_n$ introduced in Section 1.1. We say that $(h, \mu)$ satisfies the margin condition with exponent $\gamma > 0$ if there exists $C > 0$ such that, for all $\varepsilon > 0$,

$$\mu(B_\varepsilon^h) \leq C\varepsilon^\gamma \quad \text{where} \quad B_\varepsilon^h := \{\boldsymbol{x} \in \mathcal{X} \mid \operatorname{dist}(\boldsymbol{x}, \partial\Omega_h) \leq \varepsilon\} \tag{6}$$

is the neighborhood of radius $\varepsilon > 0$ around $\partial\Omega_h$ with respect to the Euclidean distance

$$\operatorname{dist}(\boldsymbol{x}, \partial\Omega_h) := \inf_{\boldsymbol{y} \in \partial\Omega_h} \|\boldsymbol{x} - \boldsymbol{y}\|_2.$$

We call $\gamma$ the margin exponent (see Christmann & Steinwart (2008); Kim et al. (2021)).

**Remark 3.** *The measure $\mu$ associated with $h$ is in general not unique: different data-generating distributions may share the same decision boundary $h$, and conversely a given marginal distribution $\mu$ may be compatible with multiple admissible boundaries in $H_{\mathscr{C}}$.*

Let $\mathcal{L}_\lambda$ be the class of all probability measures on $\mathcal{X}$ that admit an essentially bounded density with respect to the Lebesgue measure $\lambda$, i.e.,

$$\mathcal{L}_\lambda := \left\{ \mu \mid \text{ there exists } f : \mathcal{X} \to [0, \infty) \text{ such that } \int_{\mathcal{X}} f d\lambda = 1, \ d\mu = f d\lambda \text{ and } f \in L^\infty(\lambda) \right\}. \quad (7)$$

We denote by $\mathcal{P}_{\mathscr{C}}(\mathcal{M})$ the set of all pairs $(h, \mu)$ that satisfy the margin condition, with $h \in H_{\mathscr{C}}$ and $\mu \in \mathcal{M} \subseteq \mathcal{L}_\lambda$. That is,

$$\mathcal{P}_{\mathscr{C}}(\mathcal{M}) := \{(h, \mu) \in H_{\mathscr{C}} \times \mathcal{M} \mid (h, \mu) \text{ satisfies } \textit{equation 6}\} \quad \text{where} \quad \mathcal{M} \subseteq \mathcal{L}_\lambda. \quad (8)$$

**Remark 4.** *In particular, the margin condition is satisfied by $(h, \mu)$ if the density $f$ of $\mu$ with respect to $\lambda$ satisfies*

$$f(\boldsymbol{x}) \lesssim \begin{cases} \min\left\{\frac{\varepsilon^\gamma}{\lambda(B_\varepsilon^h)}, 1\right\} & \text{if } \boldsymbol{x} \in B_\varepsilon^h \\ 1 & \text{otherwise,} \end{cases}$$

*for almost every $\boldsymbol{x} \in \mathcal{X}$ and for all $\varepsilon > 0$. It is also satisfied when*

$$f(\boldsymbol{x}) \lesssim \text{dist}^\gamma(\boldsymbol{x}, \partial\Omega_h) \qquad \text{for almost every} \qquad \boldsymbol{x} \in \mathcal{X}.$$

Under the above conditions, we establish lower bounds for the minimax error associated with estimating binary classifiers whose decision regions satisfy C1) on a function class $\mathscr{C}$ with regularity C2), and where the classifier–distribution pair satisfies the margin condition C3); i.e., we lower bound the following inf-sup expression

$$\mathcal{I}_n(\mathscr{C}) := \inf_{A \in \mathcal{A}_n(L^2(\lambda))} \sup_{(h, \mu) \in \mathcal{P}_{\mathscr{C}}(\mathcal{L}_\lambda)} \mathbb{E}_{\{\boldsymbol{x}_i\}_{i=1}^n \overset{iid}{\sim} \mu} \|A(\boldsymbol{S}_n) - h\|_{L^2(\mu)}^2, \quad (9)$$

where $\boldsymbol{S}_n := ((\boldsymbol{x}_i, y_i))_{i=1}^n$ denotes the sample, $\mathcal{L}_\lambda$ is defined in equation 7 and $\mathcal{P}_{\mathscr{C}}$ is as in equation 8.

**Remark 5.**

- *Throughout this work, we restrict to marginals $\mu$ with density $f = d\mu/d\lambda \in L^\infty(\lambda)$. Hence, for every $A(\boldsymbol{S}_n) \in L^2(\lambda)$, the expression $\|A(\boldsymbol{S}_n) - h\|_{L^2(\mu)}^2$ is well-defined.*

- *For simplicity, $\{\boldsymbol{x}_i\}_{i=1}^n \overset{iid}{\sim} \mu$ means that $\boldsymbol{x}_i \overset{iid}{\sim} \mu$ for all $i \in \{1, \ldots, n\}$. In equation 9, we use the notation*

$$\mathbb{E}_{\{\boldsymbol{x}_i\}_{i=1}^n \overset{iid}{\sim} \mu} := \mathbb{E}_{\boldsymbol{S}_n \sim \boldsymbol{\mu}^{\otimes n}}, \quad (10)$$

*where the marginal of $\boldsymbol{\mu}$ on $\mathcal{X}$ is $\mu$, and $(\boldsymbol{x}_1, \ldots, \boldsymbol{x}_n) \sim \mu^{\otimes n}$. We adopt this notation since the marginal distribution on $\mathcal{X}$ is mainly relevant in our analysis.*

### 1.3 Function classes

The classes of functions considered in this work are the following.

- **Hölder continuous functions.** Let $\mathcal{H}_\alpha$ denote the class of functions satisfying the Hölder continuity condition in equation 5.

- **Barron regular functions.** Various definitions of Barron function classes can be found in the literature, differing slightly in their formulation. Essentially, these are functions with a bounded first-order Fourier moment, which we make explicit below (see Barron (1993); Caragea et al. (2023); Petersen & Voigtlaender (2021)).

  A function $f : [0, 1]^{d-1} \to \mathbb{R}$ is said to be of Barron class with constant $C > 0$, if there are $c \in [-C, C]$ and a measurable function $F : \mathbb{R}^{d-1} \to \mathbb{C}$ satisfying

$$f(\boldsymbol{z}) = c + \int_{\mathbb{R}^{d-1}} (e^{i\boldsymbol{z}\cdot\boldsymbol{\xi}} - 1)F(\boldsymbol{\xi})d\boldsymbol{\xi} \quad \text{and} \quad \int_{\mathbb{R}^{d-1}} \|\boldsymbol{\xi}\|_1 |F(\boldsymbol{\xi})|d\boldsymbol{\xi} \leq C \quad (11)$$

  for all $\boldsymbol{z} \in [0, 1]^{d-1}$, where $\boldsymbol{\xi} := (\xi_1, \ldots, \xi_{d-1})$ and $\|\boldsymbol{\xi}\|_1 := \sum_{j=1}^{d-1} |\xi_j|$. The set of all Barron functions with constant $C$ is known as the Barron space and is denoted by $\mathcal{B}_C$.

**Remark 6.** *Every Barron function $f \in \mathcal{B}_C$ is Lipschitz on $[0,1]^{d-1}$, since*

$$
\begin{aligned}
|f(\boldsymbol{z}) - f(\boldsymbol{w})| &= \left| \int_{\mathbb{R}^{d-1}} \left( e^{i\boldsymbol{z}\cdot\boldsymbol{\xi}} - e^{i\boldsymbol{w}\cdot\boldsymbol{\xi}} \right) F(\boldsymbol{\xi}) d\boldsymbol{\xi} \right| \\
&\leq \int_{\mathbb{R}^{d-1}} \left| e^{i\boldsymbol{z}\cdot\boldsymbol{\xi}} - e^{i\boldsymbol{w}\cdot\boldsymbol{\xi}} \right| |F(\boldsymbol{\xi})| d\boldsymbol{\xi} \\
&\leq L_0 \int_{\mathbb{R}^{d-1}} |(\boldsymbol{z} - \boldsymbol{w}) \cdot \boldsymbol{\xi}| \, |F(\boldsymbol{\xi})| d\boldsymbol{\xi} \\
&\leq L \, \|\boldsymbol{z} - \boldsymbol{w}\|_2 \quad \text{for some constants} \quad L_0, L > 0,
\end{aligned}
$$

*and all $\boldsymbol{z}, \boldsymbol{w} \in [0,1]^{d-1}$, i.e., $f$ satisfies the equation 5 with exponent $\alpha = 1$.*

- **Convex-Lipschitz functions.** We denote by $\mathcal{C} := \mathcal{C}([0,1]^{d-1}; [0,1])$ the class of all convex functions on $[0,1]^{d-1}$ taking values in $[0,1]$, which are uniformly Lipschitz as in equation 5 with $\alpha = 1$ (see Guntuboyina & Sen (2013)).

## 1.4 Previous works and our contribution

Some related work on binary classifiers under the margin condition C3) can be found in: (Christmann & Steinwart, 2008, Section 8), for support vector machines where learning rates were sometimes as fast as $n^{-1}$, being $n$ the number of data points; Kim et al. (2021), based on neural networks with hinge loss that achieve fast convergence rates when $d \lesssim \gamma$, particularly as fast as $n^{-(q+1)/(q+2)}$ when the margin exponent $\gamma \to \infty$, where $q$ is a noise parameter; and García & Petersen (2025), where it is found that using ReLU neural networks, the strong margin conditions imply fast learning bounds that are close to $n^{-1}(1 + \log n)$. Furthermore, in García & Petersen (2025); Kim et al. (2021), a regularity condition is assumed on the decision boundary, where $\partial\Omega$ can be described by classes of functions $\mathscr{C}$ satisfying condition C1) on the elements of a covering for the set $\Omega$. In Kim et al. (2021), the class of Hölder continuous functions is considered, and in García & Petersen (2025), the Barron class. The last two works mentioned above only provide upper bounds for the learning rate when binary classifiers are approximated by neural networks under the margin condition. Our contribution now, by finding minimax lower bounds for learning rates on binary estimators, is to confirm that these learning rates are indeed optimal. It is important to highlight that the margin plays a main role in the fast learning rates obtained in each function space, when $\gamma$ is sufficiently large, the curse of dimensionality is overcome.

In Yang & Barron (1999), it was shown that through an entropy notion on density spaces, it is possible to determine lower bounds for the learning rate of binary estimators. Then, in Petersen & Voigtlaender (2021), a relation between distances in a specific set of densities and the norms $L^2(\lambda)$ in $H_{\mathscr{C}}$ and $L^1([0,1]^{d-1})$ in $\mathscr{C}$ was demonstrated, thus adapting the main results of Yang & Barron (1999) to function spaces $\mathscr{C}$ as in C1). However, the margin condition was not assumed in either Petersen & Voigtlaender (2021) or Yang & Barron (1999).

In this paper, we establish a lower bound for the minimax expression $\mathcal{I}_n(\mathscr{C})$ in equation 9 through Construction 10, which yields a finite subfamily $\mathscr{C}_\Theta \subset \mathscr{C}$ indexed by $\Theta = \{0,1\}^m$, the set of binary vectors of length $m$. More precisely, given an arbitrary algorithm $A \in \mathcal{A}_n(L^2(\lambda))$, we first show that the supremum in equation 9 can be lower bounded by its restriction to $\mathscr{C}_\Theta$ and to a carefully constructed family of densities $\mathcal{F}_\Theta$, chosen so as to satisfy Condition C3) together with some additional properties. Next, by introducing a suitable projection of the estimator $A(\boldsymbol{S}_n)$ onto the discrete set $\Theta$, we further reduce the problem to lower bounding an expression involving the Hamming distance between the projected estimator and an arbitrary element of $\Theta$. Then, after this reduction, we apply Assouad's lemma (Tsybakov, 2009, Theorem 2.12) to obtain Theorem 11.

Finally, we apply Theorem 11 to the three classes of functions introduced in Section 1.3, namely the class of Hölder continuous functions, the Barron class of functions, and the class of convex-Lipschitz functions, thereby obtaining Corollary 13. In particular, we get the following.

- For the Barron class, $\mathscr{C} = \mathcal{B}_C$, the lower bound in equation 22 is given by

$$\mathcal{I}_n(\mathcal{B}_C) \gtrsim n^{-\frac{\gamma}{\gamma + \left(\frac{2(d-1)}{d+1}\right)}} \quad \text{for all} \quad \gamma \geq 1 \quad \text{and} \quad n \in \mathbb{N},$$

where $2(d-1)/(d+1) \to 2$ when $d \to \infty$. On the other hand, in García & Petersen (2025), upper bounds for the learning rate were established in the form

$$\mathcal{I}_n(\mathcal{B}_C) \lesssim n^{-\frac{\gamma}{\gamma+2}}(1 + \log n) \quad \text{for all} \quad \gamma > 0 \quad \text{and} \quad n \in \mathbb{N}.$$

Therefore, our lower bound matches the upper bound in García & Petersen (2025) up to logarithmic factors in the high-dimensional regime for $\gamma \geq 1$. Indeed, as $d \to \infty$, the exponent of $n$ in both bounds converges to $-\gamma/(\gamma + 2)$, showing that the learning rate obtained in García & Petersen (2025) is asymptotically optimal as the dimension tends to infinity.

**Remark 7.**

– *One of the most important aspects of the Barron case is that the exponent of $n$ does not deteriorate as $d \to \infty$. Indeed, the exponent in the lower bound converges to $-\gamma/(\gamma+2)$, which is the exponent in the known upper bound up to logarithmic factors. Thus, in the high-dimensional regime, the Barron class exhibits a dimension-independent learning rate and therefore overcomes the curse of dimensionality.*

– *Moreover, if after passing to the high-dimensional regime ($d \to \infty$) one lets the margin exponent $\gamma$ tend to infinity, the limit of the exponent $-\gamma/(\gamma+2)$ tends to $-1$. Thus, strong margins lead to rates close to the fast rate $n^{-1}$, improving over the exponent $-1/3$ obtained in the Barron setting without margin improvement in Petersen & Voigtlaender (2021).*

– *The gap between the lower and upper exponents in low dimensions should be interpreted with some caution. Our current lower-bound construction identifies the correct exponent in the high-dimensional regime, but it does not decide whether, for fixed small $d$, the lower bound can be improved or the available upper bound is not sharp. Accordingly, we only claim asymptotic optimality in the high-dimensional regime, where the two exponents coincide in the limit. This interpretation is consistent with previous work on Barron-regular classifiers; see, for instance, Petersen & Voigtlaender (2021), where the lower and upper exponents also differ for fixed $d$ but become arbitrarily close as $d \to \infty$.*

- For the Hölder continuous class, $\mathscr{C} = \mathcal{H}_\alpha$, we obtain in equation 21, that

$$\mathcal{I}_n(\mathcal{H}_\alpha) \gtrsim n^{-\frac{\gamma}{\gamma+(d-1)}} \quad \text{for all} \quad \alpha \in (0,1], \quad \gamma \geq \alpha \quad \text{and} \quad n \in \mathbb{N}.$$

In contrast, under the noiseless assumption (see Remark 1), (Kim et al., 2021, Theorem 3.4) provides the upper bound

$$\mathcal{I}_n(\mathcal{H}_\alpha) \lesssim \left(\frac{\log^3 n}{n}\right)^{\frac{\alpha\gamma}{\alpha\gamma+(d-1)}} \quad \text{for all} \quad \alpha \in (0,1], \quad \gamma \geq 1 \quad \text{and} \quad n \in \mathbb{N}.$$

Comparing these two bounds, we see that for values of $\alpha$ close to 1, the exponents of $n$ in the lower and upper bounds become similar, indicating that the corresponding learning rates are nearly the same, up to logarithmic factors. Moreover, for Lipschitz functions ($\alpha = 1$), the learning rate is asymptotically optimal. However, for small values of $\alpha$, the lower and upper bounds differ substantially.

**Remark 8.** *The proofs of our main results are based on the choice we made of the family of densities $\mathcal{F}_\Theta$ defined in equation 26. These densities depend directly on the vertical distance of a point in the space $\mathcal{X}$ to the decision boundary. However, the geometric margin condition C3) (in its classical Tsybakov form, see Kim et al. (2021)) is defined using the Euclidean distance. In order to relate the vertical distance with the Euclidean one and make our densities satisfy C3), we prove in item 1) of Lemma 16 that*

$$|x_d - b(\boldsymbol{x}^{(d)})|^{\frac{1}{\alpha}} \lesssim \text{dist}\left(\boldsymbol{x}, \partial\Omega_{h_b}\right) \leq |x_d - b(\boldsymbol{x}^{(d)})| \quad \text{for all} \quad b \in \mathscr{C} \quad \text{and} \quad \boldsymbol{x} \in \mathcal{X}.$$

*From the previous inequality and the proof of item 6) of Lemma 16, it follows that*

$$\mu_{\boldsymbol{\theta}}\left(\left\{\boldsymbol{x} \in \mathcal{X} \mid |x_d - b_{\boldsymbol{\theta}}(\boldsymbol{x}^{(d)})| \leq \varepsilon\right\}\right) \leq \mu_{\boldsymbol{\theta}}\left(\{\boldsymbol{x} \in \mathcal{X} \mid \mathrm{dist}\,(\boldsymbol{x}, \partial\Omega_{h_{\boldsymbol{\theta}}}) \leq \varepsilon\}\right)$$

$$\lesssim \mu_{\boldsymbol{\theta}}\left(\left\{\boldsymbol{x} \in \mathcal{X} \mid |x_d - b_{\boldsymbol{\theta}}(\boldsymbol{x}^{(d)})| \leq \varepsilon^{\alpha}\right\}\right) \lesssim \varepsilon^{\widetilde{\gamma}\alpha}.$$

*Hence, our densities satisfy the geometric margin condition C3) with exponent $\gamma := \widetilde{\gamma}\alpha$. In the case $\alpha = 1$, the vertical and Euclidean distances are equivalent, whereas for $\alpha < 1$ they are not (the vertical distance can become dramatically larger than the Euclidean distance for points lying opposite peaks or valleys, where the graph of the decision boundary is highly non-Lipschitz). This explains the gap between the lower and upper bounds for the learning rate in the case $\alpha < 1$.*

*Nevertheless, we believe that our arguments can be extended to densities that depend directly on the Euclidean distance rather than on the vertical distance, although the proofs would become more difficult, since horizon classifiers are naturally described in terms of vertical distances.*

- For the convex-Lipschitz class, $\mathscr{C} = \mathcal{C}$, for which in equation 23, we lower bound by

$$\mathcal{I}_n(\mathcal{C}) \gtrsim n^{-\frac{\gamma}{\gamma+(d-1)/2}} \quad \text{for all} \quad \gamma \geq 1 \quad \text{and} \quad n \in \mathbb{N}.$$

Compared with the lower bound for general Lipschitz functions in the previous item (when $\alpha = 1$), the exponent for $n$ in the present bound is $-\gamma/(\gamma + (d-1)/2)$ rather than $-\gamma/(\gamma + (d-1))$. This suggests that the additional convexity assumption could lead to a faster optimal learning rate.

## 2 Main results

We begin by presenting a general result that provides a lower bound for the minimax expression in equation 9. Here we make the assumption that a finite subfamily of the function class $\mathscr{C}$ can be constructed by adding independent, localized perturbations of small amplitude to a fixed baseline function over a partition of the domain. We first describe the construction, and then state our main theorem with the corresponding lower bound.

**Remark 9.** *Before presenting the construction in full generality, we briefly explain its main idea in the case $d = 2$, illustrated in Figures 2, 3 and 4. This intuition will help clarify the role of the partition, the localized perturbations, and the binary parameter vectors introduced below.*

*To start, the domain of a baseline boundary in $\mathscr{C}$ is divided into small cells, represented by the gray grid in the background of Figures 2 and 3. On each cell, we make one binary choice: either we add a localized perturbation to the baseline boundary, or we leave it unchanged. These perturbations are the small bumps in the Hölder and Barron cases and the local secant-type perturbations in the convex-Lipschitz case; the baseline boundary is shown by the dashed green line in Figure 2 and is the constant function $1/2$ in Figures 3 and 4. Each such choice is encoded by one coordinate of a binary vector $\boldsymbol{\theta} \in \{0,1\}^m$. Repeating this independently over all cells produces a large finite family of nearby decision boundaries. This family of boundaries is then used to define a corresponding family of densities satisfying the margin condition. In Figure 3, the neighborhood $\mathcal{R}$ of the decision boundary represents the region where these densities are smaller. The next step is to project the estimation error onto suitable local regions inside $\mathcal{R}$, such as the green shaded set in Figure 4, where the densities can be lower bounded. This yields a lower bound for the minimax risk of equation 9 in terms of the Hamming distance between binary vectors. Finally, applying Assouad's lemma to this finite family of densities gives the desired minimax lower bounds.*

*However, to fix the height at which the perturbations are added and to specify their precise form in a way that is suitable for the proof of Theorem 11 and Corollary 13, the following construction introduces this finite family of decision boundaries under the required technical conditions.*

**Construction 10.** *Assume the existence of an even integer $M \geq 2$; a baseline function $b_0 \in \mathscr{C}$; a Hölder continuous function $\varphi : \mathbb{R}^{d-1} \to \mathbb{R}$ with exponent $\alpha \in (0,1]$ and constant $K_{\varphi} > 0$ with respect to the Euclidean*

*norm (as in equation 5). Further assume that $\varphi$ is Hölder continuous at $\mathbf{0}$ with exponent $\alpha$ and constant $C_\varphi > 0$ with respect to the $\ell_\infty$-norm, i.e.,*

$$|\varphi(\boldsymbol{z}) - \varphi(\mathbf{0})| \le C_\varphi \|\boldsymbol{z}\|_\infty^\alpha \quad \text{for all} \quad \boldsymbol{z} \in \mathbb{R}^{d-1}; \tag{12}$$

*and a constant $C_\mathscr{C} > 0$.*

- ***Partition of the domain.** Let $\Gamma_M := \{1, 2, \ldots, M\}^{d-1}$ be the set of all $(d-1)$-dimensional vectors with integer entries between 1 and $M$. Let $s := M/2$, $m := s^{d-1}$ and consider the set of vectors in $\Gamma_M$ with odd integer entries as $\Gamma := \{1, 3, \ldots, 2s-1\}^{d-1}$. Since $|\Gamma| = m$, we write $\Gamma = \{\boldsymbol{v}_1, \ldots, \boldsymbol{v}_m\}$, where $\boldsymbol{v}_1, \ldots, \boldsymbol{v}_m$ denote an arbitrary enumeration of the elements of $\Gamma$.*

  *We define $P := \{Q_1, \ldots, Q_m\}$ as a family of sets forming a partition of $[0, 1]^{d-1}$ up to boundaries[1], with*

  $$\begin{aligned} Q_j &:= \left\{ \boldsymbol{z} \in [0, 1]^{d-1} \mid \|\boldsymbol{z} - \boldsymbol{v}_j/M\|_\infty \le 1/M \right\} \\ &= \prod_{k=1}^{d-1} \left[ \frac{v_{jk}-1}{M}, \frac{v_{jk}+1}{M} \right] \quad \text{where} \quad \boldsymbol{v}_j = (v_{j1}, v_{j2}, \ldots, v_{j(d-1)}), \end{aligned} \tag{13}$$

  *for all $j = 1, \ldots, m$.*

- ***Localized perturbations.** Let $\varphi$ satisfy*

  $$\operatorname{supp} \varphi \subset (-1, 1)^{d-1}, \quad \varphi(\mathbf{0}) = 1, \quad \text{and} \quad 0 \le \varphi(\boldsymbol{z}) \le 1 \quad \text{for all} \quad \boldsymbol{z} \in \mathbb{R}^{d-1}. \tag{14}$$

  *We define the local perturbation function $\varphi_j : [0, 1]^{d-1} \to [0, 1]$ by*

  $$\varphi_j(\boldsymbol{z}) := \varphi\left( M(\boldsymbol{z} - \boldsymbol{v}_j/M) \right), \quad \text{for each} \quad j = 1, \ldots, m. \tag{15}$$

- ***Finite subfamily of $\mathscr{C}$.** Let $\Theta := \{0, 1\}^m$ denote the set of binary vectors of length $m$. For all $\boldsymbol{\theta} \in \Theta$, define*
  $$b_{\boldsymbol{\theta}} := b_0 + \boldsymbol{\theta} \cdot \boldsymbol{\varphi} \quad \text{with} \quad \boldsymbol{\varphi} := C_\mathscr{C}(\varphi_1, \ldots, \varphi_m), \tag{16}$$

  *and the family*
  $$\mathscr{C}_\Theta := \{b_{\boldsymbol{\theta}} \mid \boldsymbol{\theta} \in \Theta\}. \tag{17}$$

  *Assume that*

  $$C_\mathscr{C} \le M^{-\alpha}/4 \quad \text{and} \quad b_0(\boldsymbol{z}) \in [C_\mathscr{C}, 1 - 3C_\mathscr{C}] \quad \text{for all} \quad \boldsymbol{z} \in [0, 1]^{d-1}. \tag{18}$$

Note that the objects used to define $b_{\boldsymbol{\theta}}$ in equation 16 are required to satisfy equation 13, equation 14 and equation 15, and that the constant $C_\mathscr{C}$ in equation 16 will be chosen in order to ensure that $b_{\boldsymbol{\theta}}$ belongs to the class $\mathscr{C}$ for all $\boldsymbol{\theta} \in \Theta$; see Corollary 13 and Figure 2 for explicit choices. In addition, equation 18 is imposed for convenience in the proof of the following theorem.

**Theorem 11.** *Assume conditions C1), C2) with parameter $\alpha \in (0, 1]$, and C3) with margin exponent $\gamma \ge \alpha$. Let $n \in \mathbb{N}$ be arbitrary, $\widetilde{\gamma} := \gamma/\alpha$, and consider the minimax expression $\mathcal{I}_n(\mathscr{C})$ in equation 9. If Construction 10 holds,*

$$\mathscr{C}_\Theta \subseteq \mathscr{C} \quad \text{and} \quad 2^{\widetilde{\gamma}+6} n C_\mathscr{C}^{\widetilde{\gamma}} M^{-(d-1)} \le 1. \tag{19}$$

*Then,*

$$\mathcal{I}_n(\mathscr{C}) \ge \left( \frac{r^{d-1}}{8^{\widetilde{\gamma}+2}} \right) C_\mathscr{C}^{\widetilde{\gamma}}, \tag{20}$$

*where $r := \min\{1, (2C_\varphi)^{-1/\alpha}\}$, and $C_\varphi$ is as in equation 12.*

---

[1]Meaning that any intersection between two sets in this family occurs only on their boundaries.

**Remark 12.** *Although $C_{\mathscr{C}}$ and $M$ appear as abstract parameters in Theorem 11, the condition in equation 19 implicitly makes them depend on $n$. In applications of Theorem 11, the constant $C_{\mathscr{C}}$ is typically chosen depending on the parameter $M$ in Construction 10; see, for instance, the proof of Corollary 13 in Section C. Since $M$ is later chosen depending on $n$ in order to satisfy equation 19, the resulting constant $C_{\mathscr{C}}$ will in general depend on $n$ through $M$.*

To prove Theorem 11 in Section B, we use the subfamily $\mathscr{C}_\Theta \subset \mathscr{C}$, obtained in Construction 10, in order to simplify the problem of lower bounding the minimax expression in equation 9 to that of lower bounding a similar expression in equation 80 (for an arbitrary algorithm $A \in \mathcal{A}_n(L^2(\lambda))$) restricted to the subfamily $\mathscr{C}_\Theta$ and to a particular class of densities $\mathcal{F}_\Theta$ defined in equation 26, which satisfy the margin condition and the properties presented in Lemmas 16 and 17. Then, under the projection equation 77 of the estimators $A(\boldsymbol{S}_n)$ onto the finite set $\Theta$, we further lower bound equation 80, thereby obtaining the lower bound equation 81, which depends on the Hamming distance between the projection of the estimator and the elements of $\Theta$. Finally, we apply in equation 83 a classical result from minimax arguments, namely Assouad's Lemma 19, and arrive at the lower bound equation 20.

The previous theorem provides a general lower bound under Construction 10. We now show how this construction applies to the function classes introduced in Section 1.3, leading to the following corollary.

**Corollary 13.** *Under the assumptions of Theorem 11, the following lower bounds hold.*

*i) For the Hölder continuous class $\mathscr{C} := \mathcal{H}_\alpha$,*

$$\mathcal{I}_n(\mathcal{H}_\alpha) \geq 2^{-\left(\frac{5(d-1)}{\alpha} + 13\widetilde{\gamma} + 6\right)} n^{-\frac{\gamma}{\gamma+(d-1)}} \quad \text{for all} \quad \alpha \in (0,1], \quad \gamma \geq \alpha \quad \text{and} \quad n \in \mathbb{N}. \quad (21)$$

*ii) For the Barron class $\mathscr{C} := \mathcal{B}_C$,*

$$\mathcal{I}_n(\mathcal{B}_C) \geq c_{d,C,\gamma} \, n^{-\frac{\gamma}{\gamma+\left(\frac{2(d-1)}{d+1}\right)}} \quad \text{for all} \quad \gamma \geq 1 \quad \text{and} \quad n \in \mathbb{N}, \quad (22)$$

*where $c_{d,C,\gamma}$ is a constant depending only on $d$, $C$ and $\gamma$.*

*iii) For the convex-Lipschitz class $\mathscr{C} := \mathcal{C}$,*

$$\mathcal{I}_n(\mathcal{C}) \geq c_{d,\gamma} \, n^{-\frac{\gamma}{\gamma+(d-1)/2}} \quad \text{for all} \quad \gamma \geq 1 \quad \text{and} \quad n \in \mathbb{N}, \quad (23)$$

*where $c_{d,\gamma}$ is a constant that depends only on $d$ and $\gamma$.*

The bounds in equation 21, equation 22 and equation 23 show how the margin exponent $\gamma$ interacts with the complexity of the underlying boundary class. In all three cases, larger values of $\gamma$ lead to faster rates, and as $\gamma \to \infty$ the exponent of $n$ approaches $-1$, corresponding to the fast rate $n^{-1}$. The dependence on the dimension enters through the additional term in the denominator: it is $d-1$ for Hölder, $2(d-1)/(d+1)$ for Barron, and $(d-1)/2$ for convex-Lipschitz. Thus, the Barron class is the only one among these three cases for which this term remains bounded as $d \to \infty$. If $\gamma$ grows proportionally to $d$ or faster, then the dimensional contribution in the denominator is compensated by the margin exponent, and the bounds approach the fast-rate regime. We refer to Remarks 7 and 8 for more discussion of the Barron and Hölder cases. The convex-Lipschitz bound occupies an intermediate position between the Hölder and Barron cases: convexity reduces the dimensional dependence from $d-1$ to $(d-1)/2$, although the exponent still deteriorates as $d \to \infty$.

The proof of Corollary 13 in Section C consists of applying Theorem 11 to each of the function classes considered in Section 1.3. In every case, we verify that Construction 10 holds by choosing an appropriate baseline function $b_0$, a suitable perturbation $\varphi$, and a constant $C_{\mathscr{C}}$ such that the resulting family $\mathscr{C}_\Theta$ is contained in the class under consideration. We then choose the scale parameter $M$ so that the condition in equation 19 is satisfied. Once these hypotheses are checked, Theorem 11 gives the desired lower bound through equation 20, for the Hölder, Barron, and convex-Lipschitz classes.

Roughly speaking, the parameter $M$ determines how many independent perturbations can be placed along the decision boundary, while $C_{\mathscr{C}}$ determines their admissible height. These choices are closely related to the Kolmogorov entropy behavior of the corresponding boundary class: different entropy behavior leads to different admissible trade-offs between the number of perturbations and their size, and hence leads to the different exponents in Corollary 13. We refer to Remark 15 for the connection with the corresponding entropy lower-bound constructions.

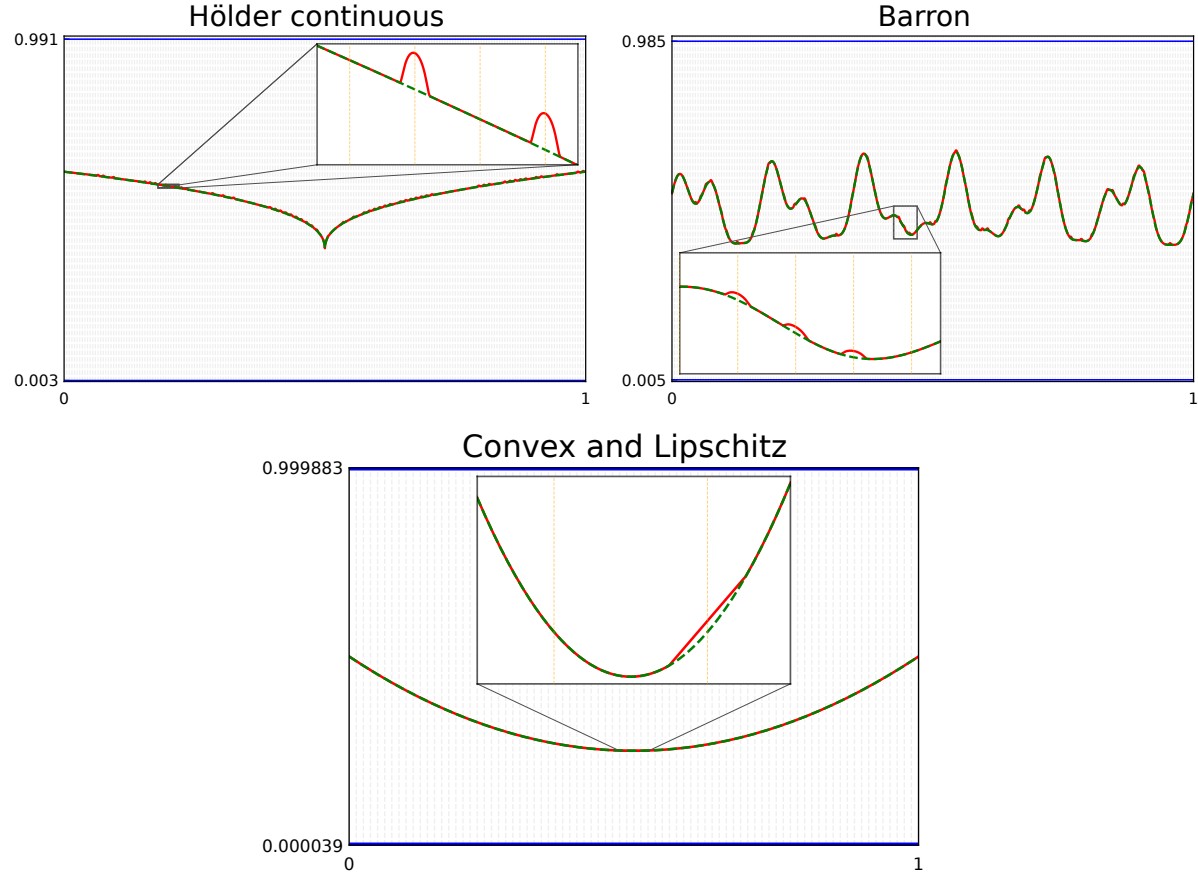

Figure 2: Examples of functions in $\mathscr{C}_{\Theta}$ for the three function classes $\mathscr{C}$ introduced in Section 1.3, with $d = 2$ and specific choices of $b_0$ and $\varphi$ in each case. The green curves represent the baseline function $b_0$, while the red curves show a function $b_{\boldsymbol{\theta}}$ obtained by adding localized perturbations as in Construction 10. The horizontal blue lines indicate the bounds $C_{\mathscr{C}}$ and $1 - 3C_{\mathscr{C}}$, as required in equation 18. The background grid in each plot corresponds to the partition associated with $M$ in equation 13. Each panel also includes a magnified view to better visualize the local effect of the perturbations on the baseline function. In particular, for the Hölder continuous case, the parameters are chosen according to Subsection C.1: $\varphi$ is defined by equation 84, with $\psi$ given by equation 97; the baseline function is $b_0(z) := 0.38 + 0.3|z - 0.5|^{\alpha}$ with $\alpha = 0.4$, which satisfies equation 18 for $C_{\mathscr{C}}$ as in equation 90. For the Barron class, the parameters are chosen following Subsection C.2: we take $b_0(z) = 0.5 + 0.15\,(0.6\sin(12\pi z) + 0.3\cos(22\pi z) + 0.2\sin(34\pi z))$, define $\varphi$ as in equation 99 with $\psi$ given by equation 97, and choose $C_{\mathscr{C}}$ as in equation 118. Lastly, for the Convex-Lipschitz case, the parameters are chosen according to Subsection C.3: $b_0(z) = 1/4 + |z - 1/2|^2$ as in equation 125, $\varphi(z) = \max\{1 - 4z^2, 0\}$ and $C_{\mathscr{C}}$ as in equation 128. For visualization purposes, we take $M = 200$ in the Hölder and Barron cases, and $M = 80$ in the Convex-Lipschitz case.

**Remark 14.** *The main results, Theorem 11 and Corollary 13, identify minimax lower bounds for the learning rates of binary classifiers from noiseless samples under a geometric margin condition. More precisely, Theorem 11 provides a general lower bound under Construction 10, while Corollary 13 applies this result to*

*the three function classes introduced in Section 1.3. There are, however, some limitations of these results that should be emphasized:*

1. *The established rates are understood in the minimax sense, where the supremum is taken over all pairs $(h, \mu)$ satisfying the margin condition. In practice, additional favorable assumptions on the data-generating distribution may hold, and the observed learning rates may therefore be faster.*

2. *As discussed in Remark 1, the analysis is restricted to the noiseless setting. This assumption is essential in order to clarify the role of the regularity of the decision boundary and the margin condition in the lower bounds, since in the presence of noise it may be unclear whether the difficulty comes from the noise or from the geometry of the classifier (see (Petersen & Voigtlaender, 2021, Section 1.1, Point 1)). Still, noisy data appears very often in applications. While the lower bounds derived here remain valid in the presence of label noise, a dedicated analysis of noisy classification problems is beyond the scope of this work.*

3. *The multiplicative constants in the lower bounds of Corollary 13 depend on the dimension $d$ and have an exponential dependence on $d$. This is explicit in the Hölder lower bound equation 21, and the corresponding prefactors in the Barron and convex-Lipschitz cases are derived in Sections C.2 and C.3. Hence, their behavior as $d \to \infty$ is not controlled by our results. Thus, the favorable high-dimensional behavior discussed above concerns the exponents of $n$ and does not yield dimension-free bounds for the full prefactors. This distinction between dimension-independent exponents and dimension-dependent prefactors also appears in Petersen & Voigtlaender (2021), where the high-dimensional interpretation concerns the exponent of the sample size, while the constants in the entropy and learning estimates are allowed to depend on $d$. A similar distinction appears in related fast-rate results; for instance, the bounds in Kim et al. (2021) are stated using implicit constants without a dimension-uniform control of their prefactors.*

**Remark 15.** *The proof of Theorem 11 is inspired by the use of Assouad's lemma in minimax lower bound arguments; see for instance (Tsybakov, 2009, Section 2.7.2) or (Mammen & Tsybakov, 1999, Proof of Theorem 3). For Corollary 13, the constructions in the particular function classes are motivated by the arguments used to derive lower bounds for Kolmogorov entropy in those spaces; see for instance Clements (1963) or Kolmogorov & Tihomirov (1993) for the Hölder case, Petersen & Voigtlaender (2021) for the Barron case, and Guntuboyina & Sen (2013) for the convex-Lipschitz case.*

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

## A  Auxiliary results

This section collects technical results that are used in the proof of the main results.

Under the assumption of Construction 10 and using the class $\mathscr{C}_\Theta$ in equation 17, we define an associated family of densities as follows. For each $\boldsymbol{\theta} \in \Theta$ and $b_{\boldsymbol{\theta}} \in \mathscr{C}_\Theta$, let the tube $\mathcal{T}_{\boldsymbol{\theta}}$ around the boundary $\partial\Omega_{\boldsymbol{\theta}}$ be given by

$$\mathcal{T}_{\boldsymbol{\theta}} := \left\{ \boldsymbol{x} \in \mathcal{X} \mid |x_d - b_{\boldsymbol{\theta}}(\boldsymbol{x}^{(d)})| \leq C_{\mathscr{C}} + \boldsymbol{\theta} \cdot \boldsymbol{\varphi}(\boldsymbol{x}^{(d)}) \right\}. \tag{24}$$

Define the region

$$\mathcal{R} := \left\{ \boldsymbol{x} \in \mathcal{X} \mid x_d \in [b_0(\boldsymbol{x}^{(d)}) - C_{\mathscr{C}}, b_0(\boldsymbol{x}^{(d)}) + 3C_{\mathscr{C}}] \right\} \tag{25}$$

and the density function $f_{\boldsymbol{\theta}} : \mathcal{X} \to [0, \infty)$ with respect to Lebesgue, as[2]

$$f_{\boldsymbol{\theta}}(\boldsymbol{x}) := \frac{1}{2}|x_d - b_{\boldsymbol{\theta}}(\boldsymbol{x}^{(d)})|^{\widetilde{\gamma}-1}\mathbb{1}_{\mathcal{R}\cap\mathcal{T}_{\boldsymbol{\theta}}}(\boldsymbol{x}) + \frac{1}{2}(C_{\mathscr{C}} - \boldsymbol{\theta} \cdot \boldsymbol{\varphi}(\boldsymbol{x}^{(d)}))^{\widetilde{\gamma}-1}\mathbb{1}_{\mathcal{R}\setminus\mathcal{T}_{\boldsymbol{\theta}}}(\boldsymbol{x}) + C_{\boldsymbol{\theta}}\mathbb{1}_{\mathcal{X}\setminus\mathcal{R}}(\boldsymbol{x}), \tag{26}$$

where $\widetilde{\gamma} \geq 1$ is a suitable constant, $\boldsymbol{\varphi}$ and $C_{\mathscr{C}}$ are as in equation 16, and $C_{\boldsymbol{\theta}}$ is a normalizing constant depending only on $\boldsymbol{\theta}$, i.e., such that $\int_{\mathcal{X}} f_{\boldsymbol{\theta}} d\lambda = 1$. See Figure 3 for an example of the regions defining $f_{\boldsymbol{\theta}}$ in a particular case. We denote the set of all these densities as $\mathcal{F}_\Theta = \{f_{\boldsymbol{\theta}} \mid \boldsymbol{\theta} \in \Theta\}$. Each density $f_{\boldsymbol{\theta}}$ induces a

---

[2]In the case $\widetilde{\gamma} = 1$, we use the convention $0^0 := 1$.

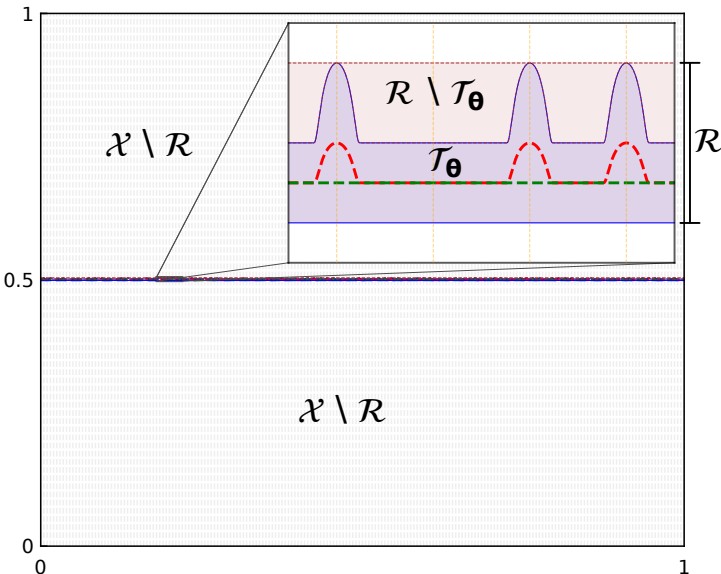

Figure 3: Illustration of the regions used to define $f_{\boldsymbol{\theta}}$ in the case $b_0 := 1/2$. The figure shows the sets $\mathcal{X} \setminus \mathcal{R}$, $\mathcal{T}_{\boldsymbol{\theta}}$, $\mathcal{R} \setminus \mathcal{T}_{\boldsymbol{\theta}}$, and $\mathcal{R}$. The background grid corresponds to the partition associated with $M$ in equation 13, and the inset provides a magnified view of the localized perturbations. For visualization purposes, we take $M = 200$ and choose $\varphi$ as a bump function, although any function satisfying the conditions of Construction 10 could be used.

probability measure $\mu_{\boldsymbol{\theta}}$, and we denote $\mathcal{M}_{\Theta} = \{\mu_{\boldsymbol{\theta}} \mid \boldsymbol{\theta} \in \Theta\}$ as the collection of all such measures. We write $h_{\boldsymbol{\theta}} = h_{b_{\boldsymbol{\theta}}}$ for all $h_{b_{\boldsymbol{\theta}}} \in H_{\mathscr{C}_{\Theta}}$.

We introduce the Hamming distance on $\Theta$, defined as

$$\rho_{Ham}(\boldsymbol{\theta}, \boldsymbol{\theta}') = \#J(\boldsymbol{\theta}, \boldsymbol{\theta}') \quad \text{where} \quad J(\boldsymbol{\theta}, \boldsymbol{\theta}') := \left\{ j \in \{1, \ldots, m\} \mid \theta_j \neq \theta'_j \right\}. \tag{27}$$

In the following lemma, we establish additional properties of the families $\mathscr{C}_{\Theta}$, $\mathcal{F}_{\Theta}$ and $\mathcal{M}_{\Theta}$ that are needed in the proof of the lower bound equation 20. In particular, they provide control of the geometry of the perturbed decision boundaries, ensure that the constructed classifier-distribution pairs satisfy the margin condition in C3), and yield the estimates required later for the application of Assouad's Lemma 19.

**Lemma 16.** *Let $\mathscr{C}$ satisfy C2) with $\alpha \in (0, 1]$. The following statements hold.*

*1) For all $b \in \mathscr{C}$ and all $\boldsymbol{x} \in \mathcal{X}$,*

$$\left( \frac{1}{\widetilde{K_b}} |x_d - b(\boldsymbol{x}^{(d)})| \right)^{\frac{1}{\alpha}} \leq \operatorname{dist}\left(\boldsymbol{x}, \partial\Omega_{h_b}\right) \leq |x_d - b(\boldsymbol{x}^{(d)})|, \tag{28}$$

*where*

$$1 < \widetilde{K_b} := \widetilde{K_b}(\alpha) = \begin{cases} \sqrt{1 + K_b^2} & \text{if } \alpha = 1 \\ \max\{2^{\alpha}, 2K_b\} & \text{if } \alpha < 1, \end{cases} \tag{29}$$

*and $K_b > 0$ is a constant satisfying equation 5.*

*2) For all $j, k \in \{1, \ldots, m\}$ with $j \neq k$, it holds (see equation 15) that*

$$\operatorname{supp} \varphi_j \cap \operatorname{supp} \varphi_k = \emptyset. \tag{30}$$

*Furthermore, for each $\boldsymbol{z} \in [0, 1]^{d-1}$, there exists $j \in \{1, \ldots, m\}$ such that*

$$\boldsymbol{\varphi}(\boldsymbol{z}) = C_{\mathscr{C}} \varphi_j(\boldsymbol{z}) \boldsymbol{e}_j, \tag{31}$$

where $\boldsymbol{e}_j$ denotes the vector with $1$ in its $j$-th entry and $0$ in all others.

3) Let $K_{b_0} > 0$ be the constant satisfying equation 5 for $b_0 \in \mathscr{C}$, and let $K_\varphi$ be the Hölder continuity constant of $\varphi$. Then, equation 5 is satisfied for all $b \in \mathscr{C}_\Theta$ with constant

$$K_b = K_\Theta := K_{b_0} + K_\varphi/2. \tag{32}$$

That is, $\mathscr{C}_\Theta$ satisfies C2) uniformly with exponent $\alpha$ and constant $K_\Theta$.

4) For all $\boldsymbol{\theta} \in \Theta$, $\mathcal{T}_{\boldsymbol{\theta}} \subseteq \mathcal{R}$. Moreover,

$$\mathcal{R} \setminus \mathcal{T}_{\boldsymbol{\theta}} = \left\{ \boldsymbol{x} \in \mathcal{X} \mid b_0(\boldsymbol{x}^{(d)}) + C_{\mathscr{C}} + 2\boldsymbol{\theta} \cdot \boldsymbol{\varphi}(\boldsymbol{x}^{(d)}) < x_d \le b_0(\boldsymbol{x}^{(d)}) + 3C_{\mathscr{C}} \right\}. \tag{33}$$

5) For each $\boldsymbol{\theta} \in \Theta$,

$$C_{\boldsymbol{\theta}} = \frac{1 - \int_{\mathcal{R}} f_{\boldsymbol{\theta}}(\boldsymbol{x}) d\boldsymbol{x}}{1 - 4C_{\mathscr{C}}} \quad and \quad \frac{1 - 2C_{\mathscr{C}}}{1 - 4C_{\mathscr{C}}} \le C_{\boldsymbol{\theta}} \le \frac{1}{1 - 4C_{\mathscr{C}}}, \tag{34}$$

where $C_{\boldsymbol{\theta}}$ is defined in equation 26.

6) If $\widetilde{\gamma} := \gamma/\alpha$ is chosen in equation 26 with $\gamma \ge \alpha$, then every pair $(h_{\boldsymbol{\theta}}, \mu_{\boldsymbol{\theta}}) \in H_{\mathscr{C}_\Theta} \times \mathcal{M}_\Theta$ satisfies the margin condition C3) with margin exponent $\gamma$.

7) For all $\boldsymbol{\theta}, \boldsymbol{\theta}' \in \Theta$ with $\rho_{Ham}(\boldsymbol{\theta}, \boldsymbol{\theta}') = 1$, we have that

$$\int_{\mathcal{X}} |f_{\boldsymbol{\theta}} - f_{\boldsymbol{\theta}'}| \, d\lambda \le 2^{\widetilde{\gamma}+3} C_{\mathscr{C}}^{\widetilde{\gamma}} M^{-(d-1)} \tag{35}$$

and

$$\int_{\{\boldsymbol{x} \in \mathcal{X} \mid h_{\boldsymbol{\theta}}(\boldsymbol{x}) \ne h_{\boldsymbol{\theta}'}(\boldsymbol{x})\}} (f_{\boldsymbol{\theta}} + f_{\boldsymbol{\theta}'}) d\lambda \le 2^{\widetilde{\gamma}+1} C_{\mathscr{C}}^{\widetilde{\gamma}} M^{-(d-1)}. \tag{36}$$

Here, $\rho_{Ham}$ is the Hamming distance defined in equation 27.

*Proof.* We prove each case separately.

1) Let $K_b > 0$ be a constant satisfying equation 5. For all $\boldsymbol{x}' \in \partial\Omega_{h_b} = \left\{ \boldsymbol{x} \in \mathcal{X} \mid b(\boldsymbol{x}^{(d)}) = x_d \right\}$ (see Remark 2), and all $\boldsymbol{x} \in \mathcal{X}$, we know that

$$\|\boldsymbol{x} - \boldsymbol{x}'\|_2^2 = \left\| \boldsymbol{x}^{(d)} - \boldsymbol{x}'^{(d)} \right\|_2^2 + |x_d - b(\boldsymbol{x}'^{(d)})|^2. \tag{37}$$

We set $t := \left\| \boldsymbol{x}^{(d)} - \boldsymbol{x}'^{(d)} \right\|_2$ and $w := |x_d - b(\boldsymbol{x}^{(d)})|$, and consider two possibilities:

- If $w - K_b t^\alpha \le 0$, we get $t^2 \ge (w/K_b)^{2/\alpha}$, and

$$\|\boldsymbol{x} - \boldsymbol{x}'\|_2^2 = t^2 + |x_d - b(\boldsymbol{x}'^{(d)})|^2 \ge (w/K_b)^{2/\alpha}. \tag{38}$$

- If $w - K_b t^\alpha \ge 0$, condition C2) implies

$$w = |x_d - b(\boldsymbol{x}^{(d)})| \le |x_d - b(\boldsymbol{x}'^{(d)})| + |b(\boldsymbol{x}^{(d)}) - b(\boldsymbol{x}'^{(d)})|$$
$$\le |x_d - b(\boldsymbol{x}'^{(d)})| + K_b \left\| \boldsymbol{x}^{(d)} - \boldsymbol{x}'^{(d)} \right\|_2^\alpha$$
$$= |x_d - b(\boldsymbol{x}'^{(d)})| + K_b t^\alpha,$$

therefore

$$\|\boldsymbol{x} - \boldsymbol{x}'\|_2^2 = t^2 + |x_d - b(\boldsymbol{x}'^{(d)})|^2 \ge t^2 + (w - K_b t^\alpha)^2. \tag{39}$$

So, when $\alpha = 1$, it follows that

$$
\begin{aligned}
\|\boldsymbol{x} - \boldsymbol{x}'\|_2^2 &\geq (1 + K_b^2)t^2 - 2wK_b t + w^2 \\
&\geq \frac{4w^2(1 + K_b^2) - (2wK_b)^2}{4(1 + K_b^2)} \\
&= \frac{w^2}{1 + K_b^2}.
\end{aligned}
\tag{40}
$$

Otherwise, $\alpha < 1$ and we consider a threshold $t_0 := (w/(2K_b))^{1/\alpha}$ to minimize the right-hand side of equation 39 on $t$. By cases:

    − If $t \leq t_0$, we have that $w - K_b t^\alpha \geq w - K_b t_0^\alpha = w/2 > 0$, therefore

$$
\|\boldsymbol{x} - \boldsymbol{x}'\|_2^2 \geq t^2 + (w - K_b t_0^\alpha)^2 = t^2 + (w/2)^2 \geq (w/2)^2.
\tag{41}
$$

    − If $t > t_0$, we obtain

$$
\|\boldsymbol{x} - \boldsymbol{x}'\|_2^2 \geq t^2 + (w - K_b t^\alpha)^2 \geq t^2 > (w/(2K_b))^{2/\alpha}.
\tag{42}
$$

Thus, equation 38, equation 40, equation 41 and equation 42, imply

$$
\|\boldsymbol{x} - \boldsymbol{x}'\|_2 \geq \left( \frac{1}{\widetilde{K_b}} |x_d - b(\boldsymbol{x}^{(d)})| \right)^{\frac{1}{\alpha}},
$$

where $\widetilde{K_b}$ is as in equation 29. Then, by the above inequality and equation 37, we arrive at

$$
\left( \frac{1}{\widetilde{K_b}} |x_d - b(\boldsymbol{x}^{(d)})| \right)^{\frac{1}{\alpha}} \leq \inf_{\boldsymbol{x}' \in \partial\Omega_{h_b}} \|\boldsymbol{x} - \boldsymbol{x}'\|_2 \leq |x_d - b(\boldsymbol{x}^{(d)})|.
$$

Here, the upper bound was obtained using that $(x_1, \ldots, x_{d-1}, b(\boldsymbol{x}^{(d)})) \in \partial\Omega_{h_b}$.

2) From equation 14, we know that $\operatorname{supp}\varphi \subset (-1, 1)^{d-1} = \{\boldsymbol{z} \in \mathbb{R}^{d-1} \mid \|\boldsymbol{z}\|_\infty < 1\}$. Hence, by equation 13 and equation 15,

$$
\begin{aligned}
\operatorname{supp}\varphi_j &= \{\boldsymbol{z} \in [0, 1]^{d-1} \mid M(\boldsymbol{z} - \boldsymbol{v}_j/M) \in \operatorname{supp}\varphi\} \\
&\subset \{\boldsymbol{z} \in [0, 1]^{d-1} \mid \|M(\boldsymbol{z} - \boldsymbol{v}_j/M)\|_\infty < 1\} \\
&= \operatorname{int} Q_j \quad \text{for all} \quad j \in \{1, \ldots, m\},
\end{aligned}
$$

where $\operatorname{int} Q_j$ denotes the interior of $Q_j$. Therefore,

$$
\operatorname{supp}\varphi_j \cap \operatorname{supp}\varphi_k \subseteq \operatorname{int} Q_j \cap \operatorname{int} Q_k = \emptyset \quad \text{for all} \quad j, k \in \{1, \ldots, m\} \quad \text{with} \quad j \neq k,
$$

since $P$ is a partition of $[0, 1]^{d-1}$ up to boundaries (see equation 13). Thus, equation 30 holds, and equation 31 follows from the definition of $\boldsymbol{\varphi}$ in equation 16 together with equation 30.

3) Let $\boldsymbol{\theta} \in \Theta$ and $\boldsymbol{z}, \boldsymbol{w} \in [0, 1]^{d-1}$. By equation 15 and since $\varphi$ is Hölder continuous with exponent $\alpha$ and constant $K_\varphi$, we obtain

$$
\begin{aligned}
|\varphi_j(\boldsymbol{z}) - \varphi_j(\boldsymbol{w})| &= |\varphi(M(\boldsymbol{z} - \boldsymbol{v}_j/M)) - \varphi(M(\boldsymbol{w} - \boldsymbol{v}_j/M))| \\
&\leq K_\varphi M^\alpha \|\boldsymbol{z} - \boldsymbol{w}\|_2^\alpha.
\end{aligned}
$$

Furthermore, equation 30 and equation 31 imply that there exist $j_{\boldsymbol{z}}, j_{\boldsymbol{w}} \in \{1, \ldots, m\}$, such that

$$
\boldsymbol{\varphi}(\boldsymbol{z}) = C_{\mathscr{C}} \varphi_{j_{\boldsymbol{z}}}(\boldsymbol{z}) \boldsymbol{e}_{j_{\boldsymbol{z}}}, \quad \boldsymbol{\varphi}(\boldsymbol{w}) = C_{\mathscr{C}} \varphi_{j_{\boldsymbol{w}}}(\boldsymbol{w}) \boldsymbol{e}_{j_{\boldsymbol{w}}},
$$

and $\varphi_{j_z}(\boldsymbol{w}) = \varphi_{j_{\boldsymbol{w}}}(\boldsymbol{z}) = 0$ whenever $j_{\boldsymbol{z}} \neq j_{\boldsymbol{w}}$. Thus,

$$
\begin{aligned}
C_{\mathscr{C}}^{-1}|\boldsymbol{\theta} \cdot (\boldsymbol{\varphi}(\boldsymbol{z}) - \boldsymbol{\varphi}(\boldsymbol{w}))| &= |\boldsymbol{\theta} \cdot (\varphi_{j_z}(\boldsymbol{z})\boldsymbol{e}_{j_z} - \varphi_{j_{\boldsymbol{w}}}(\boldsymbol{w})\boldsymbol{e}_{j_{\boldsymbol{w}}})| \\
&\leq \begin{cases} |\varphi_j(\boldsymbol{z}) - \varphi_j(\boldsymbol{w})| & \text{if } j_{\boldsymbol{w}} = j_{\boldsymbol{z}} = j \\ |\varphi_{j_z}(\boldsymbol{z}) - \varphi_{j_z}(\boldsymbol{w})| + |\varphi_{j_{\boldsymbol{w}}}(\boldsymbol{z}) - \varphi_{j_{\boldsymbol{w}}}(\boldsymbol{w})| & \text{if } j_{\boldsymbol{w}} \neq j_{\boldsymbol{z}} \end{cases} \\
&\leq 2K_\varphi M^\alpha \|\boldsymbol{z} - \boldsymbol{w}\|_2^\alpha .
\end{aligned}
\tag{43}
$$

We know that $K_{b_0} > 0$ is the constant satisfying equation 5 for $b_0 \in \mathscr{C}$, so using equation 16, equation 43, we get

$$
\begin{aligned}
|b_{\boldsymbol{\theta}}(\boldsymbol{z}) - b_{\boldsymbol{\theta}}(\boldsymbol{w})| &= |b_0(\boldsymbol{z}) + \boldsymbol{\theta} \cdot \boldsymbol{\varphi}(\boldsymbol{z}) - (b_0(\boldsymbol{w}) + \boldsymbol{\theta} \cdot \boldsymbol{\varphi}(\boldsymbol{w}))| \\
&\leq |b_0(\boldsymbol{z}) - b_0(\boldsymbol{w})| + |\boldsymbol{\theta} \cdot \boldsymbol{\varphi}(\boldsymbol{z}) - \boldsymbol{\theta} \cdot \boldsymbol{\varphi}(\boldsymbol{w})| \\
&\leq K_{b_0} \|\boldsymbol{z} - \boldsymbol{w}\|_2^\alpha + |\boldsymbol{\theta} \cdot (\boldsymbol{\varphi}(\boldsymbol{z}) - \boldsymbol{\varphi}(\boldsymbol{w}))| \\
&\leq (K_{b_0} + 2K_\varphi C_{\mathscr{C}} M^\alpha) \|\boldsymbol{z} - \boldsymbol{w}\|_2^\alpha .
\end{aligned}
$$

We assumed in equation 18 that $C_{\mathscr{C}} \leq M^{-\alpha}/4$, then

$$
|b_{\boldsymbol{\theta}}(\boldsymbol{z}) - b_{\boldsymbol{\theta}}(\boldsymbol{w})| \leq (K_{b_0} + K_\varphi/2) \|\boldsymbol{z} - \boldsymbol{w}\|_2^\alpha .
$$

Since $\boldsymbol{\theta}$ is arbitrary, we conclude that $\mathscr{C}_\Theta$ satisfies C2) uniformly with $\alpha$ and $K_\Theta$ as in equation 32.

4) We rewrite equation 24 and equation 25 as

$$
\mathcal{T}_{\boldsymbol{\theta}} := \left\{ \boldsymbol{x} \in \mathcal{X} \mid b_0(\boldsymbol{x}^{(d)}) - C_{\mathscr{C}} \leq x_d \leq b_0(\boldsymbol{x}^{(d)}) + C_{\mathscr{C}} + 2\boldsymbol{\theta} \cdot \boldsymbol{\varphi}(\boldsymbol{x}^{(d)}) \right\} \quad \text{and}
$$

$$
\mathcal{R} := \left\{ \boldsymbol{x} \in \mathcal{X} \mid b_0(\boldsymbol{x}^{(d)}) - C_{\mathscr{C}} \leq x_d \leq b_0(\boldsymbol{x}^{(d)}) + 3C_{\mathscr{C}} \right\} .
$$

Then, by equation 15 and equation 31, we get

$$
\mathcal{T}_{\boldsymbol{\theta}} = \left\{ \boldsymbol{x} \in \mathcal{X} \mid b_0(\boldsymbol{x}^{(d)}) - C_{\mathscr{C}} \leq x_d \leq b_0(\boldsymbol{x}^{(d)}) + C_{\mathscr{C}} + 2C_{\mathscr{C}}\varphi_j(\boldsymbol{x}^{(d)})(\boldsymbol{\theta} \cdot \boldsymbol{e}_j), \text{ for some } j \right\},
\tag{44}
$$

and since $\varphi_j(\boldsymbol{x}^{(d)})(\boldsymbol{\theta} \cdot \boldsymbol{e}_j) \leq 1$, we obtain

$$
\mathcal{T}_{\boldsymbol{\theta}} \subseteq \mathcal{R} \quad \text{for all} \quad \boldsymbol{\theta} \in \Theta.
$$

Moreover, from the above it is clear that equation 33 holds.

5) The constant $C_{\boldsymbol{\theta}}$ was defined in equation 26 such that $\int_{\mathcal{X}} f_{\boldsymbol{\theta}} d\lambda = 1$, for all $\boldsymbol{\theta} \in \Theta$. Therefore,

$$
1 = \int_{\mathcal{X}} f_{\boldsymbol{\theta}}(\boldsymbol{x})d\boldsymbol{x} = \int_{\mathcal{R}} f_{\boldsymbol{\theta}}(\boldsymbol{x})d\boldsymbol{x} + \int_{\mathcal{X} \backslash \mathcal{R}} C_{\boldsymbol{\theta}} d\boldsymbol{x} = \int_{\mathcal{R}} f_{\boldsymbol{\theta}}(\boldsymbol{x})d\boldsymbol{x} + C_{\boldsymbol{\theta}}(1 - \lambda(\mathcal{R})).
$$

By equation 18 and equation 25, we obtain $\lambda(\mathcal{R}) = 4C_{\mathscr{C}}$ and

$$
C_{\boldsymbol{\theta}} = \frac{1 - \int_{\mathcal{R}} f_{\boldsymbol{\theta}}(\boldsymbol{x})d\boldsymbol{x}}{1 - 4C_{\mathscr{C}}}.
$$

Furthermore, we use equation 14, equation 15, equation 16, equation 18 and equation 31 to see that

$$
b_{\boldsymbol{\theta}}(\boldsymbol{z}) = b_0(\boldsymbol{z}) + C_{\mathscr{C}}\varphi_j(\boldsymbol{z})(\boldsymbol{\theta} \cdot \boldsymbol{e}_j) \in [0, 1] \quad \text{and}
\tag{45}
$$

$$
(C_{\mathscr{C}} - \boldsymbol{\theta} \cdot \boldsymbol{\varphi}(\boldsymbol{z}))^{\widetilde{\gamma}-1} = C_{\mathscr{C}}^{\widetilde{\gamma}-1}(1 - \varphi_j(\boldsymbol{z})(\boldsymbol{\theta} \cdot \boldsymbol{e}_j))^{\widetilde{\gamma}-1} \in [0, 1],
\tag{46}
$$

for each $\boldsymbol{z} \in [0,1]^{d-1}$, for some $j = j(\boldsymbol{z}) \in \{1, \ldots, m\}$ and for all $\widetilde{\gamma} \geq 1$. Then, equation 26 implies

$$0 \leq f_{\boldsymbol{\theta}} \mathbb{1}_{\mathcal{R}}(\boldsymbol{x}) = \frac{1}{2} \left( |x_d - b_{\boldsymbol{\theta}}(\boldsymbol{x}^{(d)})|^{\widetilde{\gamma}-1} \mathbb{1}_{\mathcal{R} \cap \mathcal{T}_{\boldsymbol{\theta}}}(\boldsymbol{x}) + (C_{\mathscr{C}} - \boldsymbol{\theta} \cdot \boldsymbol{\varphi}(\boldsymbol{x}^{(d)}))^{\widetilde{\gamma}-1} \mathbb{1}_{\mathcal{R} \setminus \mathcal{T}_{\boldsymbol{\theta}}}(\boldsymbol{x}) \right)$$

$$\leq \frac{1}{2} \left( \mathbb{1}_{\mathcal{R} \cap \mathcal{T}_{\boldsymbol{\theta}}}(\boldsymbol{x}) + \mathbb{1}_{\mathcal{R} \setminus \mathcal{T}_{\boldsymbol{\theta}}}(\boldsymbol{x}) \right)$$

$$= \frac{1}{2} \mathbb{1}_{\mathcal{R}}(\boldsymbol{x}),$$

and

$$1 - 2C_{\mathscr{C}} = 1 - \lambda(\mathcal{R})/2 \leq 1 - \int_{\mathcal{R}} f_{\boldsymbol{\theta}}(\boldsymbol{x}) d\boldsymbol{x} \leq 1.$$

Hence,

$$\frac{1 - 2C_{\mathscr{C}}}{1 - 4C_{\mathscr{C}}} \leq \frac{1 - \int_{\mathcal{R}} f_{\boldsymbol{\theta}}(\boldsymbol{x}) d\boldsymbol{x}}{1 - 4C_{\mathscr{C}}} \leq \frac{1}{1 - 4C_{\mathscr{C}}}$$

and equation 34 holds.

6) Let $\widetilde{\gamma} := \gamma/\alpha$ in equation 26 with $\gamma \geq \alpha$, and let $\boldsymbol{\theta} \in \Theta$ be arbitrary. By items 1) and 3) above, we get

$$\left( \frac{1}{\widetilde{K_{\Theta}}} |x_d - b_{\boldsymbol{\theta}}(\boldsymbol{x}^{(d)})| \right)^{\frac{1}{\alpha}} \leq \text{dist}\left(\boldsymbol{x}, \partial\Omega_{h_{\boldsymbol{\theta}}}\right).$$

Therefore, with the notation in condition C3), we obtain

$$B_{\varepsilon}^{h_{\boldsymbol{\theta}}} = \{\boldsymbol{x} \in \mathcal{X} \mid \text{dist}\left(\boldsymbol{x}, \partial\Omega_{h_{\boldsymbol{\theta}}}\right) \leq \varepsilon\}$$

$$\subseteq \left\{ \boldsymbol{x} \in \mathcal{X} \mid |x_d - b_{\boldsymbol{\theta}}(\boldsymbol{x}^{(d)})| \leq \widetilde{K_{\Theta}} \varepsilon^{\alpha} \right\}, \quad \text{for all} \quad \varepsilon > 0. \tag{47}$$

By the previous item 4), $\mathcal{T}_{\boldsymbol{\theta}} \subseteq \mathcal{R}$, and therefore

$$\mathcal{R} \cap \mathcal{T}_{\boldsymbol{\theta}} = \mathcal{T}_{\boldsymbol{\theta}}. \tag{48}$$

So, we consider the following cases.

- If $\widetilde{K_{\Theta}} \varepsilon^{\alpha} \leq C_{\mathscr{C}}$, then equation 24, equation 48 and equation 47 imply $B_{\varepsilon}^{h_{\boldsymbol{\theta}}} \subseteq \mathcal{T}_{\boldsymbol{\theta}} = \mathcal{R} \cap \mathcal{T}_{\boldsymbol{\theta}}$. Therefore,

$$\mu_{\boldsymbol{\theta}}(B_{\varepsilon}^{h_{\boldsymbol{\theta}}}) = \mu_{\boldsymbol{\theta}}((\mathcal{R} \cap \mathcal{T}_{\boldsymbol{\theta}}) \cap B_{\varepsilon}^{h_{\boldsymbol{\theta}}})$$

$$= \int_{B_{\varepsilon}^{h_{\boldsymbol{\theta}}}} f_{\boldsymbol{\theta}} \mathbb{1}_{\mathcal{R} \cap \mathcal{T}_{\boldsymbol{\theta}}} d\lambda$$

$$\leq \frac{1}{2} \int_{|x_d - b_{\boldsymbol{\theta}}(\boldsymbol{x}^{(d)})| \leq \widetilde{K_{\Theta}} \varepsilon^{\alpha}} |x_d - b_{\boldsymbol{\theta}}(\boldsymbol{x}^{(d)})|^{\widetilde{\gamma}-1} d\boldsymbol{x}$$

$$\leq \frac{1}{2} (\widetilde{K_{\Theta}} \varepsilon^{\alpha})^{\widetilde{\gamma}-1} \int_{[0,1]^{d-1}} \int_{|x_d - b_{\boldsymbol{\theta}}(\boldsymbol{x}^{(d)})| \leq \widetilde{K_{\Theta}} \varepsilon^{\alpha}} 1 dx_d d\boldsymbol{x}^{(d)}$$

$$\leq (\widetilde{K_{\Theta}} \varepsilon^{\alpha})^{\widetilde{\gamma}}$$

$$= \widetilde{K_{\Theta}}^{\widetilde{\gamma}} \varepsilon^{\gamma}. \tag{49}$$

The inequality above shows that condition C3) is satisfied with constant $\widetilde{K_{\Theta}}^{\widetilde{\gamma}}$ and margin exponent $\gamma$.

- If $\widetilde{K_{\Theta}} \varepsilon^{\alpha} > C_{\mathscr{C}}$. We have, similarly to equation 49, that

$$\mu_{\boldsymbol{\theta}}(B_{\varepsilon}^{h_{\boldsymbol{\theta}}}) = \int_{B_{\varepsilon}^{h_{\boldsymbol{\theta}}}} f_{\boldsymbol{\theta}} \mathbb{1}_{\mathcal{R} \cap \mathcal{T}_{\boldsymbol{\theta}}} d\lambda + \int_{B_{\varepsilon}^{h_{\boldsymbol{\theta}}}} f_{\boldsymbol{\theta}} \mathbb{1}_{\mathcal{X} \setminus (\mathcal{R} \cap \mathcal{T}_{\boldsymbol{\theta}})} d\lambda$$

$$\leq \widetilde{K_{\Theta}}^{\widetilde{\gamma}} \varepsilon^{\gamma} + \int_{\mathcal{X} \setminus (\mathcal{R} \cap \mathcal{T}_{\boldsymbol{\theta}})} f_{\boldsymbol{\theta}} d\lambda.$$

By equation 26,

$$\int_{\mathcal{X}\setminus(\mathcal{R}\cap\mathcal{T}_{\boldsymbol{\theta}})} f_{\boldsymbol{\theta}} d\lambda = \int_{\mathcal{X}} \left( \frac{1}{2}(C_{\mathscr{C}} - \boldsymbol{\theta}\cdot\boldsymbol{\varphi}(\boldsymbol{x}^{(d)}))^{\widetilde{\gamma}-1} \mathbb{1}_{\mathcal{R}\setminus\mathcal{T}_{\boldsymbol{\theta}}}(\boldsymbol{x}) + C_{\boldsymbol{\theta}} \mathbb{1}_{\mathcal{X}\setminus\mathcal{R}}(\boldsymbol{x}) \right) d\boldsymbol{x}.$$

Using equation 34, we obtain

$$\mu_{\boldsymbol{\theta}}(B_{\varepsilon}^{h_{\boldsymbol{\theta}}}) \le \widetilde{K_{\Theta}}^{\widetilde{\gamma}}\varepsilon^{\gamma} + \int_{\mathcal{X}} \left( \frac{1}{2}(C_{\mathscr{C}} - \boldsymbol{\theta}\cdot\boldsymbol{\varphi}(\boldsymbol{x}^{(d)}))^{\widetilde{\gamma}-1} \mathbb{1}_{\mathcal{R}\setminus\mathcal{T}_{\boldsymbol{\theta}}}(\boldsymbol{x}) + C_{\boldsymbol{\theta}} \mathbb{1}_{\mathcal{X}\setminus\mathcal{R}}(\boldsymbol{x}) \right) d\boldsymbol{x}$$

$$\le \widetilde{K_{\Theta}}^{\widetilde{\gamma}}\varepsilon^{\gamma} + \frac{1}{2}C_{\mathscr{C}}^{\widetilde{\gamma}-1}\int_{\mathcal{R}\setminus\mathcal{T}_{\boldsymbol{\theta}}} 1 d\boldsymbol{x} + (1 - 4C_{\mathscr{C}})^{-1}\int_{\mathcal{X}\setminus\mathcal{R}} 1 d\boldsymbol{x}$$

$$\le \widetilde{K_{\Theta}}^{\widetilde{\gamma}}\varepsilon^{\gamma} + \frac{1}{2}C_{\mathscr{C}}^{\widetilde{\gamma}-1} + (1 - 4C_{\mathscr{C}})^{-1}.$$

So, we use $\varepsilon^{\gamma} > (C_{\mathscr{C}}/\widetilde{K_{\Theta}})^{\gamma/\alpha}$ in the above inequality and set

$$C_0 := \left( \widetilde{K_{\Theta}}^{\widetilde{\gamma}} + \left( \frac{1}{2}C_{\mathscr{C}}^{\widetilde{\gamma}-1} + (1 - 4C_{\mathscr{C}})^{-1} \right) (C_{\mathscr{C}}/\widetilde{K_{\Theta}})^{-\widetilde{\gamma}} \right),$$

to obtain

$$\mu_{\boldsymbol{\theta}}(B_{\varepsilon}^{h_{\boldsymbol{\theta}}}) \le C_0 \varepsilon^{\gamma}.$$

Thus, we conclude that C3) is fulfilled with constant $C_0$ and margin exponent $\gamma$.

In both cases above, for all $(h_{\boldsymbol{\theta}}, \mu_{\boldsymbol{\theta}}) \in H_{\mathscr{C}_{\Theta}} \times \mathcal{M}_{\Theta}$, the margin condition holds with the same margin exponent $\gamma$ and in particular for every constant $C \ge C_0$.

7) For all $\boldsymbol{\theta}, \boldsymbol{\theta}' \in \Theta$ with $\rho_{Ham}(\boldsymbol{\theta}, \boldsymbol{\theta}') = 1$, the index set $J := J(\boldsymbol{\theta}, \boldsymbol{\theta}')$ defined in equation 27 contains exactly one element, i.e., $\#J = 1$. Moreover, there exists a unique $j^* \in \{1, \ldots, m\}$, such that $J = \{j^*\}$. We assume without loss of generality that $\boldsymbol{\theta} - \boldsymbol{\theta}' = \boldsymbol{e}_{j^*}$. Then, by equation 15 and equation 16, we get

$$\boldsymbol{\theta}\cdot\boldsymbol{\varphi}(\boldsymbol{z}) - \boldsymbol{\theta}'\cdot\boldsymbol{\varphi}(\boldsymbol{z}) = (\boldsymbol{\theta} - \boldsymbol{\theta}')\cdot\boldsymbol{\varphi}(\boldsymbol{z})$$
$$= \boldsymbol{e}_{j^*}\cdot\boldsymbol{\varphi}(\boldsymbol{z})$$
$$= C_{\mathscr{C}}\left(\boldsymbol{e}_{j^*}\cdot(\varphi_1(\boldsymbol{z}), \ldots, \varphi_m(\boldsymbol{z}))\right)$$
$$= C_{\mathscr{C}}\varphi_{j^*}(\boldsymbol{z}), \quad \text{for all} \quad \boldsymbol{z} \in [0,1]^{d-1}.$$

Therefore,

$$\boldsymbol{\theta}\cdot\boldsymbol{\varphi}(\boldsymbol{z}) = \boldsymbol{\theta}'\cdot\boldsymbol{\varphi}(\boldsymbol{z}) \quad \text{for all} \quad \boldsymbol{z} \in [0,1]^{d-1}\setminus\text{supp}\,\varphi_{j^*}, \tag{50}$$

and

$$f_{\boldsymbol{\theta}}\mathbb{1}_{\mathcal{R}}(\boldsymbol{x}) = f_{\boldsymbol{\theta}'}\mathbb{1}_{\mathcal{R}}(\boldsymbol{x}) \quad \text{for all} \quad \boldsymbol{x} \in ([0,1]^{d-1}\setminus\text{supp}\,\varphi_{j^*}) \times [0,1], \tag{51}$$

since equation 24, equation 26, equation 33 and equation 34 depend only on the expression $\boldsymbol{\theta}\cdot\boldsymbol{\varphi}$ in the region $\mathcal{R}$.

By equation 18, equation 25, equation 26 and equation 34, we see that

$$\int_{\mathcal{X}\setminus\mathcal{R}} |f_{\boldsymbol{\theta}} - f_{\boldsymbol{\theta}'}| d\lambda = |C_{\boldsymbol{\theta}} - C_{\boldsymbol{\theta}'}| \int_{\mathcal{X}\setminus\mathcal{R}} 1 d\lambda$$

$$= \frac{1 - \lambda(\mathcal{R})}{1 - 4C_{\mathscr{C}}} \left| 1 - \int_{\mathcal{R}} f_{\boldsymbol{\theta}} d\lambda - \left( 1 - \int_{\mathcal{R}} f_{\boldsymbol{\theta}'} d\lambda \right) \right|$$

$$= \left| \int_{\mathcal{R}} f_{\boldsymbol{\theta}'} d\lambda - \int_{\mathcal{R}} f_{\boldsymbol{\theta}} d\lambda \right|$$

$$\le \int_{\mathcal{R}} |f_{\boldsymbol{\theta}} - f_{\boldsymbol{\theta}'}| d\lambda.$$

Then, by the above inequality and equation 51,

$$
\begin{aligned}
\int_{\mathcal{X}} |f_{\boldsymbol{\theta}} - f_{\boldsymbol{\theta}'}| \, d\lambda &= \int_{\mathcal{R} \cup (\mathcal{X} \setminus \mathcal{R})} |f_{\boldsymbol{\theta}} - f_{\boldsymbol{\theta}'}| \, d\lambda \\
&\leq 2 \int_{\mathcal{R}} |f_{\boldsymbol{\theta}} - f_{\boldsymbol{\theta}'}| \, d\lambda \\
&= 2 \int_{\mathcal{R} \cap \mathcal{S}_{j^*}} |f_{\boldsymbol{\theta}} - f_{\boldsymbol{\theta}'}| \, d\lambda \quad \text{where} \quad \mathcal{S}_{j^*} := \operatorname{supp} \varphi_{j^*} \times [0,1].
\end{aligned}
\tag{52}
$$

To analyze the upper bound in equation 52, we look closely at the set $\mathcal{R} \cap \mathcal{S}_{j^*}$ as well as the behavior of $f_{\boldsymbol{\theta}}$ and $f_{\boldsymbol{\theta}'}$ in this region. Note that

$$
\mathcal{R} \cap \mathcal{S}_{j^*} = \mathcal{U}_{\boldsymbol{\theta}''}^{(1)} \cup \mathcal{U}_{\boldsymbol{\theta}''}^{(2)} \quad \text{with} \quad \mathcal{U}_{\boldsymbol{\theta}''}^{(1)} \cap \mathcal{U}_{\boldsymbol{\theta}''}^{(2)} = \emptyset,
\tag{53}
$$

where

$$
\mathcal{U}_{\boldsymbol{\theta}''}^{(1)} := ((\mathcal{R} \cap \mathcal{T}_{\boldsymbol{\theta}''}) \cap \mathcal{S}_{j^*}) \quad \text{and} \quad \mathcal{U}_{\boldsymbol{\theta}''}^{(2)} := ((\mathcal{R} \setminus \mathcal{T}_{\boldsymbol{\theta}''}) \cap \mathcal{S}_{j^*}) \quad \text{for all} \quad \boldsymbol{\theta}'' \in \Theta.
$$

Since we assumed $\boldsymbol{\theta} - \boldsymbol{\theta}' = \boldsymbol{e}_{j^*}$, we have $\theta_{j^*} = 1$ and $\theta'_{j^*} = 0$. So, we consider the following two cases.

- For $\boldsymbol{\theta}$: We use that $\theta_{j^*} = 1$, equation 30 and equation 48 to obtain

$$
\begin{aligned}
\mathcal{U}_{\boldsymbol{\theta}}^{(1)} &= \mathcal{T}_{\boldsymbol{\theta}} \cap \mathcal{S}_{j^*} \\
&= \left\{ \boldsymbol{x} \in \mathcal{S}_{j^*} \mid |x_d - b_{\boldsymbol{\theta}}(\boldsymbol{x}^{(d)})| \leq C_{\mathscr{C}} + \boldsymbol{\theta} \cdot \boldsymbol{\varphi}(\boldsymbol{x}^{(d)}) \right\} \\
&= \left\{ \boldsymbol{x} \in \mathcal{S}_{j^*} \mid |x_d - (b_0(\boldsymbol{x}^{(d)}) + C_{\mathscr{C}}(\theta_{j^*} \varphi_{j^*}(\boldsymbol{x}^{(d)})))| \leq C_{\mathscr{C}} + C_{\mathscr{C}}(\theta_{j^*} \varphi_{j^*}(\boldsymbol{x}^{(d)})) \right\} \\
&= \left\{ \boldsymbol{x} \in \mathcal{S}_{j^*} \mid |x_d - b_0(\boldsymbol{x}^{(d)}) - C_{\mathscr{C}} \varphi_{j^*}(\boldsymbol{x}^{(d)})| \leq C_{\mathscr{C}} \left( 1 + \varphi_{j^*}(\boldsymbol{x}^{(d)}) \right) \right\}.
\end{aligned}
\tag{54}
$$

By equation 30 and equation 33, we have

$$
\mathcal{U}_{\boldsymbol{\theta}}^{(2)} = \left\{ \boldsymbol{x} \in \mathcal{S}_{j^*} : b_0(\boldsymbol{x}^{(d)}) + C_{\mathscr{C}} + 2C_{\mathscr{C}} \varphi_{j^*}(\boldsymbol{x}^{(d)}) \leq x_d \leq b_0(\boldsymbol{x}^{(d)}) + 3C_{\mathscr{C}} \right\}.
\tag{55}
$$

Moreover, equation 26 and equation 30 imply

$$
\begin{aligned}
f_{\boldsymbol{\theta}} \mathbb{1}_{\mathcal{U}_{\boldsymbol{\theta}}^{(1)}}(\boldsymbol{x}) &= \frac{1}{2} |x_d - b_0(\boldsymbol{x}^{(d)}) - C_{\mathscr{C}} \varphi_{j^*}(\boldsymbol{x}^{(d)})|^{\widetilde{\gamma}-1} \mathbb{1}_{\mathcal{U}_{\boldsymbol{\theta}}^{(1)}}(\boldsymbol{x}) \quad \text{and} \\
f_{\boldsymbol{\theta}} \mathbb{1}_{\mathcal{U}_{\boldsymbol{\theta}}^{(2)}}(\boldsymbol{x}) &= \frac{1}{2} C_{\mathscr{C}}^{\widetilde{\gamma}-1} (1 - \varphi_{j^*}(\boldsymbol{x}^{(d)}))^{\widetilde{\gamma}-1} \mathbb{1}_{\mathcal{U}_{\boldsymbol{\theta}}^{(2)}}(\boldsymbol{x}).
\end{aligned}
\tag{56}
$$

- For $\boldsymbol{\theta}'$: We know that $\theta'_{j^*} = 0$ and analogously,

$$
\begin{aligned}
\mathcal{U}_{\boldsymbol{\theta}'}^{(1)} &= \mathcal{T}_{\boldsymbol{\theta}'} \cap \mathcal{S}_{j^*} = \left\{ \boldsymbol{x} \in \mathcal{S}_{j^*} \mid |x_d - b_0(\boldsymbol{x}^{(d)})| \leq C_{\mathscr{C}} \right\}, \\
\mathcal{U}_{\boldsymbol{\theta}'}^{(2)} &= \left\{ \boldsymbol{x} \in \mathcal{S}_{j^*} : b_0(\boldsymbol{x}^{(d)}) + C_{\mathscr{C}} \leq x_d \leq b_0(\boldsymbol{x}^{(d)}) + 3C_{\mathscr{C}} \right\}, \\
f_{\boldsymbol{\theta}'} \mathbb{1}_{\mathcal{U}_{\boldsymbol{\theta}'}^{(1)}}(\boldsymbol{x}) &= \frac{1}{2} |x_d - b_0(\boldsymbol{x}^{(d)})|^{\widetilde{\gamma}-1} \mathbb{1}_{\mathcal{U}_{\boldsymbol{\theta}'}^{(1)}}(\boldsymbol{x}) \quad \text{and} \\
f_{\boldsymbol{\theta}'} \mathbb{1}_{\mathcal{U}_{\boldsymbol{\theta}'}^{(2)}}(\boldsymbol{x}) &= \frac{1}{2} C_{\mathscr{C}}^{\widetilde{\gamma}-1} \mathbb{1}_{\mathcal{U}_{\boldsymbol{\theta}'}^{(2)}}(\boldsymbol{x}).
\end{aligned}
\tag{57}
$$

Thus, by equation 52 and equation 53,

$$
\begin{aligned}
\frac{1}{2} \int_{\mathcal{X}} |f_{\boldsymbol{\theta}} - f_{\boldsymbol{\theta}'}| \, d\lambda &\leq \int_{\mathcal{R} \cap \mathcal{S}_{j^*}} f_{\boldsymbol{\theta}} \, d\lambda + \int_{\mathcal{R} \cap \mathcal{S}_{j^*}} f_{\boldsymbol{\theta}'} \, d\lambda, \\
&= \int_{\mathcal{U}_{\boldsymbol{\theta}}^{(1)}} f_{\boldsymbol{\theta}} \, d\lambda + \int_{\mathcal{U}_{\boldsymbol{\theta}}^{(2)}} f_{\boldsymbol{\theta}} \, d\lambda + \int_{\mathcal{U}_{\boldsymbol{\theta}'}^{(1)}} f_{\boldsymbol{\theta}'} \, d\lambda + \int_{\mathcal{U}_{\boldsymbol{\theta}'}^{(2)}} f_{\boldsymbol{\theta}'} \, d\lambda,
\end{aligned}
\tag{58}
$$

where using equation 52, equation 53, equation 54, equation 55, equation 56, and equation 57, we obtain first

- $\displaystyle\int_{\mathcal{U}_{\boldsymbol{\theta}}^{(1)}} f_{\boldsymbol{\theta}}\,d\lambda \leq \frac{1}{2}\int_{|x_d-b_0(\boldsymbol{x}^{(d)})-C_{\mathscr{C}}\varphi_{j^*}(\boldsymbol{x}^{(d)})|\leq C_{\mathscr{C}}\left(1+\varphi_{j^*}(\boldsymbol{x}^{(d)})\right)} |x_d-b_0(\boldsymbol{x}^{(d)})-C_{\mathscr{C}}\varphi_{j^*}(\boldsymbol{x}^{(d)})|^{\widetilde{\gamma}-1}d\boldsymbol{x}$

$$= \frac{1}{\widetilde{\gamma}}\int_{[0,1]^{d-1}} C_{\mathscr{C}}^{\widetilde{\gamma}}\left(1+\varphi_{j^*}(\boldsymbol{x}^{(d)})\right)^{\widetilde{\gamma}}d\boldsymbol{x}^{(d)}$$

$$= \frac{1}{\widetilde{\gamma}}\int_{[0,1]^{d-1}} C_{\mathscr{C}}^{\widetilde{\gamma}}\left(1+\varphi\left(M(\boldsymbol{z}-\boldsymbol{v}_{j^*}/M)\right)\right)^{\widetilde{\gamma}}d\boldsymbol{z}$$

$$= \frac{1}{\widetilde{\gamma}}C_{\mathscr{C}}^{\widetilde{\gamma}}M^{-(d-1)}\|1+\varphi\|_{L^{\widetilde{\gamma}}([0,1]^{d-1})}^{\widetilde{\gamma}}. \tag{59}$$

Next,

- $\displaystyle\int_{\mathcal{U}_{\boldsymbol{\theta}'}^{(1)}} f_{\boldsymbol{\theta}'}\,d\lambda = \frac{1}{2}\int_{\mathcal{U}_{\boldsymbol{\theta}'}^{(1)}} |x_d-b_0(\boldsymbol{x}^{(d)})|^{\widetilde{\gamma}-1}d\boldsymbol{x}$

$$= \frac{1}{2}\int_{\mathrm{supp}\,\varphi_{j^*}}\int_{|x_d-b_0(\boldsymbol{x}^{(d)})|\leq C_{\mathscr{C}}} |x_d-b_0(\boldsymbol{x}^{(d)})|^{\widetilde{\gamma}-1}dx_d\,d\boldsymbol{x}^{(d)}$$

$$= \frac{1}{\widetilde{\gamma}}C_{\mathscr{C}}^{\widetilde{\gamma}}\int_{\left\{\boldsymbol{z}\in[0,1]^{d-1}|M(\boldsymbol{z}-\boldsymbol{v}_{j^*}/M)\in\mathrm{supp}\,\varphi\right\}} 1\,d\boldsymbol{z}$$

$$= \frac{1}{\widetilde{\gamma}}C_{\mathscr{C}}^{\widetilde{\gamma}}M^{-(d-1)}\int_{\mathrm{supp}\,\varphi} 1\,d\boldsymbol{z}$$

$$\leq \frac{1}{\widetilde{\gamma}}C_{\mathscr{C}}^{\widetilde{\gamma}}M^{-(d-1)}. \tag{60}$$

Moreover,

- $\displaystyle\int_{\mathcal{U}_{\boldsymbol{\theta}}^{(2)}} f_{\boldsymbol{\theta}}\,d\lambda = \frac{1}{2}\int_{\mathcal{U}_{\boldsymbol{\theta}}^{(2)}} C_{\mathscr{C}}^{\widetilde{\gamma}-1}(1-\varphi_{j^*}(\boldsymbol{x}^{(d)}))^{\widetilde{\gamma}-1}d\boldsymbol{x}$

$$\leq \int_{[0,1]^{d-1}} C_{\mathscr{C}}^{\widetilde{\gamma}}(1-\varphi_{j^*}(\boldsymbol{x}^{(d)}))^{\widetilde{\gamma}}d\boldsymbol{x}^{(d)}$$

$$= C_{\mathscr{C}}^{\widetilde{\gamma}}M^{-(d-1)}\|1-\varphi\|_{L^{\widetilde{\gamma}}([0,1]^{d-1})}^{\widetilde{\gamma}}.$$

Finally,

- $\displaystyle\int_{\mathcal{U}_{\boldsymbol{\theta}'}^{(2)}} f_{\boldsymbol{\theta}'}\,d\lambda = \frac{1}{2}\int_{\mathcal{U}_{\boldsymbol{\theta}'}^{(2)}} C_{\mathscr{C}}^{\widetilde{\gamma}-1}d\boldsymbol{x}$

$$\leq C_{\mathscr{C}}^{\widetilde{\gamma}}M^{-(d-1)}.$$

Therefore, combining the above bounds on the four integrals with equation 58, we obtain

$$\frac{1}{2}\int_{\mathcal{X}}|f_{\boldsymbol{\theta}}-f_{\boldsymbol{\theta}'}|\,d\lambda \leq C_{\mathscr{C}}^{\widetilde{\gamma}}M^{-(d-1)}\left(\frac{1}{\widetilde{\gamma}}\|1+\varphi\|_{L^{\widetilde{\gamma}}([0,1]^{d-1})}^{\widetilde{\gamma}} + \|1-\varphi\|_{L^{\widetilde{\gamma}}([0,1]^{d-1})}^{\widetilde{\gamma}} + \frac{1}{\widetilde{\gamma}}+1\right)$$

$$\leq 2^{\widetilde{\gamma}+2}C_{\mathscr{C}}^{\widetilde{\gamma}}M^{-(d-1)}$$

and equation 35 holds.

Next, to prove equation 36 we see that equation 3 implies

$$G_{\boldsymbol{\theta},\boldsymbol{\theta}'} := \{\boldsymbol{x}\in\mathcal{X}\mid h_{\boldsymbol{\theta}}(\boldsymbol{x})\neq h_{\boldsymbol{\theta}'}(\boldsymbol{x})\}$$

$$= \{\boldsymbol{x}\in\mathcal{X}\mid h_{\boldsymbol{\theta}}(\boldsymbol{x})=0\wedge h_{\boldsymbol{\theta}'}(\boldsymbol{x})=1\}\cup\{\boldsymbol{x}\in\mathcal{X}\mid h_{\boldsymbol{\theta}}(\boldsymbol{x})=1\wedge h_{\boldsymbol{\theta}'}(\boldsymbol{x})=0\}$$

$$= \left\{\boldsymbol{x}\in\mathcal{X}\mid b_{\boldsymbol{\theta}'}(\boldsymbol{x}^{(d)})\leq x_d<b_{\boldsymbol{\theta}}(\boldsymbol{x}^{(d)})\right\}\cup\left\{\boldsymbol{x}\in\mathcal{X}\mid b_{\boldsymbol{\theta}}(\boldsymbol{x}^{(d)})\leq x_d<b_{\boldsymbol{\theta}'}(\boldsymbol{x}^{(d)})\right\}$$

$$= \left\{\boldsymbol{x}\in\mathcal{X}\mid \min\left\{b_{\boldsymbol{\theta}}(\boldsymbol{x}^{(d)}),b_{\boldsymbol{\theta}'}(\boldsymbol{x}^{(d)})\right\}\leq x_d<\max\left\{b_{\boldsymbol{\theta}}(\boldsymbol{x}^{(d)}),b_{\boldsymbol{\theta}'}(\boldsymbol{x}^{(d)})\right\}\right\}. \tag{61}$$

Using equation 45, we obtain $b_0(\boldsymbol{z}) \leq b_{\boldsymbol{\theta}''}(\boldsymbol{z}) \leq b_0(\boldsymbol{z}) + C_{\mathscr{C}}$ for all $\boldsymbol{z} \in [0,1]^{d-1}$ and all $\boldsymbol{\theta}'' \in \Theta$. Thus,

$$
\begin{aligned}
G_{\boldsymbol{\theta},\boldsymbol{\theta}'} &\subseteq \left\{ \boldsymbol{x} \in \mathcal{X} \mid b_0(\boldsymbol{x}^{(d)}) \leq x_d \leq b_0(\boldsymbol{x}^{(d)}) + C_{\mathscr{C}} \right\} \\
&\subset \left\{ \boldsymbol{x} \in \mathcal{X} \mid b_0(\boldsymbol{x}^{(d)}) - C_{\mathscr{C}} \leq x_d \leq b_0(\boldsymbol{x}^{(d)}) + C_{\mathscr{C}} + 2\boldsymbol{\theta}'' \cdot \boldsymbol{\varphi}(\boldsymbol{x}^{(d)}) \right\} \\
&= \mathcal{T}_{\boldsymbol{\theta}''} \quad \text{for all} \quad \boldsymbol{\theta}'' \in \Theta.
\end{aligned}
\tag{62}
$$

Moreover, equation 50 implies

$$
b_{\boldsymbol{\theta}}(\boldsymbol{z}) = b_{\boldsymbol{\theta}'}(\boldsymbol{z}) \quad \text{for all} \quad \boldsymbol{z} \in [0,1]^{d-1} \setminus \operatorname{supp} \varphi_{j^*},
$$

and therefore

$$
G_{\boldsymbol{\theta},\boldsymbol{\theta}'} \subseteq \mathcal{S}_{j^*} = \operatorname{supp} \varphi_{j^*} \times [0,1].
\tag{63}
$$

Then, by equation 62 and equation 63, we get

$$
G_{\boldsymbol{\theta},\boldsymbol{\theta}'} \subseteq \mathcal{T}_{\boldsymbol{\theta}''} \cap \mathcal{S}_{j^*} \quad \text{for all} \quad \boldsymbol{\theta}'' \in \Theta,
$$

and with the notation in equation 54, equation 57, we obtain

$$
\begin{aligned}
\int_{\{\boldsymbol{x} \in \mathcal{X} \mid h_{\boldsymbol{\theta}}(\boldsymbol{x}) \neq h_{\boldsymbol{\theta}'}(\boldsymbol{x})\}} (f_{\boldsymbol{\theta}} + f_{\boldsymbol{\theta}'}) d\lambda &= \int_{G_{\boldsymbol{\theta},\boldsymbol{\theta}'}} f_{\boldsymbol{\theta}} d\lambda + \int_{G_{\boldsymbol{\theta},\boldsymbol{\theta}'}} f_{\boldsymbol{\theta}'} d\lambda \\
&\leq \int_{\mathcal{T}_{\boldsymbol{\theta}} \cap \mathcal{S}_{j^*}} f_{\boldsymbol{\theta}} d\lambda + \int_{\mathcal{T}_{\boldsymbol{\theta}'} \cap \mathcal{S}_{j^*}} f_{\boldsymbol{\theta}'} d\lambda \\
&= \int_{\mathcal{U}_{\boldsymbol{\theta}}^{(1)}} f_{\boldsymbol{\theta}} d\lambda + \int_{\mathcal{U}_{\boldsymbol{\theta}'}^{(1)}} f_{\boldsymbol{\theta}'} d\lambda.
\end{aligned}
\tag{64}
$$

In conclusion, from equation 59, equation 60 and equation 64, we arrive at

$$
\begin{aligned}
\int_{\{\boldsymbol{x} \in \mathcal{X} \mid h_{\boldsymbol{\theta}}(\boldsymbol{x}) \neq h_{\boldsymbol{\theta}'}(\boldsymbol{x})\}} (f_{\boldsymbol{\theta}} + f_{\boldsymbol{\theta}'}) d\lambda &\leq C_{\mathscr{C}}^{\widetilde{\gamma}} M^{-(d-1)} \left( \frac{1}{\widetilde{\gamma}} \|1 + \varphi\|_{L^{\widetilde{\gamma}}([0,1]^{d-1})}^{\widetilde{\gamma}} + \frac{1}{\widetilde{\gamma}} \right) \\
&\leq 2^{\widetilde{\gamma}+1} C_{\mathscr{C}}^{\widetilde{\gamma}} M^{-(d-1)}.
\end{aligned}
$$

$\qquad\qquad\qquad\qquad\qquad\qquad\qquad\qquad\qquad\qquad\qquad\qquad\qquad\qquad\qquad\qquad\qquad\qquad\qquad$ $\square$

In what follows, we use the Hellinger distance to compare different probability measures. For densities $f$ and $g$ with respect to a measure $\nu$, the Hellinger distance is defined by

$$
\rho_{H,\nu}(f,g) = \left( \int_{\mathcal{X}} \left( \sqrt{f} - \sqrt{g} \right)^2 d\nu \right)^{1/2}.
\tag{65}
$$

The next lemma provides an upper bound on the Hellinger distance between product measures corresponding to parameters at Hamming distance one.

**Lemma 17.** *Let $\mathcal{D}$ be defined as*

$$
\mathcal{D} := \left\{ \boldsymbol{f}_{\boldsymbol{\theta}}^{\otimes n} \mid \boldsymbol{\theta} \in \Theta \right\}, \quad \text{where} \quad \boldsymbol{f}_{\boldsymbol{\theta}}(\boldsymbol{x},y) := \frac{f_{\boldsymbol{\theta}}(\boldsymbol{x}) \mathbb{1}_{\{y=h_{\boldsymbol{\theta}}(\boldsymbol{x})\}}}{\eta(\{y\})},
$$

*and $f_{\boldsymbol{\theta}}$ is as in equation 26. Then, $\mathcal{D}$ is a family of densities on $\Lambda^n$ with respect to $\boldsymbol{\lambda}^{\otimes n}$ (see equation 1), and for all $\boldsymbol{\theta}, \boldsymbol{\theta}' \in \Theta$ with $\rho_{Ham}(\boldsymbol{\theta}, \boldsymbol{\theta}') = 1$, we obtain*

$$
\rho_{H,\boldsymbol{\lambda}^{\otimes n}}^2 (\boldsymbol{f}_{\boldsymbol{\theta}}^{\otimes n}, \boldsymbol{f}_{\boldsymbol{\theta}'}^{\otimes n}) \leq 2^{\widetilde{\gamma}+4} n C_{\mathscr{C}}^{\widetilde{\gamma}} M^{-(d-1)}.
\tag{66}
$$

**Remark 18.** *In Lemma 17, we denote by $\boldsymbol{\mu}_{\boldsymbol{\theta}}$ the probability measure on $\Lambda$ having density $\boldsymbol{f}_{\boldsymbol{\theta}}$ with respect to $\boldsymbol{\lambda}$. Accordingly, $\boldsymbol{\mu}_{\boldsymbol{\theta}}$ has marginal $\mu_{\boldsymbol{\theta}}$ on $\mathcal{X}$, and $\boldsymbol{f}_{\boldsymbol{\theta}}^{\otimes n}$ is the density of the probability measure $\boldsymbol{\mu}_{\boldsymbol{\theta}}^{\otimes n}$ on $\Lambda^n$ with respect to $\boldsymbol{\lambda}^{\otimes n}$.*

*Proof.* Note that $\boldsymbol{f_\theta}$ is indeed a density function, since

$$
\begin{aligned}
\int_\Lambda \boldsymbol{f_\theta} d\boldsymbol{\lambda} &= \int_{\mathcal{X}\times\{0,1\}} \frac{f_\theta(\boldsymbol{x})\mathbb{1}_{\{y=h_\theta(\boldsymbol{x})\}}}{\eta(\{y\})} d\lambda(\boldsymbol{x})d\eta(y) \\
&= \int_{\mathcal{X}\times\{0\}} \frac{f_\theta(\boldsymbol{x})\mathbb{1}_{\{y=h_\theta(\boldsymbol{x})\}}}{\eta(\{y\})} d\lambda(\boldsymbol{x})d\eta(y) + \int_{\mathcal{X}\times\{1\}} \frac{f_\theta(\boldsymbol{x})\mathbb{1}_{\{y=h_\theta(\boldsymbol{x})\}}}{\eta(\{y\})} d\lambda(\boldsymbol{x})d\eta(y) \\
&= \int_\mathcal{X} f_\theta(\boldsymbol{x})\mathbb{1}_{\{h_\theta(\boldsymbol{x})=0\}}d\boldsymbol{x} + \int_\mathcal{X} f_\theta(\boldsymbol{x})\mathbb{1}_{\{h_\theta(\boldsymbol{x})=1\}}d\boldsymbol{x} \\
&= \int_\mathcal{X} f_\theta d\lambda \\
&= 1.
\end{aligned}
$$

Then, $\mathcal{D}$ is a family of densities on $\Lambda^n$ with respect to $\boldsymbol{\lambda}^{\otimes n}$. In addition (see (Tsybakov, 2009, Section 2.4)),

$$
\rho^2_{H,\boldsymbol{\lambda}^{\otimes n}}(\boldsymbol{f_\theta}^{\otimes n}, \boldsymbol{f_{\theta'}}^{\otimes n}) = 2\left(1 - \left(1 - \frac{\rho^2_{H,\boldsymbol{\lambda}}(\boldsymbol{f_\theta},\boldsymbol{f_{\theta'}})}{2}\right)^n\right) \leq n\rho^2_{H,\boldsymbol{\lambda}}(\boldsymbol{f_\theta},\boldsymbol{f_{\theta'}}), \tag{67}
$$

and

$$
\begin{aligned}
\rho^2_{H,\boldsymbol{\lambda}}(\boldsymbol{f_\theta},\boldsymbol{f_{\theta'}}) &= \int_\Lambda (\sqrt{\boldsymbol{f_\theta}} - \sqrt{\boldsymbol{f_{\theta'}}})^2 d\boldsymbol{\lambda} \\
&\leq \int_\Lambda |\boldsymbol{f_\theta} - \boldsymbol{f_{\theta'}}| d\boldsymbol{\lambda} \\
&= \int_\mathcal{X}\int_{\{0,1\}} \left| \frac{f_\theta(\boldsymbol{x})\mathbb{1}_{\{y=h_\theta(\boldsymbol{x})\}}}{\eta(\{y\})} - \frac{f_{\theta'}(\boldsymbol{x})\mathbb{1}_{\{y=h_{\theta'}(\boldsymbol{x})\}}}{\eta(\{y\})} \right| d\eta(y)d\lambda(\boldsymbol{x}) \\
&= \sum_{j\in\{0,1\}} \int_\mathcal{X} \left| f_\theta(\boldsymbol{x})\mathbb{1}_{\{h_\theta(\boldsymbol{x})=j\}} - f_{\theta'}(\boldsymbol{x})\mathbb{1}_{\{h_{\theta'}(\boldsymbol{x})=j\}} \right| d\boldsymbol{x} \\
&= \sum_{j\in\{0,1\}} \int_{\{\boldsymbol{x}\in\mathcal{X}|h_\theta(\boldsymbol{x})=h_{\theta'}(\boldsymbol{x})\}} |f_\theta(\boldsymbol{x}) - f_{\theta'}(\boldsymbol{x})| \, \mathbb{1}_{\{h_\theta(\boldsymbol{x})=j\}} d\boldsymbol{x} \\
&\quad + \sum_{j\in\{0,1\}} \int_{\{\boldsymbol{x}\in\mathcal{X}|h_\theta(\boldsymbol{x})\neq h_{\theta'}(\boldsymbol{x})\}} \left| f_\theta(\boldsymbol{x})\mathbb{1}_{\{h_\theta(\boldsymbol{x})=j\}} - f_{\theta'}(\boldsymbol{x})\mathbb{1}_{\{h_{\theta'}(\boldsymbol{x})=j\}} \right| d\boldsymbol{x} \\
&= \int_{\{\boldsymbol{x}\in\mathcal{X}|h_\theta(\boldsymbol{x})=h_{\theta'}(\boldsymbol{x})\}} |f_\theta - f_{\theta'}| d\lambda + \int_{\{\boldsymbol{x}\in\mathcal{X}|h_\theta(\boldsymbol{x})\neq h_{\theta'}(\boldsymbol{x})\}} (f_\theta + f_{\theta'})d\lambda.
\end{aligned}
$$

Therefore, by item 7) of Lemma 16 together with the above inequality, it follows that

$$
\begin{aligned}
\rho^2_{H,\boldsymbol{\lambda}}(\boldsymbol{f_\theta},\boldsymbol{f_{\theta'}}) &\leq \int_\mathcal{X} |f_\theta - f_{\theta'}| d\lambda + \int_{\{\boldsymbol{x}\in\mathcal{X}|h_\theta(\boldsymbol{x})\neq h_{\theta'}(\boldsymbol{x})\}} (f_\theta + f_{\theta'})d\lambda \\
&\leq 2^{\widetilde{\gamma}+3}C_\mathscr{C}^{\widetilde{\gamma}}M^{-(d-1)} + 2^{\widetilde{\gamma}+1}C_\mathscr{C}^{\widetilde{\gamma}}M^{-(d-1)} \\
&\leq 2^{\widetilde{\gamma}+4}C_\mathscr{C}^{\widetilde{\gamma}}M^{-(d-1)}. \tag{68}
\end{aligned}
$$

Finally, equation 67 and equation 68 imply

$$
\rho^2_{H,\boldsymbol{\lambda}^{\otimes n}}(\boldsymbol{f_\theta}^{\otimes n}, \boldsymbol{f_{\theta'}}^{\otimes n}) \leq n\rho^2_{H,\boldsymbol{\lambda}}(\boldsymbol{f_\theta},\boldsymbol{f_{\theta'}}) \leq 2^{\widetilde{\gamma}+4}nC_\mathscr{C}^{\widetilde{\gamma}}M^{-(d-1)}.
$$

$\square$

## B  Proof of Theorem 11

We know by equation 12 that $\varphi$ is Hölder continuous at $\boldsymbol{0}$ with exponent $\alpha \in (0,1]$ and constant $C_\varphi$, then there exists $r \in (0,1]$ such that

$$
\varphi(\boldsymbol{z}) \geq 1/2, \quad \text{for all} \quad \boldsymbol{z} \in \mathbb{R}^{d-1} \quad \text{satisfying} \quad \|\boldsymbol{z}\|_\infty \leq r. \tag{69}
$$

Indeed, we fix

$$r := \min\{1, (2C_\varphi)^{-1/\alpha}\}$$

and using equation 12, equation 14, we see that

$$1 - \varphi(\boldsymbol{z}) = |\varphi(\boldsymbol{z}) - \varphi(\boldsymbol{0})| \leq C_\varphi \|\boldsymbol{z}\|_\infty^\alpha \quad \text{for all} \quad \boldsymbol{z} \in \mathbb{R}^{d-1},$$

which implies

$$\varphi(\boldsymbol{z}) \geq 1 - C_\varphi \|\boldsymbol{z}\|_\infty^\alpha \geq 1 - \min\{C_\varphi, 1/2\} \geq 1/2 \quad \text{for all} \quad \boldsymbol{z} \in \mathbb{R}^{d-1} \quad \text{satisfying} \quad \|\boldsymbol{z}\|_\infty \leq r,$$

and equation 69 holds. Thus, by equation 15 and equation 69, we obtain

$$\varphi_j(\boldsymbol{z}) \geq 1/2, \quad \text{for all} \quad \boldsymbol{z} \in \mathbb{R}^{d-1} \quad \text{satisfying} \quad \|\boldsymbol{z} - \boldsymbol{v}_j/M\|_\infty \leq r/M, \tag{70}$$

and for each $j \in \{1, \dots, m\}$. We define the region (see Figure 4)

$$E_j = \left\{\boldsymbol{x} \in \mathcal{X} \mid \boldsymbol{x}^{(d)} \in N_j \quad \text{and} \quad C_\mathscr{C}\varphi_j(\boldsymbol{x}^{(d)})/2 \leq x_d - b_0(\boldsymbol{x}^{(d)}) \leq 3C_\mathscr{C}\varphi_j(\boldsymbol{x}^{(d)})/4\right\} \tag{71}$$

where (see equation 13)

$$N_j = \left\{\boldsymbol{z} \in [0,1]^{d-1} \mid \|\boldsymbol{z} - \boldsymbol{v}_j/M\|_\infty \leq r/M\right\} \subseteq Q_j \in P, \tag{72}$$

and equation 70, equation 71 and equation 72 imply

$$\begin{aligned}
\lambda(E_j) &= \int_{N_j} \int_{C_\mathscr{C}\varphi_j(\boldsymbol{x}^{(d)})/2 \leq x_d - b_0(\boldsymbol{x}^{(d)}) \leq 3C_\mathscr{C}\varphi_j(\boldsymbol{x}^{(d)})/4} 1 dx_d d\boldsymbol{x}^{(d)} \\
&= \int_{N_j} C_\mathscr{C}\varphi_j(\boldsymbol{x}^{(d)})/4 d\boldsymbol{x}^{(d)} \\
&\geq \frac{1}{8}C_\mathscr{C} \int_{\|\boldsymbol{z} - \boldsymbol{v}_j/M\|_\infty \leq r/M} 1 d\boldsymbol{z} \\
&= \frac{1}{8}(2r)^{d-1}C_\mathscr{C}M^{-(d-1)}.
\end{aligned} \tag{73}$$

For all $\boldsymbol{\theta} \in \Theta$, from equation 24, equation 48 and equation 71, we see that

$$E_j \subset \mathcal{T}_{\boldsymbol{\theta}} = \mathcal{R} \cap \mathcal{T}_{\boldsymbol{\theta}} \quad \text{and} \quad h_{\boldsymbol{\theta}}(\boldsymbol{x}) = 1 - \theta_j \quad \text{for all} \quad \boldsymbol{x} \in E_j. \tag{74}$$

Using equation 30, equation 71, equation 72 and equation 74, we get

$$\begin{aligned}
E_j &= \left\{\boldsymbol{x} \in \mathcal{X} \mid \boldsymbol{x}^{(d)} \in N_j \quad \text{and} \quad (1/2 - \theta_j)C_\mathscr{C}\varphi_j(\boldsymbol{x}^{(d)}) \leq x_d - b_{\boldsymbol{\theta}}(\boldsymbol{x}^{(d)}) \leq (3/4 - \theta_j)C_\mathscr{C}\varphi_j(\boldsymbol{x}^{(d)})\right\} \\
&\subseteq L_{E_j} := \left\{\boldsymbol{x} \in \mathcal{X} \mid \boldsymbol{x}^{(d)} \in N_j \quad \text{and} \quad |x_d - b_{\boldsymbol{\theta}}(\boldsymbol{x}^{(d)})| \geq C_\mathscr{C}\varphi_j(\boldsymbol{x}^{(d)})/4\right\},
\end{aligned}$$

and

$$f_{\boldsymbol{\theta}}\mathbb{1}_{E_j}(\boldsymbol{x}) = f_{\boldsymbol{\theta}}\mathbb{1}_{E_j \cap L_{E_j}}(\boldsymbol{x}) = \frac{1}{2}|x_d - b_{\boldsymbol{\theta}}(\boldsymbol{x}^{(d)})|^{\widetilde{\gamma}-1}\mathbb{1}_{E_j \cap L_{E_j}}(\boldsymbol{x}) \geq \frac{1}{2}(C_\mathscr{C}\varphi_j(\boldsymbol{x}^{(d)})/4)^{\widetilde{\gamma}-1}. \tag{75}$$

By definition in equation 2, for all $A \in \mathcal{A}_n(L^2(\lambda))$ and for all $\boldsymbol{S}_n \in \Lambda^n$, we have $A(\boldsymbol{S}_n) \in L^2(\lambda)$. Since $f_{\boldsymbol{\theta}}$ in equation 26 is upper bounded, then $A(\boldsymbol{S}_n) \in L^2(\mu_{\boldsymbol{\theta}})$. Therefore, by equation 70, equation 71, equation 72, equation 74 and equation 75, we obtain

$$\begin{aligned}
\|A(\boldsymbol{S}_n) - h_{\boldsymbol{\theta}}\|_{L^2(\mu_{\boldsymbol{\theta}})}^2 &= \int_{\mathcal{X}} |A(\boldsymbol{S}_n) - h_{\boldsymbol{\theta}}|^2 f_{\boldsymbol{\theta}} d\lambda \\
&\geq \sum_{j=1}^m \int_{E_j} |A(\boldsymbol{S}_n) - h_{\boldsymbol{\theta}}|^2 f_{\boldsymbol{\theta}} d\lambda \\
&\geq \frac{1}{2}(C_\mathscr{C}/4)^{\widetilde{\gamma}-1} \sum_{j=1}^m \int_{E_j} |A(\boldsymbol{S}_n)(\boldsymbol{x}) - t(\theta_j)|^2 (\varphi_j(\boldsymbol{x}^{(d)}))^{\widetilde{\gamma}-1} d\boldsymbol{x} \\
&\geq \frac{1}{2}(C_\mathscr{C}/8)^{\widetilde{\gamma}-1} \sum_{j=1}^m \int_{E_j} |A(\boldsymbol{S}_n) - t(\theta_j)|^2 d\lambda \quad \text{where} \quad t(\theta_j) := 1 - \theta_j.
\end{aligned} \tag{76}$$

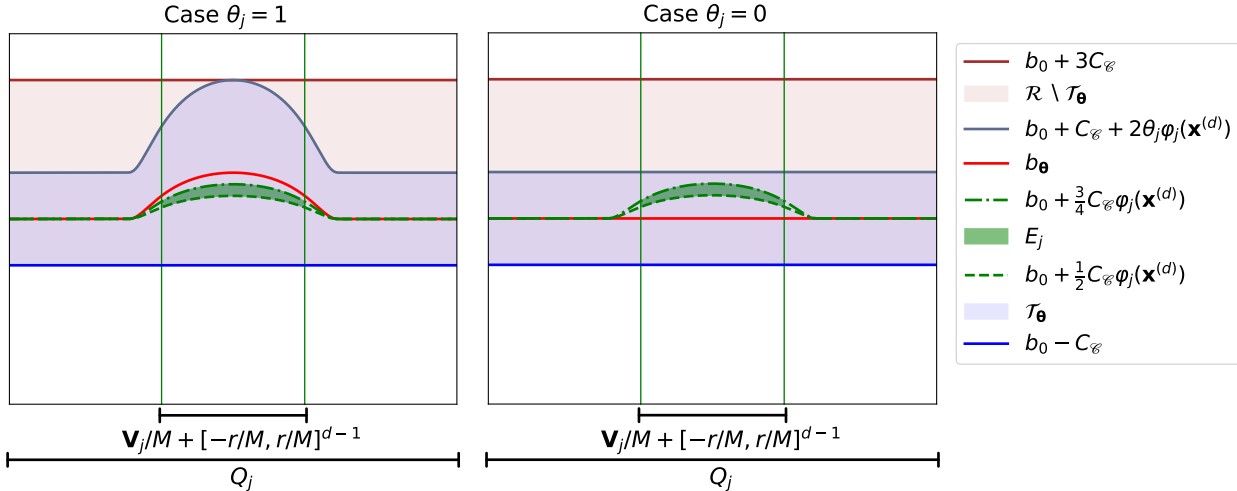

Figure 4: Illustration of $E_j$ for $d = 2$, where $\varphi$ is a bump function and $b_0 := 1/2$. In the image on the left, the bump is active and the function $b_{\boldsymbol{\theta}}$ lies above the region $E_j$, which implies by equation 3 that $h_{\boldsymbol{\theta}}(\boldsymbol{x}) = 0$ for all $\boldsymbol{x} \in E_j$. In contrast, in the image on the right, the bump is inactive and the function $b_{\boldsymbol{\theta}}$ lies below $E_j$; therefore $h_{\boldsymbol{\theta}}(\boldsymbol{x}) = 1$, for all $\boldsymbol{x} \in E_j$. In both cases the region $E_j$ is sufficiently far from the decision boundary, and therefore $f_{\boldsymbol{\theta}}$ can be bounded from below on $E_j$, as in equation 75. Consequently, $E_j$ is a suitable region for projecting the estimator $A(\boldsymbol{S}_n)$ onto the discrete set $\Theta$, as in equation 77. Moreover, by taking different parameters $d$, $b_0$, and $\varphi$, the same conclusions about $E_j$ still hold. For visualization purposes, we use a particular choice of parameters. The same idea applies for arbitrary choices of $d$, $b_0$, and $\varphi$ satisfying the hypotheses of Theorem 11.

For each $A \in \mathcal{A}_n(L^2(\lambda))$, we define the projection $\widehat{\boldsymbol{\theta}}_A : \Lambda^n \to \Theta$ of $A(\boldsymbol{S}_n)$ onto $\Theta$, such that

$$\widehat{\boldsymbol{\theta}}_A(\boldsymbol{S}_n) := (\widehat{\theta}_{1\,A}(\boldsymbol{S}_n), \ldots, \widehat{\theta}_{m\,A}(\boldsymbol{S}_n)) \quad \text{where} \quad \widehat{\theta}_{j\,A}(\boldsymbol{S}_n) \in \arg\min_{\theta \in \{0,1\}} \|A(\boldsymbol{S}_n) - t(\theta)\|_{L^2(\lambda|_{E_j})}. \tag{77}$$

Here we used the notation $\|\cdot\|^2_{L^2(\lambda|_{E_j})} := \int_{E_j} |\cdot|^2 d\lambda$ and $t(\theta) := 1 - \theta$. If both values $\theta = 0$ and $\theta = 1$ attain the minimum in equation 77, we fix $\widehat{\theta}_{j\,A}(\boldsymbol{S}_n) := 0$. Then,

$$\begin{aligned}
|\widehat{\theta}_{j\,A}(\boldsymbol{S}_n) - \theta_j|\sqrt{\lambda(E_j)} &= \left\|\widehat{\theta}_{j\,A}(\boldsymbol{S}_n) - \theta_j\right\|_{L^2(\lambda|_{E_j})} \\
&= \left\|t(\widehat{\theta}_{j\,A}(\boldsymbol{S}_n)) - t(\theta_j)\right\|_{L^2(\lambda|_{E_j})} \\
&\le \left\|A(\boldsymbol{S}_n) - t(\widehat{\theta}_{j\,A}(\boldsymbol{S}_n))\right\|_{L^2(\lambda|_{E_j})} + \|A(\boldsymbol{S}_n) - t(\theta_j)\|_{L^2(\lambda|_{E_j})} \\
&\le 2\|A(\boldsymbol{S}_n) - t(\theta_j)\|_{L^2(\lambda|_{E_j})}. 
\end{aligned} \tag{78}$$

So, we use equation 76 and equation 78, to obtain

$$\begin{aligned}
\|A(\boldsymbol{S}_n) - h_{\boldsymbol{\theta}}\|^2_{L^2(\mu_{\boldsymbol{\theta}})} &\ge \frac{1}{2}(C_{\mathscr{C}}/8)^{\widetilde{\gamma}-1} \sum_{j=1}^m \|A(\boldsymbol{S}_n) - t(\theta_j)\|^2_{L^2(\lambda|_{E_j})} \\
&\ge \frac{1}{8}(C_{\mathscr{C}}/8)^{\widetilde{\gamma}-1} \sum_{j=1}^m |\widehat{\theta}_{j\,A}(\boldsymbol{S}_n) - \theta_j|\lambda(E_j),
\end{aligned}$$

and by equation 27, equation 73, it follows that

$$\|A(\boldsymbol{S}_n) - h_{\boldsymbol{\theta}}\|^2_{L^2(\mu_{\boldsymbol{\theta}})} \geq \frac{1}{8}(C_{\mathscr{C}}/8)^{\widetilde{\gamma}-1} \sum_{j \in J(\widehat{\boldsymbol{\theta}}_A(\boldsymbol{S}_n), \boldsymbol{\theta})} \lambda(E_j)$$

$$\geq \frac{(2r)^{d-1}}{8^{\widetilde{\gamma}+1}} C^{\widetilde{\gamma}}_{\mathscr{C}} M^{-(d-1)} \sum_{j \in J(\widehat{\boldsymbol{\theta}}_A(\boldsymbol{S}_n), \boldsymbol{\theta})} 1$$

$$= \left(\frac{(2r)^{d-1}}{8^{\widetilde{\gamma}+1}}\right) C^{\widetilde{\gamma}}_{\mathscr{C}} M^{-(d-1)} \rho_{Ham}(\widehat{\boldsymbol{\theta}}_A(\boldsymbol{S}_n), \boldsymbol{\theta}).$$

Taking the expectation $\mathbb{E}_{\boldsymbol{\theta}} := \mathbb{E}_{\{\boldsymbol{x}_i\}^n_{i=1} \overset{iid}{\sim} \mu_{\boldsymbol{\theta}}}$ on both sides of the last inequality, we arrive at

$$\mathbb{E}_{\boldsymbol{\theta}} \|A(\boldsymbol{S}_n) - h_{\boldsymbol{\theta}}\|^2_{L^2(\mu_{\boldsymbol{\theta}})} \geq \left(\frac{(2r)^{d-1}}{8^{\widetilde{\gamma}+1}}\right) C^{\widetilde{\gamma}}_{\mathscr{C}} M^{-(d-1)} \mathbb{E}_{\boldsymbol{\theta}} \rho_{Ham}(\widehat{\boldsymbol{\theta}}_A(\boldsymbol{S}_n), \boldsymbol{\theta}). \tag{79}$$

In conclusion, equation 79 together with the notation in equation 9, imply

$$\sup_{(h,\mu) \in \mathcal{P}_{\mathscr{C}}(\mathcal{L}_\lambda)} \mathbb{E}_{\{\boldsymbol{x}_i\}^n_{i=1} \overset{iid}{\sim} \mu} \|A(\boldsymbol{S}_n) - h\|^2_{L^2(\mu)} \geq \max_{(h,\mu) \in \mathcal{P}_{\mathscr{C}_\Theta}(\mathcal{M}_\Theta)} \mathbb{E}_{\{\boldsymbol{x}_i\}^n_{i=1} \overset{iid}{\sim} \mu} \|A(\boldsymbol{S}_n) - h\|^2_{L^2(\mu)} \tag{80}$$

$$\geq \max_{\boldsymbol{\theta} \in \Theta} \mathbb{E}_{\boldsymbol{\theta}} \|A(\boldsymbol{S}_n) - h_{\boldsymbol{\theta}}\|^2_{L^2(\mu_{\boldsymbol{\theta}})}$$

$$\geq \left(\frac{(2r)^{d-1}}{8^{\widetilde{\gamma}+1}}\right) C^{\widetilde{\gamma}}_{\mathscr{C}} M^{-(d-1)} \max_{\boldsymbol{\theta} \in \Theta} \mathbb{E}_{\boldsymbol{\theta}} \rho_{Ham}(\widehat{\boldsymbol{\theta}}_A(\boldsymbol{S}_n), \boldsymbol{\theta}), \tag{81}$$

since $\mathscr{C}_\Theta \subseteq \mathscr{C}$ (by equation 19), and

$$\mathcal{P}_{\mathscr{C}}(\mathcal{L}_\lambda) \supseteq \mathcal{P}_{\mathscr{C}_\Theta}(\mathcal{M}_\Theta) \supseteq \{(h,\mu) \in H_{\mathscr{C}_\Theta} \times \mathcal{M}_\Theta \mid (h,\mu) = (h_{\boldsymbol{\theta}}, \mu_{\boldsymbol{\theta}})\}.$$

We state Assouad's lemma as in (Tsybakov, 2009, Theorem 2.12 (Hellinger version)) to relate it to the lower bound in equation 81.

**Lemma 19.** *Let $\mathcal{D} := \{\boldsymbol{P}_{\boldsymbol{\theta}} \mid \boldsymbol{\theta} \in \Theta\}$ be a set of $2^m$ probability densities on $\Lambda^n$ with respect to $\boldsymbol{\lambda}^{\otimes n}$, where $\boldsymbol{\lambda}$ is defined in equation 1. If*

$$\rho^2_{H,\boldsymbol{\lambda}^{\otimes n}}(\boldsymbol{P}_{\boldsymbol{\theta}}, \boldsymbol{P}_{\boldsymbol{\theta}'}) \leq \vartheta < 2 \quad \text{for all} \quad \boldsymbol{\theta}, \boldsymbol{\theta}' \in \Theta, \quad \text{such that} \quad \rho_{Ham}(\boldsymbol{\theta}, \boldsymbol{\theta}') = 1,$$

*where $\rho^2_{H,\cdot}$ denotes the Hellinger distance defined in equation 65. Then*

$$\inf_{\widehat{\boldsymbol{\theta}}:\Lambda^n \to \Theta} \max_{\boldsymbol{\theta} \in \Theta} \mathbb{E}_{\boldsymbol{\theta}} \rho_{Ham}(\widehat{\boldsymbol{\theta}}(\boldsymbol{S}_n), \boldsymbol{\theta}) \geq \frac{m}{2}\left(1 - \sqrt{\vartheta(1 - \vartheta/4)}\right), \tag{82}$$

*where $\mathbb{E}_{\boldsymbol{\theta}} := \mathbb{E}_{\boldsymbol{S}_n \sim \nu_{\boldsymbol{P}_{\boldsymbol{\theta}}}}$ and $\boldsymbol{\nu}_{\boldsymbol{P}_{\boldsymbol{\theta}}}$ is the probability measure induced by $\boldsymbol{P}_{\boldsymbol{\theta}}$.*

To apply Lemma 19, we use the family of densities

$$\mathcal{D} := \left\{\boldsymbol{f}^{\otimes n}_{\boldsymbol{\theta}} \mid \boldsymbol{\theta} \in \Theta\right\}$$

defined in Lemma 17. By equation 19 and equation 66, for all $\boldsymbol{\theta}, \boldsymbol{\theta}' \in \Theta$ with $\rho_{Ham}(\boldsymbol{\theta}, \boldsymbol{\theta}') = 1$, we have

$$\rho^2_{H,\boldsymbol{\lambda}^{\otimes n}}(\boldsymbol{f}^{\otimes n}_{\boldsymbol{\theta}}, \boldsymbol{f}^{\otimes n}_{\boldsymbol{\theta}'}) \leq 2^{\widetilde{\gamma}+4} n C^{\widetilde{\gamma}}_{\mathscr{C}} M^{-(d-1)} \leq 1/4 < 2.$$

Therefore, the hypotheses of Lemma 19 are satisfied with $\vartheta := 2^{\widetilde{\gamma}+4} n C^{\widetilde{\gamma}}_{\mathscr{C}} M^{-(d-1)}$ and $m = (M/2)^{d-1}$ (see Construction 10). In conclusion, since $\vartheta \leq 1/4$, equation 82 implies

$$\inf_{\widehat{\boldsymbol{\theta}}:\Lambda^n \to \Theta} \max_{\boldsymbol{\theta} \in \Theta} \mathbb{E}_{\boldsymbol{\theta}} \rho_{Ham}(\widehat{\boldsymbol{\theta}}(\boldsymbol{S}_n), \boldsymbol{\theta}) \geq 2^{-d} M^{d-1} \left(1 - \sqrt{\vartheta(1 - \vartheta/4)}\right)$$

$$\geq 2^{-d-1} M^{d-1}, \tag{83}$$

where from the notation in equation 10 and Remark 18, we know that

$$\mathbb{E}_{\boldsymbol{\theta}} := \mathbb{E}_{\boldsymbol{S}_n \sim \boldsymbol{\mu}_{\boldsymbol{\theta}}^{\otimes n}} := \mathbb{E}_{\{\boldsymbol{x}_i\}_{i=1}^n \overset{iid}{\sim} \mu_{\boldsymbol{\theta}}}.$$

Since $A$ in equation 81 is an arbitrary element of $\mathcal{A}_n(L^2(\lambda))$, then equation 81 combined with equation 83 imply

$$
\begin{aligned}
\sup_{(h,\mu)\in\mathcal{P}_{\mathscr{C}}(\mathcal{L}_\lambda)} \mathbb{E}_{\{\boldsymbol{x}_i\}_{i=1}^n \overset{iid}{\sim} \mu} \|A(\boldsymbol{S}_n) - h\|_{L^2(\mu)}^2 &\geq \left(\frac{(2r)^{d-1}}{8^{\widetilde{\gamma}+1}}\right) C_{\mathscr{C}}^{\widetilde{\gamma}} M^{-(d-1)} \max_{\boldsymbol{\theta}\in\Theta} \mathbb{E}_{\boldsymbol{\theta}} \rho_{Ham}(\widehat{\boldsymbol{\theta}}_A(\boldsymbol{S}_n), \boldsymbol{\theta}) \\
&\geq \left(\frac{(2r)^{d-1}}{8^{\widetilde{\gamma}+1}}\right) C_{\mathscr{C}}^{\widetilde{\gamma}} M^{-(d-1)} \inf_{\widehat{\boldsymbol{\theta}}:\Lambda^n\to\Theta} \max_{\boldsymbol{\theta}\in\Theta} \mathbb{E}_{\boldsymbol{\theta}} \rho_{Ham}(\widehat{\boldsymbol{\theta}}(\boldsymbol{S}_n), \boldsymbol{\theta}) \\
&\geq \left(\frac{r^{d-1}}{8^{\widetilde{\gamma}+2}}\right) C_{\mathscr{C}}^{\widetilde{\gamma}}.
\end{aligned}
$$

Finally, by taking infimums in the previous inequality over all $A \in \mathcal{A}_n(L^2(\lambda))$, we conclude

$$\mathcal{I}_n(\mathscr{C}) \geq \left(\frac{r^{d-1}}{8^{\widetilde{\gamma}+2}}\right) C_{\mathscr{C}}^{\widetilde{\gamma}},$$

and equation 20 is fulfilled. $\qquad\square$

## C  Proof of Corollary 13

### C.1  Hölder space

Let $\psi \in C_c^\infty(\mathbb{R}; \mathbb{R})$ be any 1-dimensional bump function, with $\psi(x) = 0$ for all $|x| \geq 1$, $0 \leq \psi(x) \leq 1$ for all $x \in \mathbb{R}$, and $\psi(0) = 1$. For Construction 10, we set $\varphi : \mathbb{R}^{d-1} \to \mathbb{R}$ as

$$\varphi(\boldsymbol{z}) := \psi(2\|\boldsymbol{z}\|_\infty). \tag{84}$$

By definition above, $\varphi(\boldsymbol{0}) = 1$, $0 \leq \varphi(\boldsymbol{z}) \leq 1$ for all $\boldsymbol{z} \in \mathbb{R}^{d-1}$ and

$$
\begin{aligned}
\operatorname{supp} \varphi &= \left\{\boldsymbol{z} \in \mathbb{R}^{d-1} \mid 2\|\boldsymbol{z}\|_\infty \in \operatorname{supp} \psi \subset (-1,1)\right\} \\
&\subseteq \left\{\boldsymbol{z} \in \mathbb{R}^{d-1} \mid \|\boldsymbol{z}\|_\infty \leq 1/2\right\} \\
&\subset (-1,1)^{d-1},
\end{aligned}
\tag{85}
$$

therefore equation 14 holds. By the mean value theorem, since $\psi \in C_c^\infty(\mathbb{R}; \mathbb{R})$, we have

$$
\begin{aligned}
|\varphi(\boldsymbol{z}) - \varphi(\boldsymbol{w})| &= |\psi(2\|\boldsymbol{z}\|_\infty) - \psi(2\|\boldsymbol{w}\|_\infty)| \\
&\leq 2\|\psi'\|_\infty |\|\boldsymbol{z}\|_\infty - \|\boldsymbol{w}\|_\infty| \\
&\leq 2\|\psi'\|_\infty \|\boldsymbol{z} - \boldsymbol{w}\|_\infty \quad \text{for all} \quad \boldsymbol{z}, \boldsymbol{w} \in \mathbb{R}^{d-1},
\end{aligned}
$$

where $\|\psi'\|_\infty = \max_{x\in\mathbb{R}} |\psi'(x)|$ and $\psi'$ denotes the first derivative of $\psi$. Thus, for $\alpha \in (0,1]$, and for all $\boldsymbol{z}, \boldsymbol{w} \in \mathbb{R}^{d-1}$, we see that

- If $\|\boldsymbol{z} - \boldsymbol{w}\|_\infty \leq 1$,

$$|\varphi(\boldsymbol{z}) - \varphi(\boldsymbol{w})| \leq 2\|\psi'\|_\infty \|\boldsymbol{z} - \boldsymbol{w}\|_\infty \leq 2\|\psi'\|_\infty \|\boldsymbol{z} - \boldsymbol{w}\|_\infty^\alpha. \tag{86}$$

- If $\|\boldsymbol{z} - \boldsymbol{w}\|_\infty > 1$, since $0 \leq \varphi(\boldsymbol{v}) \leq 1$ for all $\boldsymbol{v} \in \mathbb{R}^{d-1}$, then

$$|\varphi(\boldsymbol{z}) - \varphi(\boldsymbol{w})| \leq 1 \leq \|\boldsymbol{z} - \boldsymbol{w}\|_\infty^\alpha. \tag{87}$$

Therefore, equation 86 and equation 87 imply that

$$|\varphi(\boldsymbol{z}) - \varphi(\boldsymbol{w})| \le K_\varphi \|\boldsymbol{z} - \boldsymbol{w}\|_\infty^\alpha \tag{88}$$

$$\le K_\varphi \|\boldsymbol{z} - \boldsymbol{w}\|_2^\alpha \quad \text{for all} \quad \boldsymbol{z}, \boldsymbol{w} \in \mathbb{R}^{d-1}, \tag{89}$$

where $K_\varphi := \max\{1, 2\|\psi'\|_\infty\}$. So, $\varphi$ is a Hölder continuous function with exponent $\alpha \in (0, 1]$ and constant $K_\varphi$ with respect to the Euclidean norm. Moreover, equation 88 implies that $\varphi$ is Hölder continuous at $\boldsymbol{0}$ with exponent $\alpha$ and constant $C_\varphi := K_\varphi$ with respect to the $\ell_\infty$-norm (see equation 12).

Up to this point, all the properties required for $\varphi$ in Construction 10 have been verified. It remains to choose $C_{\mathscr{C}}$ in equation 16 such that $\mathscr{C}_\Theta \subseteq \mathscr{C}$, as required by Theorem 11 in equation 19. The remaining conditions on $C_{\mathscr{C}}$ will then be checked, while all other elements of the construction (the parameters $M$, $m$, the partition $P$, etc) are taken exactly as in Construction 10.

In Section 1.3, we defined the class of Hölder continuous functions $\mathcal{H}_\alpha$ as the class $\mathscr{C}$ satisfying Condition C2). Then, by item 3) in Lemma 16, we know that

$$\mathscr{C} := \mathcal{H}_\alpha \supseteq \mathscr{C}_\Theta.$$

Furthermore, the only important assumption in the proof of item 3) in Lemma 16, apart from the hypotheses on $\varphi$ and the fact that $\mathscr{C}$ satisfies Condition C2), is that $C_{\mathscr{C}} \le M^{-\alpha}/4$ in equation 18. Hence, by choosing

$$C_{\mathscr{C}} := M^{-\alpha}/4, \tag{90}$$

we conclude that $\mathscr{C}_\Theta \subseteq \mathscr{C}$.

Next, we still need to ensure that the baseline function $b_0 \in \mathscr{C}$ satisfies

$$b_0(\boldsymbol{z}) \in [C_{\mathscr{C}}, 1 - 3C_{\mathscr{C}}] \quad \text{for all} \quad \boldsymbol{z} \in [0, 1]^{d-1}$$

in equation 18. This can be achieved either by assuming that $b_0 \in \mathcal{H}_\alpha$ satisfies this condition, or by fixing a specific choice, for instance $b_0 \equiv 1/2$ or $b_0$ as in Figure 2, which clearly belongs to $\mathcal{H}_\alpha$ and satisfies the above condition.

Finally, by equation 90 and in order to satisfy the right-hand side in equation 19, it suffices to choose $M$ such that

$$1 \ge 2^{\widetilde{\gamma}+6} n C_{\mathscr{C}}^{\widetilde{\gamma}} M^{-(d-1)}$$

$$= 2^{\widetilde{\gamma}+6} n (M^{-\alpha}/4)^{\widetilde{\gamma}} M^{-(d-1)}$$

$$= 2^{-\widetilde{\gamma}+6} n M^{-(d-1)-\gamma}.$$

Therefore, we choose $M$ as the smallest even integer such that

$$M \ge 2^{\frac{-\widetilde{\gamma}+6}{\gamma+(d-1)}} n^{\frac{1}{\gamma+(d-1)}} := M_*, \quad \text{that is} \quad M := 2\lceil M_*/2 \rceil, \tag{91}$$

where $\lceil . \rceil$ denotes the ceiling function for integers. In conclusion, we have shown that all the conditions of Theorem 11 are satisfied. So, equation 20, equation 90 and equation 91 imply

$$\mathcal{I}_n(\mathscr{C}) \ge \left(\frac{r^{d-1}}{8^{\widetilde{\gamma}+2}}\right) C_{\mathscr{C}}^{\widetilde{\gamma}} = \left(\frac{r^{d-1}}{8^{\widetilde{\gamma}+2}}\right) (M^{-\alpha}/4)^{\widetilde{\gamma}} = \left(\frac{4^{-\widetilde{\gamma}} r^{d-1}}{8^{\widetilde{\gamma}+2}}\right) (2\lceil M_*/2 \rceil)^{-\gamma} \quad \text{for all} \quad n \in \mathbb{N}, \tag{92}$$

where $r = \min\{1, (2C_\varphi)^{-1/\alpha}\}$ and $C_\varphi = K_\varphi = \max\{1, 2\|\psi'\|_\infty\}$. Note that

$$\lceil M_*/2 \rceil \le M_*/2 + 1 \quad \text{and} \quad M_* \ge 2^{-\frac{\widetilde{\gamma}}{\gamma+(d-1)}} = \frac{1}{C_* - 1/2}, \quad \text{with} \quad C_* := 2^{\frac{\widetilde{\gamma}}{\gamma+(d-1)}} + 1/2.$$

Thus, $\lceil M_*/2 \rceil \le M_*/2 + 1 \le C_* M_*$ and

$$(2\lceil M_*/2 \rceil)^{-\gamma} \ge (2C_* M_*)^{-\gamma}. \tag{93}$$

In addition, since $C_\varphi = \max\{1, 2\,\|\psi'\|_\infty\} \geq 1$, we have

$$r = \min\{1, (2C_\varphi)^{-1/\alpha}\} = (2C_\varphi)^{-1/\alpha}. \tag{94}$$

Then, from equation 92, equation 93 and equation 94, it follows that

$$\mathcal{I}_n(\mathscr{C}) \geq \left(\frac{4^{-\widetilde{\gamma}} r^{d-1}}{8^{\widetilde{\gamma}+2}}\right)(2C_* M_*)^{-\gamma} = C^* n^{-\frac{\gamma}{\gamma+(d-1)}} \quad \text{for all} \quad n \in \mathbb{N}, \tag{95}$$

where

$$
\begin{aligned}
C^* &:= \left(\frac{4^{-\widetilde{\gamma}}\left((2C_\varphi)^{-1/\alpha}\right)^{d-1}}{8^{\widetilde{\gamma}+2}}\right)\left(\left(2^{\frac{\widetilde{\gamma}}{\gamma+(d-1)}+1}+1\right)\left(2^{\frac{-\widetilde{\gamma}+6}{\gamma+(d-1)}}\right)\right)^{-\gamma} \\
&\geq C_\varphi^{-\frac{d-1}{\alpha}}\left(2^{-2\widetilde{\gamma}}\left(2^{-1/\alpha}\right)^{d-1} 2^{-3\widetilde{\gamma}-6}\right)\left(\left(2+2^{\frac{-\widetilde{\gamma}}{\gamma+(d-1)}}\right)\left(2^{\frac{6}{\gamma+(d-1)}}\right)\right)^{-\gamma} \\
&\geq C_\varphi^{-\frac{d-1}{\alpha}}\left(2^{-\frac{d-1}{\alpha}} 2^{-5\widetilde{\gamma}-6}\right)\left(2^{-2\gamma}\right) 2^{-6\gamma} \\
&\geq 2^{-\left(\frac{d-1}{\alpha}+13\widetilde{\gamma}+6\right)} C_\varphi^{-\frac{d-1}{\alpha}}. \tag{96}
\end{aligned}
$$

Here, the dependence on $C_\varphi$ can be removed by fixing a specific bump function $\psi$; for instance, the standard bump

$$\psi(x) = \begin{cases} e^{1-\frac{1}{1-x^2}} & \text{if } |x| < 1 \\ 0 & \text{if } |x| \geq 1. \end{cases} \tag{97}$$

Indeed,

$$|\psi'(x)| = \begin{cases} 2|x|(1-x^2)^{-2}\psi(x) & \text{if } |x| < 1 \\ 0 & \text{if } |x| \geq 1, \end{cases}$$

and

$$\psi(x) = e^{-\frac{1}{1-x^2}} e \leq (1-x^2)^2 e \quad \text{for all} \quad x \in (-1,1).$$

Therefore,

$$|\psi'(x)| \leq \begin{cases} 2|x|e & \text{if } |x| < 1 \\ 0 & \text{if } |x| \geq 1, \end{cases} \leq 2e \quad \text{for all} \quad x \in \mathbb{R}, \quad \text{which implies} \quad \|\psi'\|_\infty \leq 2e.$$

Thus,

$$C_\varphi = \max\{1, 2\,\|\psi'\|_\infty\} \leq \max\{1, 4e\} = 4e < 2^4. \tag{98}$$

Finally, from equation 95, equation 96 and equation 98, we arrive at

$$\mathcal{I}_n(\mathscr{C}) \geq 2^{-\left(\frac{5(d-1)}{\alpha}+13\widetilde{\gamma}+6\right)} n^{-\frac{\gamma}{\gamma+(d-1)}} \quad \text{for all} \quad n \in \mathbb{N},$$

and therefore equation 21 holds. $\qquad\square$

## C.2  Barron space

Let $C > 0$ be a constant. By Remark 6, every Barron function $b \in \mathscr{C} := \mathcal{B}_C$ is Lipschitz on $[0,1]^{d-1}$. So, in Construction 10, the parameter corresponding to Condition C2) is $\alpha = 1$, which in turn implies that $\widetilde{\gamma} = \gamma$.

Similarly to the Hölder case in the proof of item i), we consider a baseline 1-dimensional bump function $\psi \in C_c^\infty(\mathbb{R};\mathbb{R})$ satisfying $0 \leq \psi(x) \leq 1$ for all $x \in \mathbb{R}$, $\operatorname{supp}\psi \subset (-1,1)$, and $\psi(x) = 1$ for all $x \in [-1/2, 1/2]$. We define $\varphi : \mathbb{R}^{d-1} \to \mathbb{R}$ as

$$\varphi(\boldsymbol{z}) = \prod_{j=1}^{d-1} \psi(z_j) \quad \text{for all} \quad \boldsymbol{z} = (z_1, \ldots, z_{d-1}) \in \mathbb{R}^{d-1}. \tag{99}$$

Therefore, $0 \leq \varphi(\boldsymbol{z}) \leq 1$ for all $\boldsymbol{z} \in \mathbb{R}^{d-1}$, $\varphi(\boldsymbol{z}) = 1$ for all $\boldsymbol{z} \in [-1/2, 1/2]^{d-1}$, and

$$\operatorname{supp} \varphi = \left\{ \boldsymbol{z} \in \mathbb{R}^{d-1} \mid z_j \in \operatorname{supp} \psi \ \text{ for all } \ j \in \{1, \ldots, d-1\} \right\}$$
$$= (\operatorname{supp} \psi)^{d-1}$$
$$\subset (-1, 1)^{d-1}.$$

Thus, assumption in equation 14 is satisfied.

Since $\psi \in C_c^\infty(\mathbb{R}; \mathbb{R})$, $\operatorname{supp} \varphi = (\operatorname{supp} \psi)^{d-1}$ and $\psi(z_j)$ is the projection of $\varphi(\boldsymbol{z})$ on the $j$-component, we have $\varphi \in C_c^\infty(\mathbb{R}^{d-1}; \mathbb{R})$. Then, $\|\nabla \varphi\|_2 < \infty$, and the mean value theorem implies

$$|\varphi(\boldsymbol{z}) - \varphi(\boldsymbol{w})| \leq K_\varphi \|\boldsymbol{z} - \boldsymbol{w}\|_2 \quad \text{for all} \quad \boldsymbol{z}, \boldsymbol{w} \in [0, 1]^{d-1}, \quad \text{where} \quad K_\varphi := \sup_{\boldsymbol{v} \in \mathbb{R}^{d-1}} \|\nabla \varphi(\boldsymbol{v})\|_2,$$

i.e., $\varphi$ is a Hölder (Lipschitz) continuous function with exponent $\alpha = 1$ and constant $K_\varphi > 0$ with respect to the Euclidean norm. Moreover,

- If $\boldsymbol{z} \in [-1/2, 1/2]^{d-1}$, we know that

$$|\varphi(\boldsymbol{z}) - \varphi(\boldsymbol{0})| = |1 - 1| = 0 \leq C_\varphi \|\boldsymbol{z}\|_\infty \quad \text{for all} \quad C_\varphi > 0.$$

- If $\boldsymbol{z} \in \mathbb{R}^{d-1} \setminus [-1/2, 1/2]^{d-1}$, we have $\|\boldsymbol{z}\|_\infty > 1/2$ and

$$|\varphi(\boldsymbol{z}) - \varphi(\boldsymbol{0})| = |\varphi(\boldsymbol{z}) - 1| \leq 1 < 2 \|\boldsymbol{z}\|_\infty.$$

Here, we used that $0 \leq \varphi(\boldsymbol{z}) \leq 1$ for all $\boldsymbol{z} \in \mathbb{R}^{d-1}$.

Then, from the previous two items we get

$$|\varphi(\boldsymbol{z}) - \varphi(\boldsymbol{0})| \leq C_\varphi \|\boldsymbol{z}\|_\infty \quad \text{with} \quad C_\varphi := 2, \quad \text{for all} \quad \boldsymbol{z} \in \mathbb{R}^{d-1}, \tag{100}$$

which implies equation 12 with $\alpha = 1$. This completes the proof of the conditions on $\varphi$ in Construction 10.

In what follows, we justify the choice of $C_{\mathscr{C}}$ so as to satisfy the hypotheses in equation 19. Let $\boldsymbol{\theta} = (\theta_1, \ldots, \theta_m)$ be an arbitrary element of $\Theta$. By equation 15 and equation 16, we have

$$b_{\boldsymbol{\theta}}(\boldsymbol{z}) = b_0(\boldsymbol{z}) + C_{\mathscr{C}} \sum_{j=1}^m \theta_j \varphi\left(M(\boldsymbol{z} - \boldsymbol{v}_j/M)\right) \quad \text{for all} \quad \boldsymbol{z} \in [0, 1]^{d-1}. \tag{101}$$

By Barron definition in equation 11 and since $b_0 \in \mathscr{C}$, there exist $C_0 > 0$, $c_0 \in [-C_0, C_0]$ and a measurable function $F_0 : \mathbb{R}^{d-1} \to \mathbb{C}$ satisfying

$$b_0(\boldsymbol{z}) = c_0 + \int_{\mathbb{R}^{d-1}} (e^{i\boldsymbol{z} \cdot \boldsymbol{\xi}} - 1) F_0(\boldsymbol{\xi}) d\boldsymbol{\xi} \quad \text{and} \quad \int_{\mathbb{R}^{d-1}} \|\boldsymbol{\xi}\|_1 |F_0(\boldsymbol{\xi})| d\boldsymbol{\xi} \leq C_0 \leq C. \tag{102}$$

Therefore, we choose $b_0$ so that the corresponding constant $C_0$ satisfies $C_0 \leq C/2$ (for example, we may take $b_0 := 1/2$; or $b_0$ can be chosen depending on $C$ so that this condition holds).

Let $g : \mathbb{R}^{d-1} \to \mathbb{R}$ be defined by[3]

$$g(\boldsymbol{z}) := \sum_{j=1}^m \theta_j \varphi\left(M(\boldsymbol{z} - \boldsymbol{v}_j/M)\right). \tag{103}$$

To prove that $b_{\boldsymbol{\theta}} \in \mathscr{C}$, it suffices to show that

$$g(\boldsymbol{z}) = c_1 + \int_{\mathbb{R}^{d-1}} (e^{i\boldsymbol{z} \cdot \boldsymbol{\xi}} - 1) F_1(\boldsymbol{\xi}) d\boldsymbol{\xi} \quad \text{and} \quad \int_{\mathbb{R}^{d-1}} \|\boldsymbol{\xi}\|_1 |F_1(\boldsymbol{\xi})| d\boldsymbol{\xi} \leq C_1 \leq C_{\mathscr{C}}^{-1} C/2, \tag{104}$$

---

[3] Although $b_{\boldsymbol{\theta}} : [0, 1]^{d-1} \to [0, 1]$, we define $g$ on $\mathbb{R}^{d-1}$ with value in $\mathbb{R}$, in order to use Fourier analysis on the whole space. This causes no ambiguity, since $\varphi$ is already defined on $\mathbb{R}^{d-1}$, satisfies $\operatorname{supp} \varphi \subset (-1, 1)^{d-1}$, and $0 \leq \varphi(\boldsymbol{z}) \leq 1$ for all $\boldsymbol{z} \in \mathbb{R}^{d-1}$.

for some constant $C_1 > 0$, $c_1 \in [-C_1, C_1]$, and some measurable function $F_1 : \mathbb{R}^{d-1} \to \mathbb{C}$. Indeed, equation 101, equation 102, equation 103 and equation 104, imply that

$$b_{\boldsymbol{\theta}}(\boldsymbol{z}) = c + \int_{\mathbb{R}^{d-1}} (e^{i\boldsymbol{z}\cdot\boldsymbol{\xi}} - 1)F(\boldsymbol{\xi})d\boldsymbol{\xi}, \quad \text{with} \quad c := c_0 + C_{\mathscr{C}}c_1 \quad \text{and} \quad F(\boldsymbol{\xi}) := F_0(\boldsymbol{\xi}) + C_{\mathscr{C}}F_1(\boldsymbol{\xi}), \qquad (105)$$

where

$$\begin{aligned}
\int_{\mathbb{R}^{d-1}} \|\boldsymbol{\xi}\|_1 \, |F(\boldsymbol{\xi})| \, d\boldsymbol{\xi} &\leq \int_{\mathbb{R}^{d-1}} \|\boldsymbol{\xi}\|_1 \, |F_0(\boldsymbol{\xi})| \, d\boldsymbol{\xi} + \int_{\mathbb{R}^{d-1}} \|\boldsymbol{\xi}\|_1 \, |C_{\mathscr{C}}F_1(\boldsymbol{\xi})| \, d\boldsymbol{\xi} \\
&\leq C_0 + C_{\mathscr{C}}C_1 \\
&\leq C/2 + C_{\mathscr{C}}C_{\mathscr{C}}^{-1}C/2 \\
&= C,
\end{aligned} \qquad (106)$$

showing that $b_{\boldsymbol{\theta}} \in \mathscr{C}$.

In order to prove equation 104, we define

$$\widetilde{F}(\boldsymbol{\xi}) := \mathscr{F}[g](\boldsymbol{\xi}) = \int_{\mathbb{R}^{d-1}} g(\boldsymbol{z})e^{-2\pi i \boldsymbol{z}\cdot\boldsymbol{\xi}}d\boldsymbol{z} \quad \text{and} \quad F_1(\boldsymbol{\xi}) := (2\pi)^{-(d-1)}\widetilde{F}(\boldsymbol{\xi}/2\pi),$$

where $\mathscr{F}$ denotes the Fourier transform (see (Grafakos, 2014, Definition 2.2.8)). Once the right-hand side of equation 104 is established, the identity on the left-hand side of equation 104 follows from the Fourier inversion formula (see (Grafakos, 2014, Definition 2.2.13)). Indeed, since $\widetilde{F} = \mathscr{F}[g]$, the Fourier inversion formula yields

$$\begin{aligned}
g(\boldsymbol{z}) &= \int_{\mathbb{R}^{d-1}} e^{2\pi i \boldsymbol{z}\cdot\boldsymbol{\xi}}\widetilde{F}(\boldsymbol{\xi})d\boldsymbol{\xi} \\
&= \int_{\mathbb{R}^{d-1}} e^{i\boldsymbol{z}\cdot\boldsymbol{\xi}}(2\pi)^{-(d-1)}\widetilde{F}(\boldsymbol{\xi}/2\pi)d\boldsymbol{\xi} \\
&= \int_{\mathbb{R}^{d-1}} e^{i\boldsymbol{z}\cdot\boldsymbol{\xi}}F_1(\boldsymbol{\xi})d\boldsymbol{\xi} \\
&= c_1 + \int_{\mathbb{R}^{d-1}} (e^{i\boldsymbol{z}\cdot\boldsymbol{\xi}} - 1)F_1(\boldsymbol{\xi})d\boldsymbol{\xi}, \quad \text{with} \quad c_1 := \int_{\mathbb{R}^{d-1}} F_1(\boldsymbol{\xi})d\boldsymbol{\xi},
\end{aligned}$$

where $c_1 \in [-C_1, C_1]$ is well defined with $C_1$ as in equation 104 (see Remark 20).

Then, we proceed to prove that the right-hand side of equation 104 is satisfied. Note that

$$\begin{aligned}
\int_{\mathbb{R}^{d-1}} \|\boldsymbol{\xi}\|_1 \, |F_1(\boldsymbol{\xi})|d\boldsymbol{\xi} &= \int_{\mathbb{R}^{d-1}} \|\boldsymbol{\xi}\|_1 \left| (2\pi)^{-(d-1)}\widetilde{F}(\boldsymbol{\xi}/2\pi) \right| d\boldsymbol{\xi} \\
&= 2\pi \int_{\mathbb{R}^{d-1}} \|\boldsymbol{\xi}\|_1 \, |\widetilde{F}(\boldsymbol{\xi})|d\boldsymbol{\xi} \\
&= 2\pi \int_{\mathbb{R}^{d-1}} \|\boldsymbol{\xi}\|_1 \, |\mathscr{F}[g](\boldsymbol{\xi})|d\boldsymbol{\xi}
\end{aligned}$$

and

$$\mathscr{F}[g](\boldsymbol{\xi}) = \int_{\mathbb{R}^{d-1}} g(\boldsymbol{z}) e^{-2\pi i \boldsymbol{z}\cdot\boldsymbol{\xi}} d\boldsymbol{z}$$

$$= \sum_{j=1}^{m} \theta_j \int_{\mathbb{R}^{d-1}} \varphi\left(M(\boldsymbol{z}-\boldsymbol{v}_j/M)\right) e^{-2\pi i \boldsymbol{z}\cdot\boldsymbol{\xi}} d\boldsymbol{z}$$

$$= M^{-(d-1)} \sum_{j=1}^{m} \theta_j \int_{\mathbb{R}^{d-1}} \varphi\left(\boldsymbol{\omega}\right) e^{-2\pi i ((\boldsymbol{\omega}+\boldsymbol{v}_j)/M)\cdot\boldsymbol{\xi}} d\boldsymbol{\omega}$$

$$= M^{-(d-1)} s_{\boldsymbol{\theta}}(\boldsymbol{\xi}) \int_{\mathbb{R}^{d-1}} \varphi\left(\boldsymbol{\omega}\right) e^{-2\pi i \boldsymbol{\omega}\cdot(\boldsymbol{\xi}/M)} d\boldsymbol{\omega}$$

$$= M^{-(d-1)} \mathscr{F}[\varphi](\boldsymbol{\xi}/M) s_{\boldsymbol{\theta}}(\boldsymbol{\xi}), \quad \text{where} \quad s_{\boldsymbol{\theta}}(\boldsymbol{\xi}) := \sum_{j=1}^{m} \theta_j e^{-2\pi i (\boldsymbol{v}_j/M)\cdot\boldsymbol{\xi}}.$$

Therefore,

$$\int_{\mathbb{R}^{d-1}} \|\boldsymbol{\xi}\|_1 |F_1(\boldsymbol{\xi})| d\boldsymbol{\xi} = 2\pi M^{-(d-1)} \int_{\mathbb{R}^{d-1}} \|\boldsymbol{\xi}\|_1 |\mathscr{F}[\varphi](\boldsymbol{\xi}/M) s_{\boldsymbol{\theta}}(\boldsymbol{\xi})| d\boldsymbol{\xi}$$

and considering the partition $\{U_{\boldsymbol{k}}\}_{\boldsymbol{k}\in\mathbb{Z}^{d-1}}$ of $\mathbb{R}^{d-1}$, defined by

$$U_{\boldsymbol{k}} := [0,M)^{d-1} + M\boldsymbol{k} = \left\{\boldsymbol{x} + M\boldsymbol{k} \mid \boldsymbol{x} \in [0,M)^{d-1}\right\} \quad \text{where} \quad \mathbb{R}^{d-1} = \bigsqcup_{\boldsymbol{k}\in\mathbb{Z}^{d-1}} U_{\boldsymbol{k}},$$

we get

$$\int_{\mathbb{R}^{d-1}} \|\boldsymbol{\xi}\|_1 |F_1(\boldsymbol{\xi})| d\boldsymbol{\xi} = 2\pi M^{-(d-1)} \sum_{\boldsymbol{k}\in\mathbb{Z}^{d-1}} \int_{U_{\boldsymbol{k}}} \|\boldsymbol{\xi}\|_1 |\mathscr{F}[\varphi](\boldsymbol{\xi}/M) s_{\boldsymbol{\theta}}(\boldsymbol{\xi})| d\boldsymbol{\xi}. \tag{107}$$

We make the following observations.

- Note that

$$\int_{U_{\boldsymbol{k}}} |s_{\boldsymbol{\theta}}(\boldsymbol{\xi})|^2 d\boldsymbol{\xi} = \int_{U_{\boldsymbol{k}}} s_{\boldsymbol{\theta}}(\boldsymbol{\xi}) \overline{s_{\boldsymbol{\theta}}(\boldsymbol{\xi})} d\boldsymbol{\xi}$$

$$= \int_{U_{\boldsymbol{k}}} \left(\sum_{j=1}^{m} \theta_j e^{-2\pi i (\boldsymbol{v}_j/M)\cdot\boldsymbol{\xi}}\right) \left(\sum_{l=1}^{m} \theta_l e^{2\pi i (\boldsymbol{v}_l/M)\cdot\boldsymbol{\xi}}\right) d\boldsymbol{\xi}$$

$$= \int_{U_{\boldsymbol{k}}} \sum_{j=1}^{m}\sum_{l=1}^{m} \theta_j \theta_l e^{\frac{2\pi i}{M}(\boldsymbol{v}_l-\boldsymbol{v}_j)\cdot\boldsymbol{\xi}} d\boldsymbol{\xi}$$

$$\leq \sum_{j,l=1}^{m} \int_{U_{\boldsymbol{k}}} e^{\frac{2\pi i}{M}(\boldsymbol{v}_l-\boldsymbol{v}_j)\cdot\boldsymbol{\xi}} d\boldsymbol{\xi}$$

$$= \sum_{j,l=1}^{m} \int_{[0,M)^{d-1}} e^{\frac{2\pi i}{M}(\boldsymbol{v}_l-\boldsymbol{v}_j)\cdot\boldsymbol{\xi}} d\boldsymbol{\xi}, \tag{108}$$

where

$$\int_{[0,M)^{d-1}} e^{\frac{2\pi i}{M}(\boldsymbol{v}_l-\boldsymbol{v}_j)\cdot\boldsymbol{\xi}} d\boldsymbol{\xi} = \prod_{r=1}^{d-1} \int_0^M e^{\frac{2\pi i}{M}(v_{lr}-v_{jr})\xi_r} d\xi_r \tag{109}$$

and

$$\int_0^M e^{\frac{2\pi i}{M}(v_{lr}-v_{jr})\xi_r} d\xi_r = \begin{cases} M & \text{if } v_{lr} = v_{jr} \\ 0 & \text{if } v_{lr} \neq v_{jr}. \end{cases} \tag{110}$$

Then, from equation 108, equation 109 and equation 110, we obtain

$$
\int_{U_{\boldsymbol{k}}} |s_{\boldsymbol{\theta}}(\boldsymbol{\xi})|^2 \, d\boldsymbol{\xi} \le \sum_{j,l=1}^{m} \int_{[0,M)^{d-1}} e^{\frac{2\pi i}{M}(\boldsymbol{v}_l - \boldsymbol{v}_j)\cdot\boldsymbol{\xi}} d\boldsymbol{\xi}
$$

$$
= M^{d-1} \sum_{j=1}^{m} 1
$$

$$
= mM^{d-1}. \tag{111}
$$

- For $\boldsymbol{\xi} \in U_{\boldsymbol{k}}$, we have $Mk_j \le \xi_j < M(k_j+1)$, and this implies $|\xi_j| \le M(|k_j|+1)$. Therefore,

$$
\|\boldsymbol{\xi}\|_1 \le \sum_{j=1}^{d-1} |\xi_j| \le M \sum_{j=1}^{d-1} (|k_j|+1) \le (d-1)M(1+\|\boldsymbol{k}\|_\infty). \tag{112}
$$

In addition, let $j^* \in \{1,\ldots,d-1\}$ satisfy $\|\boldsymbol{k}\|_\infty = |k_{j^*}| = t \ge 1$. Then,

$$
0 < t \le \xi_{j^*}/M < t+1 \quad \text{whenever} \quad k_{j^*} = t
$$

and

$$
0 \le t-1 \le -\xi_{j^*}/M < t \quad \text{whenever} \quad k_{j^*} = -t,
$$

which imply

$$
\|\boldsymbol{\xi}\|_2 /M \ge |\xi_{j^*}|/M \ge t-1 = \|\boldsymbol{k}\|_\infty - 1 \ge 2^{-1}(\|\boldsymbol{k}\|_\infty + 1) - 1
$$

and therefore

$$
(1 + \|\boldsymbol{\xi}\|_2 /M)^{-(d+1)} \le 2^{d+1}(\|\boldsymbol{k}\|_\infty + 1)^{-(d+1)} \quad \text{for all} \quad \boldsymbol{k} \in \mathbb{Z}^{d-1}. \tag{113}
$$

- Since $\varphi \in C_c^\infty(\mathbb{R}^{d-1}; \mathbb{R})$, we know by (Grafakos, 2014, Definition 2.2.1 and Example 2.2.2) that $\varphi$ is in the class of Schwartz functions. Therefore, by (Grafakos, 2014, Remark 2.2.4), there exists a constant $C_d > 0$, such that

$$
|\mathscr{F}[\varphi](\boldsymbol{\xi}/M)| \le C_d(1 + \|\boldsymbol{\xi}/M\|_2)^{-(d+1)}. \tag{114}
$$

Then, from equation 112, equation 113 and equation 114, we have

$$
\int_{U_{\boldsymbol{k}}} \|\boldsymbol{\xi}\|_1^2 |\mathscr{F}[\varphi](\boldsymbol{\xi}/M)|^2 \, d\boldsymbol{\xi} \le ((d-1)M(1+\|\boldsymbol{k}\|_\infty))^2 \int_{U_{\boldsymbol{k}}} |\mathscr{F}[\varphi](\boldsymbol{\xi}/M)|^2 \, d\boldsymbol{\xi}
$$

$$
\le \left( (d-1)M(1+\|\boldsymbol{k}\|_\infty)C_d(1+\|\boldsymbol{\xi}/M\|_2)^{-(d+1)} \right)^2 \int_{U_{\boldsymbol{k}}} 1 d\boldsymbol{\xi}
$$

$$
\le \left( 2^{d+1}(d-1)MC_d(1+\|\boldsymbol{k}\|_\infty)^{-d} \right)^2 M^{d-1}. \tag{115}
$$

In conclusion, by equation 111 and equation 115, we may apply Hölder's inequality in $L^2$ to equation 107 as follows,

$$
\int_{\mathbb{R}^{d-1}} \|\boldsymbol{\xi}\|_1 |F_1(\boldsymbol{\xi})| d\boldsymbol{\xi} = 2\pi M^{-(d-1)} \sum_{\boldsymbol{k} \in \mathbb{Z}^{d-1}} \int_{U_{\boldsymbol{k}}} \|\boldsymbol{\xi}\|_1 |\mathscr{F}[\varphi](\boldsymbol{\xi}/M)s_{\boldsymbol{\theta}}(\boldsymbol{\xi})| \, d\boldsymbol{\xi}
$$

$$
\le 2\pi M^{-(d-1)} \sum_{\boldsymbol{k} \in \mathbb{Z}^{d-1}} \left( \int_{U_{\boldsymbol{k}}} \|\boldsymbol{\xi}\|_1^2 |\mathscr{F}[\varphi](\boldsymbol{\xi}/M)|^2 \, d\boldsymbol{\xi} \right)^{1/2} \left( \int_{U_{\boldsymbol{k}}} |s_{\boldsymbol{\theta}}(\boldsymbol{\xi})|^2 \, d\boldsymbol{\xi} \right)^{1/2}
$$

$$
\le 2\pi \sum_{\boldsymbol{k} \in \mathbb{Z}^{d-1}} \left( \left( 2^{d+1}(d-1)MC_d(1+\|\boldsymbol{k}\|_\infty)^{-d} \right) M^{-(d-1)/2} \right) (mM^{d-1})^{1/2}
$$

$$
= 2^{d+2}\pi C_d(d-1)M\sqrt{m} \sum_{\boldsymbol{k} \in \mathbb{Z}^{d-1}} (1+\|\boldsymbol{k}\|_\infty)^{-d}
$$

$$
= 2^{(d+5)/2}\pi C_d(d-1)M^{(d+1)/2} \sum_{\boldsymbol{k} \in \mathbb{Z}^{d-1}} (1+\|\boldsymbol{k}\|_\infty)^{-d}. \tag{116}
$$

Moreover, let $\{T_t\}_{t=0}^{\infty}$ be the partition of $\mathbb{Z}^{d-1}$, defined by

$$\mathbb{Z}^{d-1} = \bigsqcup_{t=0}^{\infty} T_t \quad \text{where} \quad T_t = \left\{ \boldsymbol{k} \in \mathbb{Z}^{d-1} \mid \|\boldsymbol{k}\|_{\infty} = t \right\}.$$

Then,

$$\sum_{\boldsymbol{k} \in \mathbb{Z}^{d-1}} (1 + \|\boldsymbol{k}\|_{\infty})^{-d} = \sum_{t=0}^{\infty} \sum_{\boldsymbol{k} \in T_t} (1 + \|\boldsymbol{k}\|_{\infty})^{-d}$$

$$= \sum_{t=0}^{\infty} (1+t)^{-d} \sum_{\boldsymbol{k} \in T_t} 1$$

$$= \sum_{t=0}^{\infty} (1+t)^{-d} \, \#T_t$$

$$= 1 + \sum_{t=1}^{\infty} (1+t)^{-d} \, \#T_t,$$

where $\#T_0 = 1$ and for all $t \geq 1$,

$$\#T_t = \# \left( \left\{ \boldsymbol{k} \in \mathbb{Z}^{d-1} \mid \|\boldsymbol{k}\|_{\infty} \leq t \right\} \setminus \left\{ \boldsymbol{k} \in \mathbb{Z}^{d-1} \mid \|\boldsymbol{k}\|_{\infty} \leq t-1 \right\} \right)$$

$$= (2t+1)^{d-1} - (2(t-1)+1)^{d-1}$$

$$\leq 2 \sum_{r=0}^{d-2} (2t+1)^{d-2-r} (2t-1)^r$$

$$\leq 2 \sum_{r=0}^{d-2} (2t+1)^{d-2}$$

$$= 2(d-1)(2t+1)^{d-2}$$

$$\leq 3^{d-1}(d-1)t^{d-2}.$$

Therefore,

$$\sum_{\boldsymbol{k} \in \mathbb{Z}^{d-1}} (1 + \|\boldsymbol{k}\|_{\infty})^{-d} = 1 + \sum_{t=1}^{\infty} (1+t)^{-d} \, \#T_t$$

$$\leq 1 + 3^{d-1}(d-1) \sum_{t=1}^{\infty} (1+t)^{-d} \, t^{d-2}$$

$$\leq 1 + 3^{d-1}(d-1) \sum_{t=1}^{\infty} t^{-2}$$

$$= 1 + 3^{d-1}(d-1)(\pi^2/6). \tag{117}$$

Combining equation 116 and equation 117, we obtain

$$\int_{\mathbb{R}^{d-1}} \|\boldsymbol{\xi}\|_1 \, |F_1(\boldsymbol{\xi})| d\boldsymbol{\xi} \leq 2^{(d+5)/2} \pi C_d (d-1) M^{(d+1)/2} \sum_{\boldsymbol{k} \in \mathbb{Z}^{d-1}} (1 + \|\boldsymbol{k}\|_{\infty})^{-d}$$

$$= \widetilde{C}_d M^{(d+1)/2},$$

where

$$\widetilde{C}_d := 2^{(d+5)/2} \pi C_d (d-1)(1 + 3^{d-1}(d-1)(\pi^2/6)).$$

Then, in the right-hand side of equation 104,

$$\text{we set} \quad C_1 := \widetilde{C}_d M^{(d+1)/2} \quad \text{and choose} \quad C_{\mathscr{C}} := (C \widetilde{C}_d^{-1}/2) M^{-(d+1)/2}, \tag{118}$$

so that equation 104 is satisfied.

**Remark 20.** *Note that $C_1 = \widetilde{C}_d M^{(d+1)/2}$ as in equation 118, also satisfies*

$$\int_{\mathbb{R}^{d-1}} |F_1(\boldsymbol{\xi})| d\boldsymbol{\xi} < C_1.$$

*Indeed, using the same argument applied to obtain equation 107 but now without the factor $\|\boldsymbol{\xi}\|_1$, we get*

$$\int_{\mathbb{R}^{d-1}} |F_1(\boldsymbol{\xi})| d\boldsymbol{\xi} = 2\pi M^{-(d-1)} \sum_{\boldsymbol{k} \in \mathbb{Z}^{d-1}} \int_{U_{\boldsymbol{k}}} |\mathscr{F}[\varphi](\boldsymbol{\xi}/M) s_{\boldsymbol{\theta}}(\boldsymbol{\xi})| \, d\boldsymbol{\xi}.$$

*By equation 111, equation 113, and equation 114, we apply Hölder's inequality in $L^2$ to the terms in the sum on the right-hand side of the previous equation and obtain*

$$
\begin{aligned}
\int_{\mathbb{R}^{d-1}} |F_1(\boldsymbol{\xi})| d\boldsymbol{\xi} &\leq 2\pi M^{-(d-1)} \sum_{\boldsymbol{k} \in \mathbb{Z}^{d-1}} \left( \int_{U_{\boldsymbol{k}}} |\mathscr{F}[\varphi](\boldsymbol{\xi}/M)|^2 \, d\boldsymbol{\xi} \right)^{1/2} \left( \int_{U_{\boldsymbol{k}}} |s_{\boldsymbol{\theta}}(\boldsymbol{\xi})|^2 \, d\boldsymbol{\xi} \right)^{1/2} \\
&\leq 2\pi M^{-(d-1)} \sqrt{m} M^{(d-1)/2} \sum_{\boldsymbol{k} \in \mathbb{Z}^{d-1}} \left( \int_{U_{\boldsymbol{k}}} |\mathscr{F}[\varphi](\boldsymbol{\xi}/M)|^2 \, d\boldsymbol{\xi} \right)^{1/2} \\
&\leq 2^{d+2} \pi \sqrt{m} M^{-(d-1)/2} C_d \sum_{\boldsymbol{k} \in \mathbb{Z}^{d-1}} (\|\boldsymbol{k}\|_\infty + 1)^{-(d+1)} \left( \int_{U_{\boldsymbol{k}}} 1 d\boldsymbol{\xi} \right)^{1/2} \\
&< 2^{d+2} \pi \sqrt{m} C_d \sum_{\boldsymbol{k} \in \mathbb{Z}^{d-1}} (\|\boldsymbol{k}\|_\infty + 1)^{-d}.
\end{aligned}
$$

*Then, the above inequality and equation 117 imply*

$$\int_{\mathbb{R}^{d-1}} |F_1(\boldsymbol{\xi})| d\boldsymbol{\xi} < 2^{(d+5)/2} \pi C_d \left( 1 + 3^{d-1}(d-1)(\pi^2/6) \right) M^{(d-1)/2} < \widetilde{C}_d M^{(d+1)/2}.$$

In summary, we have proved that $\mathscr{C}_\Theta \subseteq \mathscr{C}$, that is, the left-hand side of equation 19 holds. We now choose $M$ so that the right-hand side of equation 19 is satisfied. Thus, by equation 19 and equation 118, we have

$$2^{\widetilde{\gamma}+6} n C_{\mathscr{C}}^{\widetilde{\gamma}} M^{-(d-1)} = 2^{\gamma+6} (C\widetilde{C}_d^{-1}/2)^\gamma n M^{-(d+1)\gamma/2-(d-1)} \leq 1,$$

and therefore we choose $M$ to be

$$M := 2\lceil M_*/2 \rceil \quad \text{with} \quad M_* := \left( 2^{\gamma+6} (C\widetilde{C}_d^{-1}/2)^\gamma n \right)^{\frac{1}{(d+1)\gamma/2+(d-1)}}. \tag{119}$$

Using the same argument as in equation 93, we get

$$(2\lceil M_*/2 \rceil)^{-\gamma} \geq (2C_* M_*)^{-\gamma} \quad \text{with} \quad C_* := \left( C\widetilde{C}_d^{-1} \right)^{-\frac{\gamma}{(d+1)\gamma/2+(d-1)}} + 1/2. \tag{120}$$

With the choices of $M$ and $C_{\mathscr{C}}$ in equation 118 and equation 119, we know that the left-hand side condition in equation 18 is satisfied. However, for the right-hand side of equation 18 to hold, it is enough to choose a $b_0$ that satisfies it, for instance $b_0 \equiv 1/2$, $b_0$ as in Figure 2, or some more general choice. Therefore, since we have already verified all the hypotheses of Theorem 11, applying it and using equation 20, equation 118,

equation 119 and equation 120, we obtain that

$$
\begin{aligned}
\mathcal{I}_n(\mathscr{C}) &\geq \left(\frac{r^{d-1}}{8^{\widetilde{\gamma}+2}}\right) C_{\mathscr{C}}^{\widetilde{\gamma}} \\
&= \left(\frac{r^{d-1}}{8^{\gamma+2}}\right) (C\widetilde{C}_d^{-1}/2)^\gamma M^{-(d+1)\gamma/2} \\
&= \left(\frac{r^{d-1}}{8^{\gamma+2}}\right) (C\widetilde{C}_d^{-1}/2)^\gamma (2\lceil M_*/2\rceil)^{-(d+1)\gamma/2} \\
&\geq \left(\frac{r^{d-1}}{8^{\gamma+2}}\right) (C\widetilde{C}_d^{-1}/2)^\gamma (2C_* M_*)^{-(d+1)\gamma/2} \\
&= \left(\frac{r^{d-1}}{8^{\gamma+2}}\right) (C\widetilde{C}_d^{-1}/2)^\gamma (2C_*)^{-(d+1)\gamma/2} \left(2^{\gamma+6}(C\widetilde{C}_d^{-1}/2)^\gamma n\right)^{-\frac{(d+1)\gamma/2}{(d+1)\gamma/2+(d-1)}} , \quad \text{for all} \quad n \in \mathbb{N},
\end{aligned}
$$

where $r = \min\{1, (2C_\varphi)^{-1/\alpha}\} = \min\{1, 4^{-1}\} = 4^{-1}$ (see equation 100). In other words,

$$
\mathcal{I}_n(\mathscr{C}) \geq c_{d,C,\gamma}\, n^{-\frac{\gamma}{\gamma+\left(\frac{2(d-1)}{d+1}\right)}}, \quad \text{for all} \quad n \in \mathbb{N},
$$

where

$$
c_{d,C,\gamma} := \left(\frac{4^{-(d-1)}}{8^{\gamma+2}}\right) (C\widetilde{C}_d^{-1}/2)^\gamma (2C_*)^{-(d+1)\gamma/2} \left(2^{\gamma+6}(C\widetilde{C}_d^{-1}/2)^\gamma\right)^{-\frac{(d+1)\gamma/2}{(d+1)\gamma/2+(d-1)}}
$$

is a constant depending only on $d$, $C$ and $\gamma$. □

### C.3 Convex-Lipschitz class of functions

In this case, we take $b_0 \in \mathscr{C} := \mathcal{C}$ for Construction 10, where $\mathcal{C}$ is defined in Section 1.3. It is well known that a convex function admits subgradients at every point in the interior of its domain (see (Bubeck, 2015, Proposition 1.1)). Since $\boldsymbol{v}_j/M \in (0,1)^{d-1}$, we may choose a subgradient $\boldsymbol{g}_j \in \mathbb{R}^{d-1}$ of $b_0$ at $\boldsymbol{v}_j/M$, that is,

$$
b_0(\boldsymbol{z}) \geq \tau_j(\boldsymbol{z}) := b_0(\boldsymbol{v}_j/M) + \boldsymbol{g}_j \cdot (\boldsymbol{z} - \boldsymbol{v}_j/M) \quad \text{for all} \quad \boldsymbol{z} \in [0,1]^{d-1}.
$$

Note that $\tau_j$ defines a supporting hyperplane to the epigraph of $b_0$ at $(\boldsymbol{v}_j/M, b_0(\boldsymbol{v}_j/M))$ (see Bubeck (2015)). If we choose $b_0$ such that

$$
b_0(\boldsymbol{z}) > \tau_j(\boldsymbol{z}) \quad \text{for all} \quad \boldsymbol{z} \in \mathbb{R}^{d-1} \setminus \{\boldsymbol{v}_j/M\}, \tag{121}
$$

then we can apply a sufficiently small vertical shift to $\tau_j$ by adding some constant $\delta > 0$ satisfying

$$
\left\{\boldsymbol{z} \in [0,1]^{d-1} \mid \tau_j(\boldsymbol{z}) + \delta > b_0(\boldsymbol{z})\right\} \subseteq \left\{\boldsymbol{z} \in [0,1]^{d-1} \mid \|\boldsymbol{z} - \boldsymbol{v}_j/M\|_\infty < (2M)^{-1}\right\}. \tag{122}
$$

Hence $\tau_j + \delta$ lies above $b_0$ only on a subset of $\operatorname{int} Q_j$ around $\boldsymbol{v}_j/M$, and with the notation $(\cdot)_+ := \max\{\cdot, 0\}$, we define

$$
\psi_j(\boldsymbol{z}) := (\tau_j(\boldsymbol{z}) + \delta - b_0(\boldsymbol{z}))_+, \tag{123}
$$

where by equation 122, we get

$$
\operatorname{supp} \psi_j \subseteq \left\{\boldsymbol{z} \in [0,1]^{d-1} \mid \|\boldsymbol{z} - \boldsymbol{v}_j/M\|_\infty \leq (2M)^{-1}\right\} \subset \operatorname{int} Q_j \quad \text{for all} \quad j \in \{1,\ldots,m\}. \tag{124}
$$

With the above definitions, our construction of $b_{\boldsymbol{\theta}}$ with $\boldsymbol{\theta} \in \Theta$ will be similar to the bump-type used in the Hölder and Barron cases. If the $j$-th entry of $\boldsymbol{\theta}$ is 1, then the local affine function $\tau_j(\boldsymbol{z}) + \delta$ is activated on $\operatorname{supp} \psi_j$, replacing $b_0$; whereas if $\theta_j = 0$, the function $b_{\boldsymbol{\theta}}$ remains equal to $b_0$ on $\operatorname{supp} \psi_j$ (see Figure 2).

In order to satisfy equation 121, obtain uniform localized perturbations, and a function $\varphi$ as required in Construction 10, we now fix a specific choice of $b_0 \in \mathscr{C}$, namely[4]

$$
b_0(\boldsymbol{z}) := 1/4 + (d-1)^{-1} \|\boldsymbol{z} - \mathbf{1/2}\|_2^2. \tag{125}
$$

---

[4]This choice of $b_0$ is convex, as it is a translation and a positive multiple of the convex function $\boldsymbol{z} \mapsto \|\boldsymbol{z}\|_2^2$, and it is Lipschitz on $[0,1]^{d-1}$ since its gradient is bounded on $[0,1]^{d-1}$. Here, $\mathbf{1/2} := (1/2,\ldots,1/2) \in \mathbb{R}^{d-1}$.

Since $b_0$ is differentiable on $(0,1)^{d-1}$, the subgradient $\boldsymbol{g}_j$ at $\boldsymbol{v}_j/M$ is uniquely determined and coincides with the gradient of $b_0$ at the same point, that is,

$$\boldsymbol{g}_j = \nabla b_0(\boldsymbol{v}_j/M) = 2(d-1)^{-1}(\boldsymbol{v}_j/M - \boldsymbol{1/2}).$$

Therefore,

$$
\begin{aligned}
\psi_j(\boldsymbol{z}) &= (\tau_j(\boldsymbol{z}) + \delta - b_0(\boldsymbol{z}))_+ \\
&= \left((d-1)^{-1}\|\boldsymbol{v}_j/M - \boldsymbol{1/2}\|_2^2 + \nabla b_0(\boldsymbol{v}_j/M)\cdot(\boldsymbol{z} - \boldsymbol{v}_j/M) + \delta - (d-1)^{-1}\|\boldsymbol{z} - \boldsymbol{1/2}\|_2^2\right)_+ \\
&= \left(-(d-1)^{-1}\left(\|\boldsymbol{z} - \boldsymbol{1/2}\|_2^2 - \|\boldsymbol{v}_j/M - \boldsymbol{1/2}\|_2^2\right) + \nabla b_0(\boldsymbol{v}_j/M)\cdot(\boldsymbol{z} - \boldsymbol{v}_j/M) + \delta\right)_+ \\
&= \left(-(d-1)^{-1}\left(\|\boldsymbol{z} - \boldsymbol{v}_j/M\|_2^2 + 2(\boldsymbol{v}_j/M - \boldsymbol{1/2})(\boldsymbol{z} - \boldsymbol{v}_j/M)\right) + \nabla b_0(\boldsymbol{v}_j/M)\cdot(\boldsymbol{z} - \boldsymbol{v}_j/M) + \delta\right)_+ \\
&= \left(-(d-1)^{-1}\|\boldsymbol{z} - \boldsymbol{v}_j/M\|_2^2 + \delta\right)_+ \\
&= \delta\left(1 - (\delta(d-1))^{-1}M^{-2}\|M(\boldsymbol{z} - \boldsymbol{v}_j/M)\|_2^2\right)_+,
\end{aligned}
$$

where we need to choose $\delta$ satisfying equation 122. In particular, we know from the previous identity that

$$
\begin{aligned}
\left\{\boldsymbol{z} \in [0,1]^{d-1} \mid \tau_j(\boldsymbol{z}) + \delta > b_0(\boldsymbol{z})\right\} &= \left\{\boldsymbol{z} \in [0,1]^{d-1} \mid \|\boldsymbol{z} - \boldsymbol{v}_j/M\|_2 < \sqrt{\delta(d-1)}\right\} \\
&\subseteq \left\{\boldsymbol{z} \in [0,1]^{d-1} \mid \|\boldsymbol{z} - \boldsymbol{v}_j/M\|_\infty < \sqrt{\delta(d-1)}\right\},
\end{aligned}
$$

so we can set $\delta$ such that $\sqrt{\delta(d-1)} = (2M)^{-1}$, i.e., $\delta = (2M)^{-2}(d-1)^{-1}$. Then

$$\psi_j(\boldsymbol{z}) = (2M)^{-2}(d-1)^{-1}\left(1 - 4\|M(\boldsymbol{z} - \boldsymbol{v}_j/M)\|_2^2\right)_+, \tag{126}$$

and to ensure that the hypotheses in Construction 10 hold, we define $\boldsymbol{\varphi}$ in equation 16 by

$$
\begin{aligned}
\boldsymbol{\varphi}(\boldsymbol{z}) &:= (\psi_1(\boldsymbol{z}), \ldots, \psi_m(\boldsymbol{z})) \\
&= (2M)^{-2}(d-1)^{-1}\left(\left(1 - 4\|M(\boldsymbol{z} - \boldsymbol{v}_1/M)\|_2^2\right)_+, \ldots, \left(1 - 4\|M(\boldsymbol{z} - \boldsymbol{v}_m/M)\|_2^2\right)_+\right).
\end{aligned}
\tag{127}
$$

Accordingly, we take

$$C_{\mathscr{C}} := (2M)^{-2}(d-1)^{-1}, \quad \varphi(\boldsymbol{z}) := \left(1 - 4\|\boldsymbol{z}\|_2^2\right)_+, \quad \varphi_j(\boldsymbol{z}) := \varphi(M(\boldsymbol{z} - \boldsymbol{v}_j/M)), \tag{128}$$

and verify the hypotheses of Theorem 11 with these assumptions.

By equation 128, it follows that $\varphi(\boldsymbol{0}) = 1$, $0 \le \varphi(\boldsymbol{z}) \le 1$ for all $\boldsymbol{z} \in \mathbb{R}^{d-1}$, and

$$
\begin{aligned}
\operatorname{supp}\varphi &= \overline{\{\boldsymbol{z} \in \mathbb{R}^{d-1} \mid \varphi(\boldsymbol{z}) > 0\}} \\
&= \{\boldsymbol{z} \in \mathbb{R}^{d-1} \mid \|\boldsymbol{z}\|_2 \le 1/2\} \\
&\subseteq \{\boldsymbol{z} \in \mathbb{R}^{d-1} \mid \|\boldsymbol{z}\|_\infty \le 1/2\} \\
&\subset (-1,1)^{d-1},
\end{aligned}
$$

then equation 14 is satisfied. In addition,

- If $\varphi(\boldsymbol{z}) = \varphi(\boldsymbol{w}) = 0$, we obtain $|\varphi(\boldsymbol{z}) - \varphi(\boldsymbol{w})| = 0$.

- If $\varphi(\boldsymbol{z}), \varphi(\boldsymbol{w}) \ne 0$, we have $\|\boldsymbol{z}\|_2, \|\boldsymbol{w}\|_2 < 1/2$ and

$$
\begin{aligned}
|\varphi(\boldsymbol{z}) - \varphi(\boldsymbol{w})| &\le |\,|1 - 4\|\boldsymbol{z}\|_2^2| - |1 - 4\|\boldsymbol{w}\|_2^2|\,| \\
&\le 4|\,\|\boldsymbol{z}\|_2^2 - \|\boldsymbol{w}\|_2^2\,| \\
&\le 4(|\,\|\boldsymbol{z}\|_2 + \|\boldsymbol{w}\|_2\,|)\,\|\boldsymbol{z} - \boldsymbol{w}\|_2 \\
&\le 4\|\boldsymbol{z} - \boldsymbol{w}\|_2.
\end{aligned}
$$

- If $\varphi(\boldsymbol{z}) \neq 0$ and $\varphi(\boldsymbol{w}) = 0$ (similarly, if $\varphi(\boldsymbol{z}) = 0$ and $\varphi(\boldsymbol{w}) \neq 0$), we get $\|\boldsymbol{z}\|_2 < 1/2 \leq \|\boldsymbol{w}\|_2$ and

$$
\begin{aligned}
|\varphi(\boldsymbol{z}) - \varphi(\boldsymbol{w})| &= 1 - 4\|\boldsymbol{z}\|_2^2 \\
&= (1 + 2\|\boldsymbol{z}\|_2)(1 - 2\|\boldsymbol{z}\|_2) \\
&< 4(1/2 - \|\boldsymbol{z}\|_2) \\
&= 4(\|\boldsymbol{w}\|_2 - \|\boldsymbol{z}\|_2) \\
&\leq 4\|\boldsymbol{z} - \boldsymbol{w}\|_2.
\end{aligned}
$$

Therefore,

$$
|\varphi(\boldsymbol{z}) - \varphi(\boldsymbol{w})| \leq 4\|\boldsymbol{z} - \boldsymbol{w}\|_2 \quad \text{for all} \quad \boldsymbol{z}, \boldsymbol{w} \in \mathbb{R}^{d-1},
$$

which shows that $\varphi$ is a Hölder (Lipschitz) continuous function with exponent $\alpha = 1$ and constant $K_\varphi = 4$, with respect to the Euclidean norm. Moreover, $\varphi$ is Hölder (Lipschitz) continuous at $\boldsymbol{0}$ with exponent $\alpha = 1$ and constant $C_\varphi := 4\sqrt{d-1}$ with respect to the $\ell_\infty$-norm, since

$$
|\varphi(\boldsymbol{z}) - \varphi(\boldsymbol{0})| \leq 4\|\boldsymbol{z}\|_2 \leq 4\sqrt{d-1}\|\boldsymbol{z}\|_\infty \quad \text{for all} \quad \boldsymbol{z} \in \mathbb{R}^{d-1}.
$$

This completes the verification of the assumptions on $\varphi$ required in Construction 10.

Now, by equation 125, we see that

$$
\begin{aligned}
1/4 \leq b_0(\boldsymbol{z}) = 1/4 + (d-1)^{-1}\|\boldsymbol{z} - \boldsymbol{1/2}\|_2^2 \\
\leq 1/4 + \|\boldsymbol{z} - \boldsymbol{1/2}\|_\infty^2 \\
\leq 1/2 \quad \text{for all} \quad \boldsymbol{z} \in [0,1]^{d-1}.
\end{aligned}
$$

With $C_\mathscr{C}$ defined as in equation 128, we obtain

$$
C_\mathscr{C} = (2M)^{-2}(d-1)^{-1} < M^{-1}/4,
$$

and in particular $C_\mathscr{C} < 1/8$, since $M \geq 2$. Then,

$$
C_\mathscr{C} < M^{-1}/4 \quad \text{and} \quad b_0(\boldsymbol{z}) \in [1/4, 1/2] \subset [C_\mathscr{C}, 1 - 3C_\mathscr{C}] \quad \text{for all} \quad \boldsymbol{z} \in [0,1]^{d-1},
$$

which imply equation 18.

Next, we only need to test the hypothesis in equation 19 of Theorem 11. We start with $\mathscr{C}_\Theta \subseteq \mathscr{C}$. From equation 124, equation 126 and equation 128, it follows that $\operatorname{supp} \psi_j = \operatorname{supp} \varphi_j$. Then, using equation 123, equation 124, equation 127 and item 2) of Lemma 16, with the notation

$$
J_1(\boldsymbol{\theta}) := \{j \in \{1, \ldots, m\} \mid \theta_j = 1\},
$$

we obtain by cases that:

- If $\boldsymbol{z} \in \operatorname{supp} \psi_k$ for some $k \in \{1, \ldots, m\}$,

$$
\begin{aligned}
b_{\boldsymbol{\theta}}(\boldsymbol{z}) &= b_0(\boldsymbol{z}) + \boldsymbol{\theta} \cdot \boldsymbol{\varphi}(\boldsymbol{z}) \\
&= b_0(\boldsymbol{z}) + \sum_{j \in J_1(\boldsymbol{\theta})} \psi_j(\boldsymbol{z}) \\
&= \begin{cases} \max\{b_0(\boldsymbol{z}), \tau_k(\boldsymbol{z}) + \delta\} & \text{if } k \in J_1(\boldsymbol{\theta}) \\ b_0(\boldsymbol{z}) & \text{otherwise.} \end{cases}
\end{aligned}
$$

- If $\boldsymbol{z} \notin \operatorname{supp} \psi_j$ for all $j \in \{1, \ldots, m\}$,

$$
\begin{aligned}
b_{\boldsymbol{\theta}}(\boldsymbol{z}) &= b_0(\boldsymbol{z}) + \boldsymbol{\theta} \cdot \boldsymbol{\varphi}(\boldsymbol{z}) \\
&= b_0(\boldsymbol{z}) + \sum_{j=1}^m \theta_j \psi_j(\boldsymbol{z}) \\
&= b_0(\boldsymbol{z}).
\end{aligned}
$$

We also know (see equation 124) that

$$\operatorname{supp}\psi_j = \overline{\{\boldsymbol{z} \in [0,1]^{d-1} \mid \tau_j(\boldsymbol{z}) + \delta > b_0(\boldsymbol{z})\}} \subset \operatorname{int} Q_j,$$

and the sets $Q_j$ form a partition of $[0,1]^{d-1}$ up to boundaries (see Construction 10). Thus,

$$
\begin{aligned}
b_{\boldsymbol{\theta}}(\boldsymbol{z}) &= \begin{cases} \max\{b_0(\boldsymbol{z}), \tau_k(\boldsymbol{z}) + \delta\} & \text{if } \boldsymbol{z} \in \operatorname{supp}\psi_k \text{ for some } k \in J_1(\boldsymbol{\theta}) \\ b_0(\boldsymbol{z}) & \text{otherwise} \end{cases} \\
&= \begin{cases} \max\{b_0(\boldsymbol{z}), \tau_k(\boldsymbol{z}) + \delta\} & \text{if } \boldsymbol{z} \in \operatorname{supp}\psi_k \text{ for some } k \in J_1(\boldsymbol{\theta}) \\ \max\{b_0(\boldsymbol{z}), \tau_k(\boldsymbol{z}) + \delta\} & \text{if } \boldsymbol{z} \notin \operatorname{supp}\psi_k \text{ for all } k \in J_1(\boldsymbol{\theta}) \end{cases} \\
&= \max\left\{b_0(\boldsymbol{z}), \{\tau_k(\boldsymbol{z}) + \delta\}_{k \in J_1(\boldsymbol{\theta})}\right\},
\end{aligned}
$$

where we used that if $\boldsymbol{z} \notin \operatorname{supp}\psi_k$ for all $k \in J_1(\boldsymbol{\theta})$, then $\tau_k(\boldsymbol{z}) + \delta \leq b_0(\boldsymbol{z})$ for all $k \in J_1(\boldsymbol{\theta})$. Therefore, for all $\boldsymbol{z} \in [0,1]^{d-1}$, the function $b_{\boldsymbol{\theta}}$ is the maximum of $b_0$ and the functions $\tau_k + \delta$ with $k \in J_1(\boldsymbol{\theta})$. Since $b_0$ is convex and Lipschitz, and each $\tau_k + \delta$ is affine (hence also convex and Lipschitz), it follows that $b_{\boldsymbol{\theta}}$, as the maximum of convex and Lipschitz functions, is itself convex and Lipschitz. In conclusion,

$$\mathscr{C}_\Theta \subseteq \mathscr{C}.$$

Finally, to choose $M$ that satisfies the right-hand side of equation 19, that is,

$$2^{\widetilde{\gamma}+6} n C_{\mathscr{C}}^{\widetilde{\gamma}} M^{-(d-1)} = 2^{\gamma+6} n ((2M)^{-2}(d-1)^{-1})^\gamma M^{-(d-1)} \leq 1,$$

we take

$$M := 2\lceil M_*/2 \rceil \quad \text{where} \quad M_* := \left(2^{-\gamma+6}(d-1)^{-\gamma} n\right)^{\frac{1}{(d-1)+2\gamma}}. \tag{129}$$

With an argument similar to that used in equation 93, we obtain

$$(2\lceil M_*/2 \rceil)^{-\gamma} \geq (2C_* M_*)^{-\gamma} \quad \text{where} \quad C_* := (2(d-1))^{\frac{\gamma}{(d-1)+2\gamma}} + 1/2.$$

Therefore, since the hypotheses of Theorem 11 are already satisfied, using equation 20, equation 128 and equation 129, we obtain that

$$
\begin{aligned}
\mathcal{I}_n(\mathscr{C}) &\geq \left(\frac{r^{d-1}}{8^{\widetilde{\gamma}+2}}\right) C_{\mathscr{C}}^{\widetilde{\gamma}} \\
&= \left(\frac{r^{d-1}}{8^{\gamma+2}}\right) (2M)^{-2\gamma}(d-1)^{-\gamma} \\
&= \left(\frac{r^{d-1}(d-1)^{-\gamma}}{2^{5\gamma+6}}\right) (2\lceil M_*/2 \rceil)^{-2\gamma} \\
&\geq \left(\frac{r^{d-1}(d-1)^{-\gamma}}{2^{5\gamma+6}}\right) (2C_* M_*)^{-2\gamma} \\
&= c_{d,\gamma}\, n^{-\frac{2\gamma}{(d-1)+2\gamma}} \quad \text{for all} \quad n \in \mathbb{N},
\end{aligned}
$$

where

$$c_{d,\gamma} := \left[\left(\frac{r^{d-1}(d-1)^{-\gamma}}{2^{5\gamma+6}}\right) (2C_*)^{-2\gamma} \left(2^{-\gamma+6}(d-1)^{-\gamma}\right)^{-\frac{2\gamma}{(d-1)+2\gamma}}\right],$$

$r := \min\{1, (2C_\varphi)^{-1}\}$ and $C_\varphi := 4\sqrt{d-1}$. Lastly,

$$\mathcal{I}_n(\mathscr{C}) \geq c_{d,\gamma}\, n^{-\frac{2\gamma}{(d-1)+2\gamma}} \quad \text{for all} \quad n \in \mathbb{N},$$

where $c_{d,\gamma}$ is a constant that depends only on $d$ and $\gamma$. $\qquad\square$

