# OpenReview forum: "Minimax learning rates for estimating binary classifiers under margin conditions"
_TMLR — Decision pending for TMLR_

### Review · Reviewer_zPS3 · 2026-04-07

**Summary Of Contributions:**

This paper studies the minimax estimation of binary classifiers in a noiseless setting under a geometric margin condition. The decision boundary is modeled via horizon functions with regularity (e.g., Hölder, Barron, convex-Lipschitz), and the goal is to characterize the worst-case error of estimating the classifier from i.i.d. samples. The main contribution is the derivation of minimax lower bounds for the estimation error under margin conditions. These bounds are shown to nearly match existing upper bounds in several regimes.

Overall, the problem of establishing minimax lower bounds for binary classification under geometric margin conditions appears to be interesting and important. On the other hand, there remain gaps between upper and lower bounds in certain regimes, particularly for the Barron and Hölder classes as the dimension or smoothness varies, leaving the optimal rates not fully resolved. Additionally, the results are restricted to the noiseless setting, which may limit their applicability to more realistic scenarios where label noise is present.

**Audience:**

Yes

**Audience Explanation:**

This paper would likely interest a subset of TMLR’s audiences, especially those working on learning theory, nonparametric methods, and margin-based analysis.

**Broader Impact Concerns:**

Not that I am aware of.

**Claims And Evidence:**

Yes

**Claims Explanation:**

The minimax lower bounds seem to be supported by the technical development: the paper builds a finite subfamily of the function class via localized perturbations, constructs matching densities satisfying the geometric margin condition, reduces estimation to a Hamming recovery problem, and then applies Assouad’s lemma.

**Requested Changes:**

1. For the Barron class, the lower bound matches the known upper bound up to logarithmic factors in the high-dimensional regime. However, in low dimensions, there appears to be a nontrivial gap between the exponents. Could you comment on whether this gap is fundamental? In particular, is it more plausible that the lower bound can be strengthened, or that the existing upper bounds are suboptimal in this regime?

2. Under the Hölder class, the lower and upper bounds are close when $\alpha$ is near 1, but differ substantially for small $\alpha$. Could you comment on this gap in the low-$\alpha$ regime?

3. The analysis assumes a noiseless setting. While this is theoretically justified, it may limit practical relevance, as real-world data often contains label noise. Could the authors elaborate on the challenges involved in extending these results to settings with (even mild) noise?

---

> ### Author Response · Authors · 2026-06-20
>
> We thank Reviewer zPS3 for the careful reading of our manuscript and for the constructive suggestions. We have carefully considered all of the reviewer’s comments and implemented the following changes in the manuscript:
> 1. We added two new remarks discussing the comparison between our lower bounds and the available upper bounds. For the Hölder case, we explain the origin of the gap that appears when \alpha<1 and discuss a possible approach that could potentially reduce it. For the Barron case, we clarify that our optimality statement concerns the asymptotic behavior of the exponent of n in the high-dimensional regime, discuss related works adopting a similar perspective, and explain why the Barron class overcomes the curse of dimensionality at the level of the learning-rate exponent. We also added further discussion on how the learning rates behave as the margin exponent \gamma varies relative to the dimension.
> 2. We expanded Remark 1 by adding two new items discussing the relevance of the noiseless setting. In particular, we explain why the noiseless case is important in its own right, provide examples of situations where it is a meaningful modeling assumption, and clarify that the lower bounds established in the noiseless setting automatically remain valid for larger classes of problems that allow label noise, since the noiseless setting corresponds to the special case of zero noise.

---

### Review · Reviewer_SHci · 2026-05-11

**Summary Of Contributions:**

This paper derived theoretical (worst case) lower bounds for learning rates in binary classification focusing on a noiseless setting where data distributions satisfy a geometric margin condition. The paper specifically works with decision boundaries belonging to three classes of functions: Barron-regular functions, Holder-continuous functions and convex-Lipschitz functions.

**Additional Comments:**

1. I am somewhat concerned about the bounds derived for the Holder classes. In the regime of small alpha, the gap between the upper and lower bounds becomes quite substantial. This naturally raises the question of whether the current upper bounds are suboptimal, or whether the lower bounds are not tight enough. While the authors acknowledge this limitation, the paper would benefit from a clearer discussion of where the gap may originate and what technical challenges prevent matching rates. Providing a more concrete roadmap for future work aimed at closing this gap would enhance the paper significantly.

2. Barren rate is particurly intersting as it effectively removes explicit dimensional dependence asymptotically, which deserves more emphasis and discussion. Does the constant factors in the rate scale with d? Sometimes the hidden constant explode as d increases.

**Audience:**

Yes

**Audience Explanation:**

As most existing relevant papers relies on noise conditions, the paper addresses a theoretical gap by providing minimax lower bounds in a noiseless setting. Thus this may be of some interests to the researchers working on this field.

**Broader Impact Concerns:**

N.A.

**Claims And Evidence:**

Yes

**Claims Explanation:**

The paper provides a solid theoretical foundation for its claims by using rigorous proofs to establish minimax lower bounds. The proofs mainly rely on Assouad's Lemma, a standard tool for proving lower bounds in statistics.

**Requested Changes:**

While the paper provides rigorous mathematical proofs, the intuition behind the results and their implications for real-world applications remain somewhat unclear to me. In particular, the noiseless setting is a very restrictive assumption. Most practical classification problems (including the MNIST and CIFAR examples illustrated in the paper are inherently noisy). Although the noiseless regime allows the authors to isolate the geometric role of the margin condition and derive clean minimax results, it may also be viewed by more classical statisticians as an overly idealized setting with limited practical relevance. It would be better to at least include a discussion how these geometric lower bounds might soften when the hard margin is replaced by a probabilistic transition. Without this, the practical utility of these rates for real-world classification may be underestimated by classical statisticians.

---

> ### Author Response · Authors · 2026-06-20
>
> We thank Reviewer Shci for the careful reading of our manuscript and for the constructive suggestions. We have carefully considered all of the reviewer’s comments and implemented the following changes in the manuscript:
> 1. We added a new remark at the beginning of the Main Results section to explain the intuition behind our lower-bound construction. Following the reviewer’s suggestion, we also made the construction more accessible by using Figures 2, 3, and 4, which illustrate the main ideas in the case d=2. In the same remark, we briefly explain how the construction leads to the application of Assouad’s lemma.
> 2. We expanded Remark 1 by adding two new items discussing the relevance of the noiseless setting. In particular, we explain why the noiseless case is important in its own right, provide examples of situations where it is a meaningful modeling assumption, and clarify that the lower bounds established in the noiseless setting automatically remain valid for larger classes of problems that allow label noise, since the noiseless setting corresponds to the special case of zero noise.
> 3. We revised the caption of Figure 1 to make the discussion of the margin structure in the considered datasets clearer and more precise.
> 4. We added a new remark immediately after the comparison between our lower bound and the existing upper bound for the Hölder class. There, we explain the origin of the gap that appears when \alpha<1 and discuss a possible approach that could potentially reduce this gap.
> 5. We also added a new remark immediately after the comparison between the Barron upper and lower bounds. In this remark, we clarify that our optimality statement concerns the asymptotic behavior of the exponent of n in the high-dimensional regime, discuss related works where a similar perspective is adopted, and highlight the fact that the Barron class overcomes the curse of dimensionality at the level of the learning-rate exponent. In addition, we added a new item in the penultimate remark before the appendix discussing the dependence of the multiplicative prefactors on the dimension d.

---

### Review · Reviewer_k6uk · 2026-06-08

**Summary Of Contributions:**

This paper studies noiseless classification, where the data is separable according to the geometric margin condition (Eq. (6)). Inuitively, the mass of the marginal distribution on the covariates $X$ which is $\varepsilon$ close to the decision boundary scales with $\varepsilon^\gamma$, for some positive, fixed $\gamma$. Larger values of the margin exponent $\gamma$ imply less mass around the boundary.

The paper proves a minimax lower bound under these hypothesis (Including C1 and C2, which focus the problem on horizon functions with some regularity conditions). In particular, the Authors remark how their main Theorem 8 translates in lower bounds for learning Holder continuous, Barron, and Convex-Lipschitz functions (Corollary 10).

**Audience:**

Yes

**Audience Explanation:**

This work characterizes the statistical complexity of learning separable data according to the assumptions C1-C3. This is a quite standard question in learning theory which does not require further motivation, and the Authors position their paper well within the related literature.

**Claims And Evidence:**

Yes

**Claims Explanation:**

The results seem correct and they are proven in the Appendix. The proof technique follows the construction of a set of classifiers indexed by binary perturbations of the decision boundary, a control of the distance between the projection of the estimator and the elements of this set, and concludes with an application of Assouad’s lemma.

While this approach seems sound (I did not proof read the Appendix), this is the first time I have seen this proof technique based on small localized perturbations of the original function, and from my perspective (which perhaps will not be that far from the one of an average reader), the Authors do not take much effort to explain the intuition behind their method and results.

Then, I believe a major change the Authors should consider doing is adding some material to make Construction 7 more digestible, which, right now, is arguably obscure... Few suggestions are, move Figure 2 earlier in the main body, and discuss heuristically what you are doing in the case $d = 2$. From here, add a proof sketch and how Assouad's Lemma is applied. Only later give the formal and complete construction. Perhaps an intuition of Corollary 10 could also be helpful, and your choices of $M$ and $C_{\mathcal C}$ for different families.

**Requested Changes:**

In the Abstract, the Authors write "almost universally satisfied in practice" referred to their geometric margin condition. I would suggest to rephrase or remove this slightly specultative statement.

I am not sure I understand the sentence "but also reveals that the CIFAR-10 data exhibits a margin, albeit a weaker one." in the Caption of Figure 1. From the figure there is no visible margin no? The evidence suggests that the two classes are not separable, or am I misunderstanding something?

I would perhaps suggest the authors a small paragraph after Corollary 10 on the limit where $\gamma$ grows comparably to $d$ or much faster. In general, I do not see much discussion on the interpretation of Eq. (21-23).

See my point on Construction 7 above.

---

> ### Author Response · Authors · 2026-06-20
>
> We thank Reviewer k6uk for the helpful feedback. We have carefully considered all of the reviewer’s suggestions and implemented the following changes in the manuscript:
>
> 1. We added a new remark at the beginning of the Main Results section to explain the intuition behind our lower-bound construction. Following the reviewer’s suggestion, we also made the construction more accessible by using Figures 2, 3, and 4, which illustrate the main ideas in the case d=2. In the same remark, we briefly explain how the construction leads to the application of Assouad’s lemma.
> 2. We added two paragraphs immediately after the Corollary to provide further intuition and interpretation of the obtained lower bounds. In particular, we discuss the role of the margin exponent, the dependence on the dimension, and the meaning of the different exponents appearing in the Hölder, Barron, and convex-Lipschitz settings. We also explain the intuition behind the choices of the parameters M and C_{\mathscr C} for the different function classes.
> 3. We removed the speculative statement “almost universally satisfied in practice” from the abstract.
> 4. We revised the caption of Figure 1 to make the discussion of the margin structure in the considered datasets clearer and more precise.